# Fine-scale population structure and widespread conservation of genetic effect sizes between human groups across traits

Sile Hu [1,2,3] ✉, Lino A. F. Ferreira [1,2], Sinan Shi [1], Garrett Hellenthal[4,5,9], Jonathan Marchini [6,9], Daniel J. Lawson [7,8,9] ✉ & Simon R. Myers [1,2,9] ✉

Understanding genetic differences between populations is essential for avoiding confounding in genome-wide association studies and improving polygenic score (PGS) portability. We developed a statistical pipeline to infer fine-scale Ancestry Components and applied it to UK Biobank data. Ancestry Components identify population structure not captured by widely used principal components, improving stratification correction for geographically correlated traits. To estimate the similarity of genetic effect sizes between groups, we developed ANCHOR, which estimates changes in the predictive power of an existing PGS in distinct local ancestry segments. ANCHOR infers highly similar (estimated correlation 0.98 ± 0.07) effect sizes between UK Biobank participants of African and European ancestry for 47 of 53 quantitative phenotypes, suggesting that gene–environment and gene–gene interactions do not play major roles in poor cross-ancestry PGS transferability for these traits in the United Kingdom, and providing optimism that shared causal mutations operate similarly in different populations.

Genome-wide association studies (GWAS) have uncovered numerous genetic influences on complex human traits, regulated by many loci with small effect sizes. For traits such as height, large sample sizes in European groups have allowed PGS to explain a substantial fraction of heritability by summing effects across many single nucleotide polymorphisms (SNPs) genome-wide[1]. Because the true causal variants are unknown, SNPs in PGS are expected not to influence a trait directly, but rather correlate with ('tag') a true causal mutation. Genetic stratification is a major confounder in GWAS, potentially causing false positives when phenotypes correlate with stratification. Methods including mixed models[2,3] and principal component analysis[4,5] have proved powerful in solving major issues, but subtle population structure still

biases effect size estimates[6,7], complicating studies of temporal trait evolution[6] and comparisons between human groups. Moreover, PGS accuracy drops in populations with different ancestry from the GWAS[8,9] cohort, particularly those with strong genetic differentiation. This lack of portability is partly caused by genetic drift making some causal variants group-specific, and population-specific linkage disequilibrium (LD) patterns affecting tagging accuracy.

As well as such 'local' differences, recent studies[10–14] have suggested that causal variants have different effect sizes on traits in different groups, because of 'global' (nonlocal) factors. Changes in causal SNP effect sizes, defined as their mean effects on a trait of interest, can arise by either gene–gene interactions not captured by the additive

[1]Department of Statistics, University of Oxford, Oxford, UK. [2]Wellcome Centre for Human Genetics, University of Oxford, Oxford, UK. [3]Human Genetics Centre of Excellence, Novo Nordisk Research Centre Oxford, Oxford, UK. [4]Department of Genetics, Evolution and Environment, University College London, London, UK. [5]UCL Genetics Institute, University College London, London, UK. [6]Regeneron Genetic Center, Tarrytown, NY, USA. [7]Department of Statistical Science, School of Mathematics, University of Bristol, Bristol, UK. [8]MRC Integrative Epidemiology Unit, Population Health Sciences, Bristol Medical School, University of Bristol, Bristol, UK. [9]These authors contributed equally: Garrett Hellenthal, Jonathan Marchini, Daniel J. Lawson, Simon R. Myers. ✉e-mail: leunghom@gmail.com; dan.lawson@bristol.ac.uk; myers@stats.ox.ac.uk

model, or gene–environment interactions, differing across populations. Varying effect sizes have been documented in populations[9], so potentially contribute to differences between populations. However, other studies suggest similarities across groups in underlying effect sizes[15–19] and, for stronger GWAS hits, in direction of effect at least[20].

One recent study[14] leverages individuals of mixed African and European ancestry, decomposing local ancestry to test whether causal variants on African versus European chromosomes show different effect sizes. By focusing on within-individual comparisons, this approach eliminated factors including gene–environment interactions[14,21] and long-range gene–gene interactions[21] that might differ across populations. It instead examines whether, for example, local gene–gene interactions alter effect sizes on different ancestral backgrounds. The study inferred strong sharing of underlying effect sizes at this within-individual level, suggesting local interactions are not major factors for most traits. However, it also inferred strong differences in causal effects across individuals possessing different continental ancestries, possibly because of gene–environment interactions. Resolving the role of such interactions is vital to successfully apply genetic findings across groups, design efficient studies and understand whether evolution or environmental differences drive interpopulation trait variation. Here, we develop an approach to do this, using models in which genetic effect sizes are shared across ancestries within individuals (as observed in ref. 14), but may vary between individuals with mainly European or African ancestry.

To analyze the impacts of population structure, we introduce a fine-scale ancestry pipeline. We use 'ancestry' throughout to refer to sharing inferred most-recent ancestors with someone from a particular self-reported ethnicity or geographic region. Expanding on our previous work[22–24], we created a pipeline inferring individuals' ancestry contributions from 127 geographically meaningful and genetically recoverable (Methods) regions worldwide. We applied this pipeline to all 487,409 participating UK Biobank[25] (UKB) individuals. We show that using detailed ancestry information in GWAS better corrects for population stratification versus other state-of-the-art methods[2,4,26], reduces likely false positives and uncovers previously undiscovered associations, but nonportability remains strong. We also develop ANCHOR, a statistical inference approach to estimating cross-population similarity in causal effect sizes using admixed individuals, complementing approaches including POPCORN[10] and XPASS[27] that only apply to nonadmixed individuals ('Discussion'). Notably, ANCHOR requires no prior assumptions about the underlying effect size distribution. As in ref. 14 and other recent studies[11–13], ANCHOR decomposes ancestry at fine scales along the genome. It assigns mutations to specific ancestries to quantify the contributions of gene–environment interactions on differences of predictive power for individuals with different ancestries. Application of ANCHOR to 8,003 UKB mixed-ancestry individuals yields different findings from recent studies, which we discuss along with implications.

## Results

### A statistical pipeline to infer precise individual ancestry
We developed an approach able to decompose the fine-scale ancestry of a genome into a mix of 127 regions, including 23 in the United Kingdom and Ireland (Supplementary Table 1). The UKB dataset comprises many individuals of mainly British ancestry, alongside a substantial fraction with ancestry from elsewhere. Our approach identified 105 regions present in at least 5 individuals for at least 10% of their ancestry. Our pipeline leverages data from previous studies[23,28–31] of human genetic variation, and uses methods[25,32] to impute these data on a common set of variants for high-quality imputation. This generates a unified 'reference panel' of haplotypes (Methods and Fig. 1a), phased using SHAPEIT2 (ref. 32) to ensure consistent data quality. Reference haplotypes are labeled through semi-supervised clustering with ChromoPainter[33] and fineSTRUCTURE[33], combined with geographic and ethnicity labels, to produce a painting reference panel.

Our pipeline uses this panel to generate an ancestry decomposition for a new 'target' sample (Fig. 1a), extending our previous approaches[22,33], by closely mirroring the steps used to generate the reference panel itself. Unified panel markers are phased and imputed in the target, using the phasing and imputation panel. For UKB, haplotype data are prephased. Imputed haplotypes are then matched against the labeled painting reference panel using ChromoPainter to quantify recent ancestor sharing. Finally, a non-negative least square (NNLS) based approach (Methods)[22] is used to infer ancestry coefficients from ChromoPainter output by fitting the observed haplotype-matching vector as a mixture of those from the 127 reference panel groups. This approach leverages haplotype information to infer population structure[33] alongside Hidden Markov Model approximations to the coalescent[33], analyzing each sample in minutes (Methods).

We applied our pipeline to 487,409 UKB participants. Because our pipeline uses 'out-of-sample' comparisons based on population structure, analyzing individuals independently, it captures population structure information in the United Kingdom and Ireland but not genetic relatedness between samples. Ancestry is defined through subjective groupings of reference individuals using genetic clusters identifiable via fineSTRUCTURE. Because the true genetic ancestry of UKB individuals is unknown, we assessed the pipeline's effectiveness by examining the relationship between inferred ancestry and individuals' birthplace and self-reported ethnicity, the ability of Ancestry Components (ACs) to predict principal components (PCs) and the capacity of our ACs to correct for ancestry in GWAS, compared with widely used approaches including PCs and BOLT-LMM[4,34]. We also developed an expectation-maximization algorithm (Methods and Supplementary Note 1) to estimate allele frequencies across SNPs. This method provides frequency estimates for each of 25,485,700 UKB imputed mutations[25] within 127 reference groups, aiding studies of regional allele frequencies within the United Kingdom.

### Fine-scale population structure across the United Kingdom
As a first test, the mean ancestries for the 434,781 UKB participants born in the United Kingdom or Republic of Ireland with self-reported white British and/or Irish (WBI) ethnicity were: British–Irish (BI), 94.9%; Dutch, 1.35%; Swiss, 0.79%; Norwegian, 0.49%; Polish, 0.29%; and Danish, 0.19%. For participants whose self-reported ethnicity is 'other white background', the BI proportion drops to 25.5%. For individuals born in the Republic of Ireland, inferred Irish ancestry averages 74.2%, within 98.4% BI ancestry, demonstrating the pipeline's accuracy in capturing geographic and ethnicity information (Extended Data Fig. 1).

The average ancestry proportion from 22 British Isles regions (Methods) varies by birthplace (Fig. 2a). These ancestry proportions are based on an out-of-sample dataset of UK individuals with grandparents born within an ~80-km radius in each region[23]. Mean BI ancestry associated with a region decreases with geographic distance from that region, indicating that DNA information is informative for birthplace (Fig. 1b and Extended Data Fig. 1). Some 41.5% of UK-born individuals have more than 50% of their ancestry from a single region, matching their birthplace 59.2% of the time, rising to 82.7% after expanding to neighboring regions (Supplementary Table 2a). There is a strong correspondence between self-identified ethnicity and birthplace for non-UK ancestries (Fig. 2b and Extended Data Fig. 2). However, regional variation exists: ancestry localization is weaker in southern and eastern England[23], and stronger in Scotland, Wales, Northern England and South West England. An entropy-based statistic (Methods) shows regional mixing, with London having the highest entropy and the highest average fraction of ancestry outside the United Kingdom. Other major cities and South East England also show strong mixing (Extended Data Fig. 3). Non-British ancestries also vary geographically, with higher Irish ancestry (>10%) seen in Liverpool, Birmingham, Manchester and London; Dutch ancestry found in the south of England and around Bristol (3.5%); and Polish

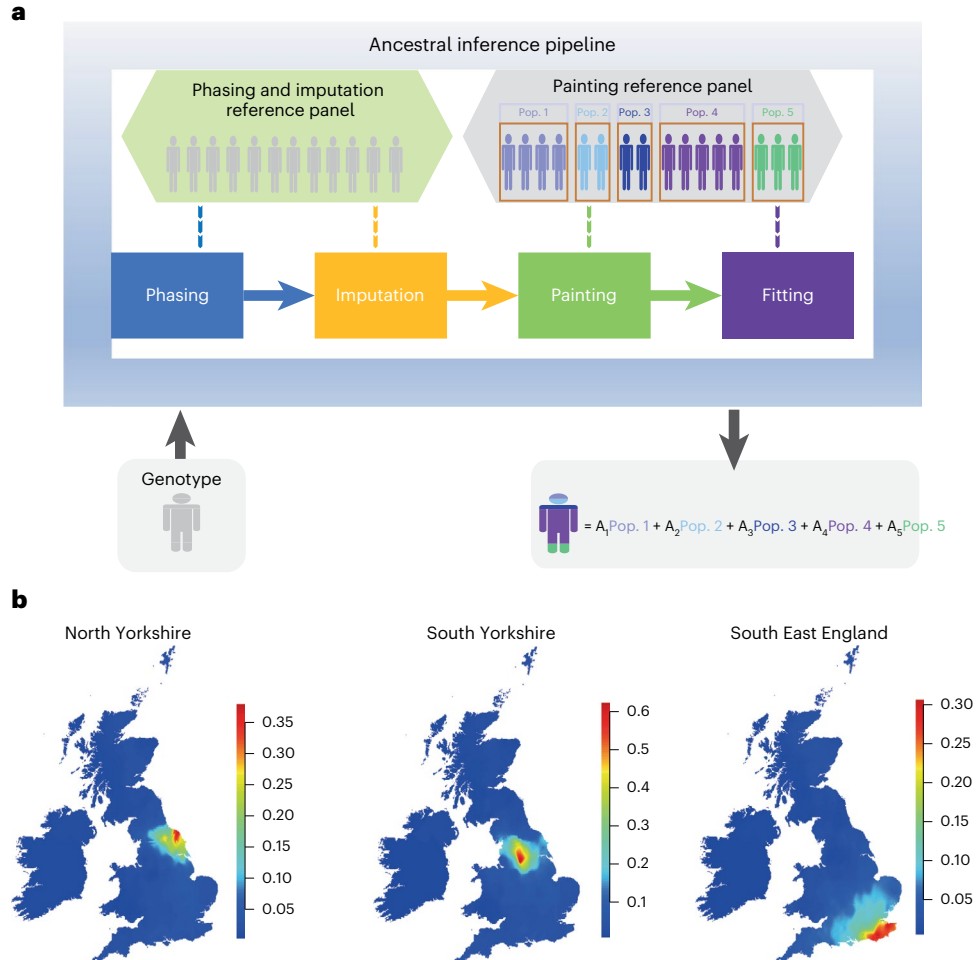

**Fig. 1 | Schematic diagram of the ancestral inference pipeline and the performance in the UKB BI individuals. a**, Main steps in the ancestral inference pipeline. The pipeline accepts individual genotype data (microarray or sequencing data) as input. The genotype data are phased and imputed against a phasing and imputation reference panel in first 'Phasing' and 'Imputation' stages, and painted against the painting reference panel including preclustered groupings (5 groups in this example; 127 in our actual analysis). In a final mixture fitting stage, non-negative coefficients summing to one and representing the proportions of ancestries from the labeled groups in the reference panel are inferred. **b**, Geographical average proportion of DNA in BI individuals, positioned according to their birthplaces (Methods), inferred to come from three regional groupings: North Yorkshire, South Yorkshire and South East England (which correspond to the excess ancestry locations colored red). Pop., population; A, proportion of ancestries from each population.

ancestry peaking near Wrexham in Wales, the site of the second highest concentration of Polish burials in the United Kingdom[35].

For non-UK-born individuals, we again achieve fine-scale ancestry resolution (Fig. 2b, Extended Data Fig. 2 and Supplementary Table 2b). Although some countries (for example, Philippines and Japan) show genetic homogeneity, perhaps because of small sample sizes, most exhibit diverse ancestry patterns. Because non-UK-born individuals reflect people who made their home in the United Kingdom, rather than unbiased samples from birthplace countries, we observe widespread BI ancestry. Other patterns are present, such as the over-representation of Gujarat ancestry among UKB individuals born in Uganda and Kenya, which might be explained by the post-colonial exodus of South Asians from East Africa.

### ACs aid GWAS stratification for geographically linked traits

To avoid false-positive GWAS associations, correcting for population stratification is crucial. PCs are widely used for this purpose[4,5,36], either alone or with mixed-model analysis[3,34]. However, selecting the number of PCs is nontrivial; too few can result in false positives, whereas too many can cause overcorrection because of extensive LD (for example, beyond the first 40 publicly released UKB PCs[36]). We compared the use of 127 ACs with PCs in GWAS. First, we predicted the first 16 UKB

PCs from these 127 ACs, and conversely predicted 16 common BI ACs from the first 140 UKB PCs (Methods, Fig. 3a,b and Supplementary Figs. 1 and 2). We found a good prediction of most PCs from ACs, but not the converse, indicating that ACs capture additional information (Supplementary Fig. 1). PC-based prediction was also poor for non-UK regions (Supplementary Fig. 2). Second, we performed GWAS on 104 UKB quantitative traits with more than 10,000 unrelated white British individuals (Methods). We compared the effectiveness of ACs versus PCs for correcting population stratification by using LD-score regression (LDSC; Extended Data Fig. 4)[37], which measures systematic inflation because of uncorrected stratification. An intercept estimate close to 1 indicates effective correction, although intercepts slightly above 1 can occur for highly heritable traits like height, or large sample sizes in practice[37,38].

Birth location is often used as a control phenotype to test for stratification[39] because few SNPs are likely to be causally associated with it. For latitude (Fig. 3c), PC-based correction or BOLT-LMM analysis (Fig. 3c, Supplementary Fig. 3a and Supplementary Table 3a,b) yielded many apparent independent hits, and substantial genome-wide inflation (LDSC intercept of 1.6608; Extended Data Fig. 4a,b), even with 100 PCs (Supplementary Fig. 3b). By contrast, AC-based correction reduced the number of association signals from 470 to 7 (Fig. 3c; with five shared

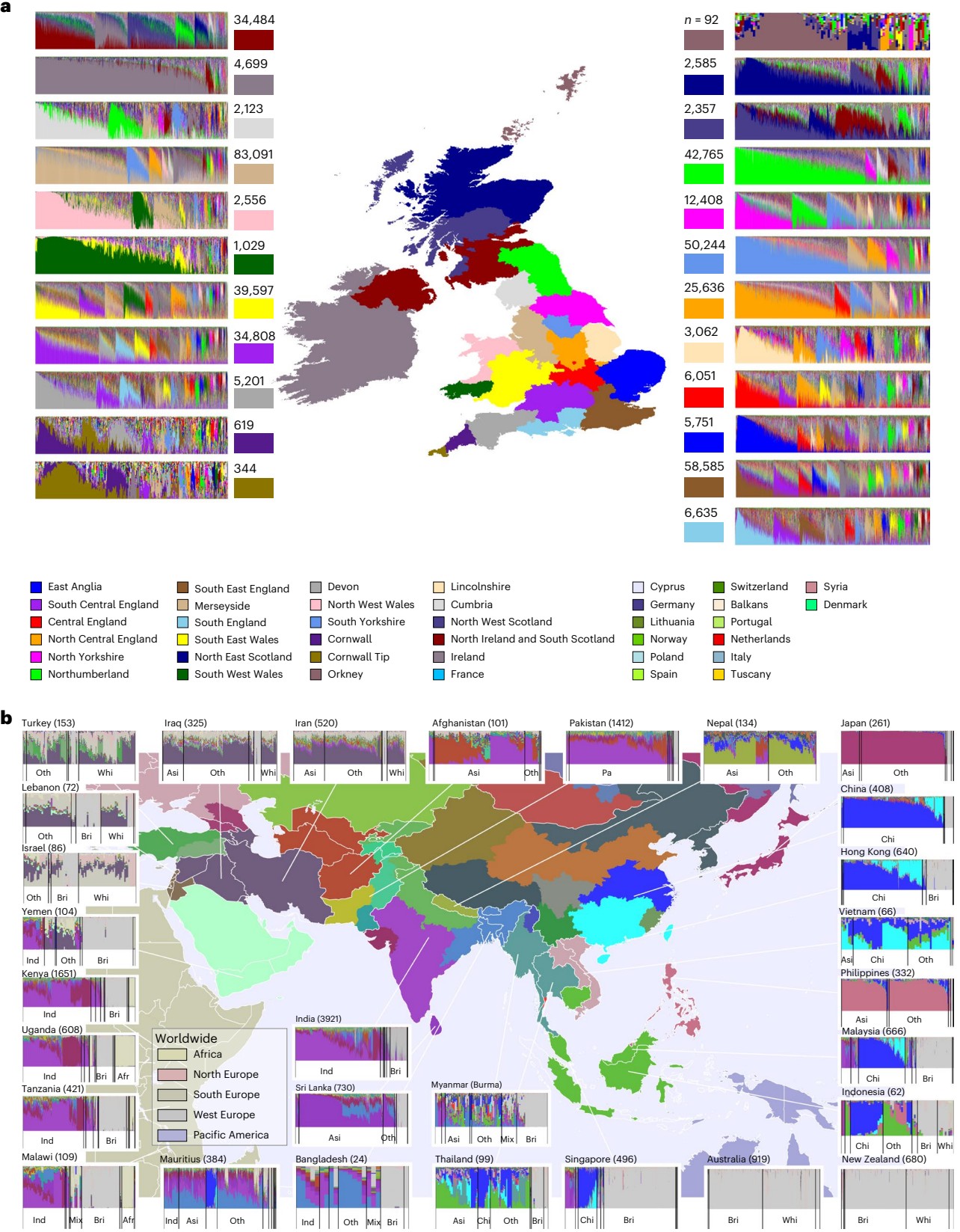

**Fig. 2 | Ancestry inference for UKB individuals born in the United Kingdom or Ireland and worldwide. a**, Ancestry inference stratified by birthplace region for UKB WBI individuals; for each regionally labeled bar plot, each column shows ancestry decomposition for a single individual, with colors representing regions shown on the map and numbers representing counts of individuals from each area. **b**, As **a**, but showing decomposition for Asian, Oceanian and selected East African countries. Colors are as shown on the map, with colors for ancestry from additional regions given in the legend. White lines on the map delineate the borders of different countries. Self-reported ethnicity labels are shown below each bar plot. Color legends differ in **a** and **b**. Self-reported ethnicity: Afr, African; Asi, Asian; Bri, British; Chi, Chinese; Ind, Indian; Ire, Irish; Mix, mixed; Other, other ethnic group; Pa, Pakistani (Asia); Whi, white (Europe).

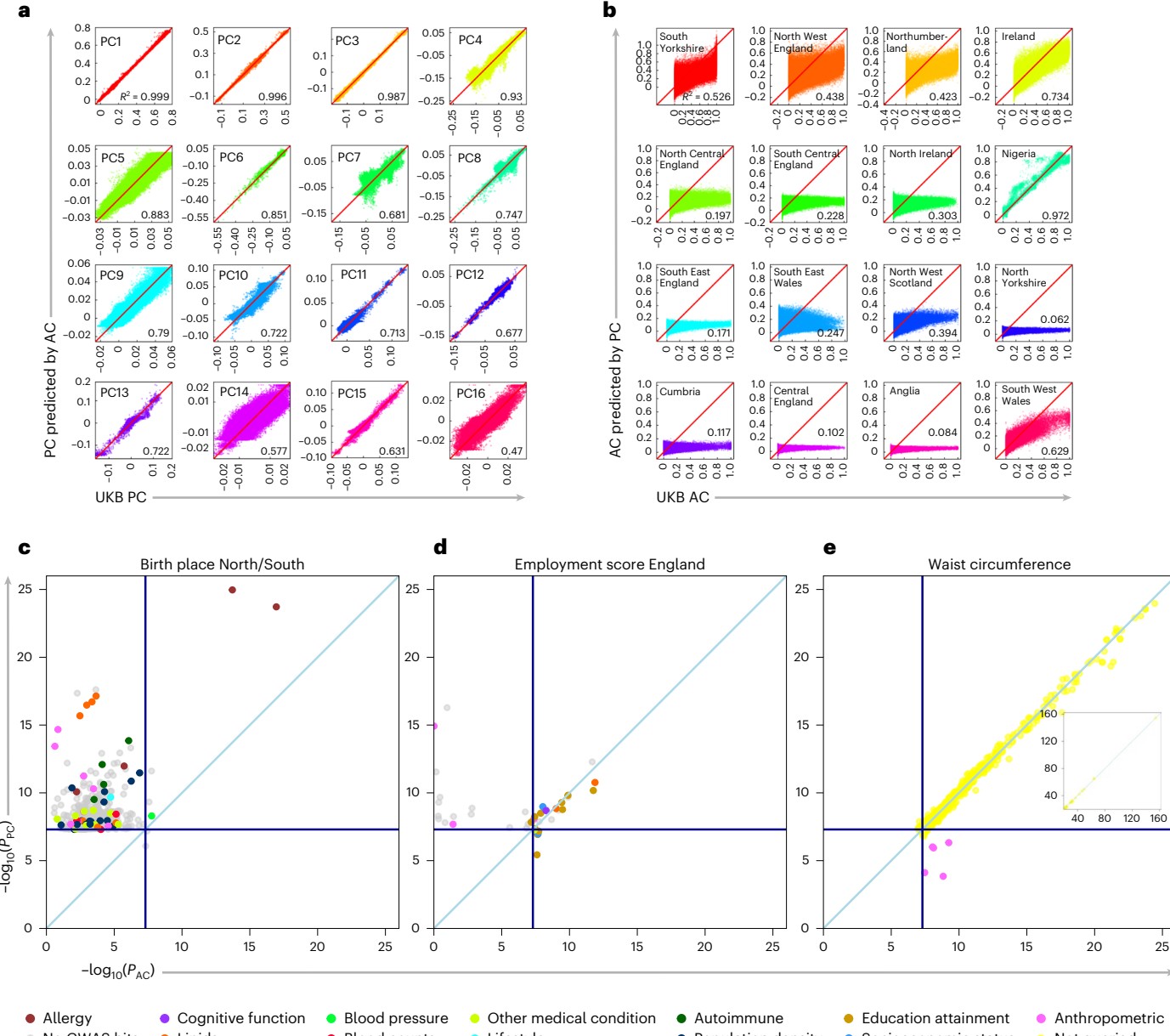

**Fig. 3 | Comparison between AC-corrected and PC-corrected GWAS. a,b,** Predictions are based on 'linear' combinations of ACs or PCs. **a,** Prediction of first 16 UKB PCs (x axes) using a linear model-based prediction from the 127 ACs (y axes) shows strong correlations ($R^2$ values). **b,** As **a**, but now predicting 16 UK and worldwide ACs from 140 PCs, often showing poor prediction. **c–e,** Comparison of AC-corrected (x axis) and PC-corrected (y axis) $-\log_{10}(P \text{ values})$ for SNPs in three exemplar GWAS for labeled traits: birthplace (**c**), employment score (**d**) and waist circumference (**e**). All plots are colored according to the legend shown at the bottom, indicating earlier evidence from GWAS for each SNP in particular phenotypic categories (gray: SNPs show no prior GWAS evidence, perhaps consistent with likely false-positive associations). The horizontal and vertical dark blue lines indicate the genome-wide P-value threshold ($P = 5 \times 10^{-8}$) in a $-\log_{10}$ scale, while the light blue line represents $y = x$. In each plot, the points show only independent SNPs with $P < 5 \times 10^{-8}$ for one or both approaches.

hits), implying that fine-scale ancestry information can effectively remove stratification impacts. Similar results were observed for home location (Extended Data Fig. 4a,b and Supplementary Fig. 3c,d). Although any residual birthplace hits may reflect inadequate adjustment for stratification, the few association signals identified using ACs show a modest enrichment in SNPs showing previous GWAS evidence (Supplementary Table 4; odds ratio (OR) = 5.9, P = 0.039 compared with 12% PC-corrected hits with previous GWAS evidence). Specifically, the strongest remaining signal ($P < 10^{-15}$) after using ACs surrounds rs5743618 on chromosome 4, linked to toll-like-receptor genes involved in innate immunity[40,41], including *TLR1* and *TLR10*, one of the strongest hay fever hits in European GWAS cohorts[42,43], with weaker asthma risk

association. Because the protective allele against hay fever is more common in southern England where hay fever is most prevalent (Extended Data Fig. 5), such geographic differentiation[44] might reflect selective migration of people carrying this variant and/or past natural selection.

We also performed a GWAS for 'employment score England', a trait defined based on the region in which a person lives. GWAS associations with such traits have been observed[6], but such associations can be confounded by regional stratification. Indeed (Fig. 3d), ACs and PCs shared a number of hits, but some signals were only observed in the PC-based analysis, and AC-based showed a stronger enrichment of previous GWAS signals (Supplementary Table 4; OR = 11, P = 0.0013) compared with PCs. In total, 18% of PC-only associations overlapped with previous

GWAS signals, similar to the null trait of latitude (OR = 1.6, $P$ = 0.44), compared with 71% for AC-based associations (OR = 18, $P$ = 1.8 × 10$^{-10}$ versus latitude). Moreover, many AC-based signals were significant in previous GWAS for educational attainment or socioeconomic status (Fig. 3d and Supplementary Table 3c). These trait classes were further enriched relative to PC-only hits (OR = 16, $P$ = 0.005), suggesting PC-only hits are likely false positives (Fig. 3d). These findings suggest that previous GWAS for 'regionally defined' traits can produce false positives, and ACs help correct for this.

Among the other 99 nonregionally defined traits, PC-based and AC-based correction provided similar $P$ values and LDSC intercepts, with subtle improvements for AC (Extended Data Fig. 4a,b). However, differences included five hits unique to the AC-corrected analysis (Methods, Fig. 3e and Supplementary Table 3e), some in strong LD with known GWAS hits for the same traits[43,45–47], and thus unlikely to be false positives. For traits including waist circumference (Fig. 3e and Supplementary Table 3e), AC-specific hits occurred at SNPs in regions of strong LD with high regional SNP loadings for particular PCs (Supplementary Table 5a–e and Extended Data Fig. 6). Therefore, applying ACs likely removed false negatives caused by PC overcorrection from PCs strongly correlated with genotypes in particular genomic regions.

## Causal effects are similar across ancestries

To study PGS portability across UKB groups, we created PGS using 343,047 white British individuals for 53 heritable (estimated heritability >5%) UKB-measured quantitative traits, correcting for ancestry using either ACs or PCs, and tested their performance in independent samples representing different ancestries; that is, with more than 50% inferred ancestry (by ACs) from seven respective labeled regions: South Central England, Northumberland, Republic of Ireland, Poland, India, China and West Africa. AC-based versus PC-based correction yielded different group-specific PGS means, but strong within-group correlations (92.7% to 99.9%) (Supplementary Fig. 4 and Supplementary Table 6) so we focus on AC-corrected PGS for the remaining analyses. By regressing the PGS against actual traits in each group, we quantified the increase in trait per unit PGS increase (that is, the regression slope) and denoted it as $\beta$, and denoted the variance explained by the PGS as $\Delta R^2$, after regressing out covariates (Methods). Both $\Delta R^2$ and $\beta$ decreased with increasing genetic distance from the British ancestry groups (Supplementary Figs. 5 and 6), particularly for sub-Saharan African ancestry, with a >2.2-fold reduction in $\Delta R^2$. Fine-scale ancestry showed this occurs even between BI regional ancestries for some traits: for standing height, forced expiration volume in 1 s (FEV1) and apolipoprotein B, $\Delta R^2$ for Northumberland differs from that for Ireland ($P$ < 0.02, $P$ < 0.005 and $P$ < 0.02, respectively).

To partition the drop-off in PGS performance across ancestries into local effects (causal variant tagging) and nonlocal effects (ancestry-specific gene–gene or gene–environment interactions), we developed ANCHOR, which leverages variation in local ancestry along the genome and between admixed individuals (Fig. 4a). ANCHOR takes PGS coefficients from an independent sample as input, and analyzes quantitative phenotype and genotype data for a group of admixed individuals. It produces an estimate of $\rho$, the mean trait increase in admixed individuals per unit increase in a perfect PGS for nonadmixed (for example, European) individuals, constructed using their (unknown) true effect sizes. $\rho$ equals the correlation in true effect sizes between populations under reasonable assumptions (Supplementary Note 2). For our analysis, we generated PGS coefficients from 343,047 UKB white British samples, and analyzed 8,003 'African ancestry' individuals with varying (mean 83.6%) inferred sub-Saharan African ancestry and BI + Europe (mean 11.5%) ancestry.

ANCHOR estimates local ancestry along the phased genome, for example, using HAPMIX[48] here, masking uncertain and short segments to ensure ancestry segments extend further than LD (Methods and Supplementary Note 2). It calculates separate PGSs for African-like

and European-like segments, yielding 'African PGS' (APGS' and 'European PGS' (EPGS) scores for each individual, each 'mean-centered' by ancestry-dependent mean SNP frequencies, a critical step both in theory (Supplementary Note 2) and in simulations. The phenotype is regressed against APGS and EPGS, including other covariates, to estimate coefficients $\beta_{Af}$ and $\beta_{Eu}$, which quantify ancestry-specific PGS predictive power as the respective average trait increases in admixed individuals (Supplementary Note 2) per unit APGS or EPGS increase. Finally, $\beta_{Eu}$ is then compared with the corresponding $\beta_{Obs.Eu}$ estimate from BI individuals to estimate $\rho$, as $\beta_{Eu}/\beta_{Obs.Eu}$ (Supplementary Note 2). Importantly, the validity of the $\rho$ estimation requires no assumptions regarding underlying effect size distribution, causal mutation frequencies, LD patterns or selection. Under a local effect only ('null') model, causal effect sizes are identical in all European-ancestry regions and $\rho$ = 1. If nonlocal ancestry-specific (gene–gene or gene–environment) interactions occur, $\rho \neq 1$, and if the variance of underlying genetic effect sizes is the same in each group, then $\rho$ < 1 (Supplementary Note 2). Therefore, testing for $\rho \neq 1$ provides a test for differing effect sizes across groups. Assuming effect sizes scale linearly with genome-wide ancestry, ANCHOR can predict $\rho$ for 100% European or African ancestry without sampling (Methods and Supplementary Note 2). Pooling $\rho$ values among traits helps reduce overall uncertainty, and confidence intervals (CIs) for all coefficients are obtained by bootstrapping (Methods).

We verified HAPMIX's[48] ability to accurately infer ancestry and construct ancestry-specific PGS by comparing it with trios with known phase (Extended Data Fig. 7 and Supplementary Fig. 7). We then tested ANCHOR by simulating 24 quantitative traits with various settings of heritabilities, clustering of causal mutations and causal marker frequency spectra, using genetic data from the 8,003 African-ancestry individuals (Methods and Supplementary Note 1). We performed GWAS and downstream analyses exactly as for the real phenotypes. In nonadmixed populations, mean-centering of genotypes has no impact on the PGS predictive power, but is crucial for ANCHOR's validity in admixed populations (Supplementary Note 2). Under the null ($\rho$ = 1), across simulations $\rho$ is correctly estimated from mean-centered EPGS and APGS, but estimates are strongly downward biased without mean-centering (Extended Data Fig. 8 and Supplementary Figs. 8 and 9), even when masking short or uncertain segments (Supplementary Figs. 8a and 9). Therefore, we use mean-centering for all ANCHOR analyses. We also observe significantly reduced $\rho$ estimates for real quantitative traits without mean-centering, fully consistent with the simulation findings (Supplementary Figs. 10–12).

If $\rho$ = 1, 95% bootstrapped CIs for 96% of individual traits contained the true value ($\rho$ = 1), with robustness to different GWAS $P$-value thresholds (0.05 versus 0.0001) (Supplementary Fig. 13). Averaging $\rho$ estimated by ANCHOR across traits for groups of individuals with similar ancestry levels also yielded $\rho$ = 1 (Extended Data Fig. 9), indicating good performance under the null across both traits and ancestry. In simulations in which $\rho$ declines, ANCHOR estimates of $\rho$ remain well calibrated (Fig. 4b, Extended Data Fig. 9 and Supplementary Fig. 14). In all cases we observe $\beta_{Af}/\beta_{Obs.Eu}$ < 1 as expected because of local effects (Supplementary Note 2), so African-ancestry segments are less predictive of traits, but $\beta_{Af}$ values varied similarly to $\beta_{Eu}$ (Extended Data Fig. 9 and Supplementary Fig. 14).

We applied ANCHOR to 53 UKB quantitative phenotypes and found strikingly constant average $\rho = \beta_{Eu}/\beta_{Obs.Eu}$ estimates across genome-wide ancestry bins (Fig. 4c and Methods), overlapping the null value $\rho$ = 1 for all bins. This indicates that European-ancestry segments retain predictive power similar to that in European-ancestry individuals. Thus, a 'true' PGS from Europeans predicts similar trait increase, on average, per unit PGS increase, in African-ancestry individuals. This implies either conserved effect sizes between groups at shared causal SNPs, across the broad range of human molecular and quantitative phenotypes we examined or—less parsimoniously—systematically

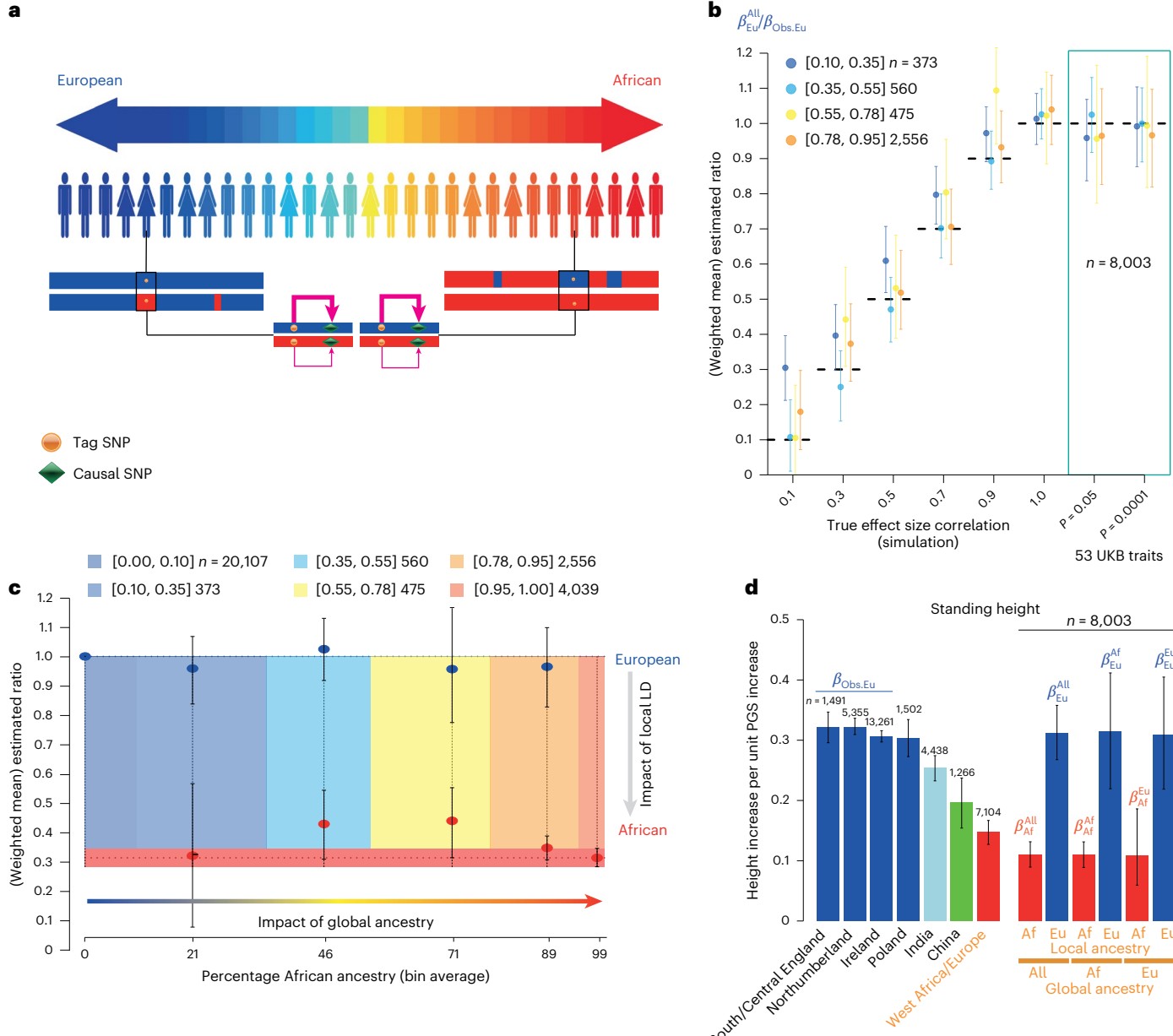

**Fig. 4 | Separation of local and nonlocal factors influencing portability. a**, Test principles: in UKB samples with European (blue) and African (red) ancestries, a causal variant contributing to a trait is captured by a tag SNP whose predictive power (pink arrow thickness) varies by 'local' ancestry (upper versus lower chromosomes), or nonlocal factors captured by genome-wide 'global' ancestry (left versus right individuals); ANCHOR separates these contributions to PGS portability. **b**–**d**, $\beta_j^i$ values refer to the mean increase in phenotype per PGS unit increase for local ancestry $j$ and global ancestry $i$ (see Methods for further details). **b**, ANCHOR performance for 24 simulated traits and 53 UKB quantitative traits with PGSs constructed using different $P$-value thresholds ($P = 0.05$ and $P = 0.0001$; right). True effect size correlations $\rho$ ($x$ axis) between African and European ancestries are compared with the ANCHOR estimator $\beta_{Eu}^{All}/\beta_{Obs.Eu}$ ($y$ axis). Colors denote African ancestry bins, as defined in **c**. **c**, Application of

ANCHOR for 53 UKB traits across varying African ancestry binned as shown ($x$ axis; colored regions). For each bin, mean estimates across traits of ratios $\beta_{Eu}/\beta_{Obs.Eu}$ (blue) and $\beta_{Af}/\beta_{Obs.Eu}$ (red) are shown. Also shown are ratio estimates for individuals of ~100% European (leftmost point at $y = 1$) or ~100% African (red horizontal bar) ancestry. CIs crossing $y = 1$ are consistent with identical effects to ~100% European-ancestry individuals, and similarly for red points or bar. **d**, Mean increase in standing height per PGS unit increase across populations (seven left-hand columns); alongside corresponding ANCHOR estimates for height (final six columns) labeled by global or local ancestry combinations. Data are presented as (weighted) means (**b**,**c**) or as estimated values (**d**) with 95% central bootstrapped CIs. Error bars indicate 95% bootstrapped CIs, Af, African; Eu, European.

larger average effects across all these phenotypes in African-ancestry individuals, in such a manner as to coincidentally balance the impact of incomplete correlation.

For individual traits, as in the simulations we combine all individuals to estimate a trait-specific $\rho$, and extrapolated to estimate effect sizes for European segments in individuals with almost 100% European

ancestry or African ancestry genome-wide (Figs. 4d and 5). For standing height (Fig. 4d), we show $\beta_{Af}$ and $\beta_{Eu}$ estimates alongside $\beta_{Obs.Eu}$, and $\beta$ estimates for various AC-defined UKB cohorts. $\beta$ estimates decline with increasing genetic distance because of LD changes, reducing PGS predictive power. However, ANCHOR shows that European-ancestry segments in African-ancestry genomes have strong predictive power,

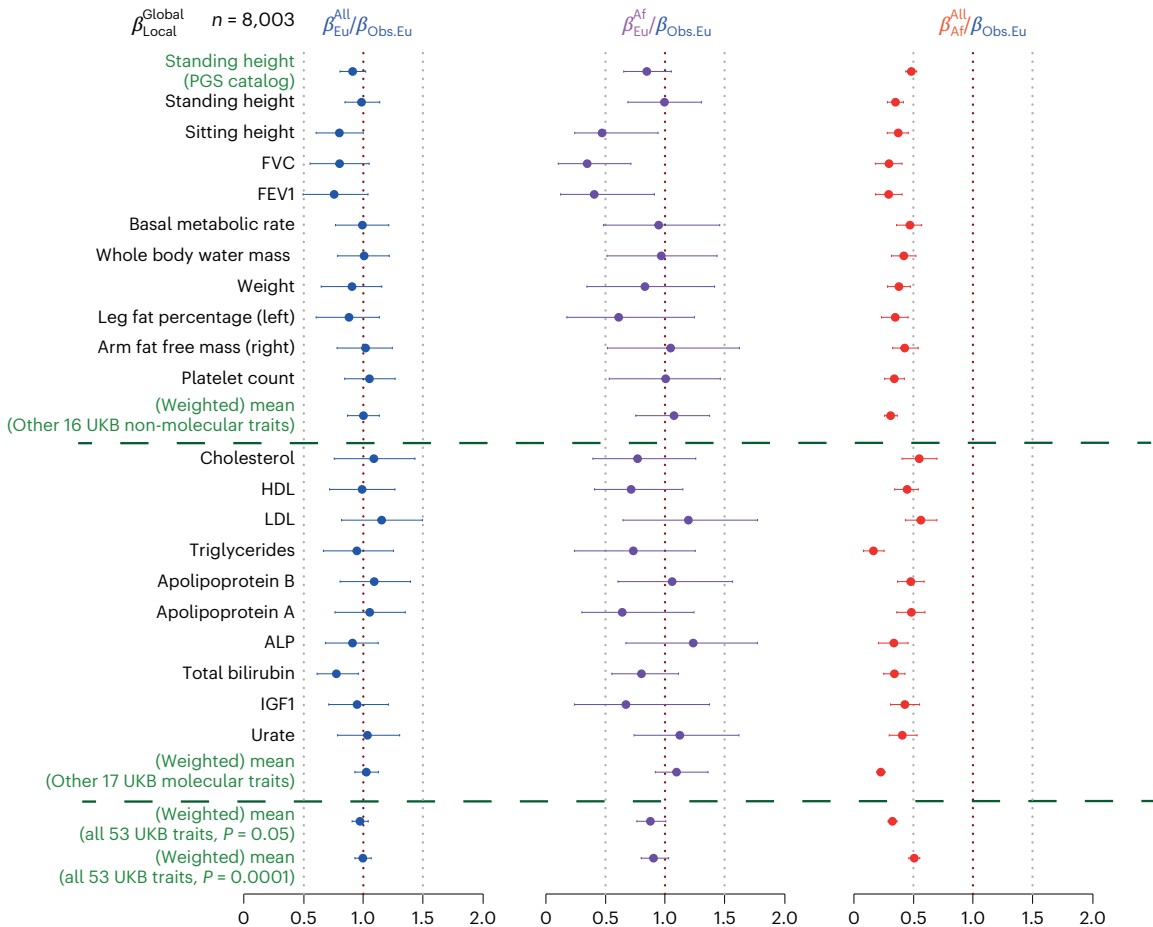

**Fig. 5 | ANCHOR results for 53 UKB traits.** Data are presented as estimated values of ratio of true effect sizes with 95% central bootstrapped CIs. $\beta_j^i$ values refer to mean increase in phenotype per PGS unit increase for local ancestry $j$ and global ancestry $i$ (see Methods for further details). Colors of $\beta_j^i$: blue, European; purple, projected to 100% African ancestry; red, African ancestry. Black rows represent individual UKB traits; the first standing height row uses an existing PGS[16]; the dark green rows show combined estimates. Columns (left to right) estimate $\rho$ for 'all' 8,003 African-ancestry individuals, $\rho$ for individuals of 100% projected African ancestry and (as expected, reduced) predictive power for African-ancestry segments. From top to bottom, the rows above and below the first horizontal dashed line represent non-molecular and molecular traits and rows above and below the second dashed line represent individual traits and their weighted average estimation. Vertical dotted lines: grey lines indicate $\rho = 0.5$ (left of the red dotted lines) and $\rho = 1.5$ (right of the red dotted lines); red lines indicate $\rho = 1$. ALP, alkaline phosphatase; FVC, forced vital capacity; HDL, high-density lipoprotein; IGF1, Insulin-like growth factor-1; LDL, low-density lipoprotein.

similar to wholly European-ancestry individuals (blue bars), and does not identify any significant effect sizes changes across the range of genome-wide ancestry. African-ancestry segments (red) show much lower predictive power, explaining nonportability because of local LD and allele frequency differences, rather than gene–gene or gene–environment interactions.

Across all 53 quantitative and molecular phenotypes (Fig. 5), $\rho$ estimates (first column) were almost indistinguishable from 1, even for individuals with the highest African ancestry (second column), indicating embedded European segments have similar predictive power to nonadmixed Europeans, but with strongly reduced overall PGS power (third column). The joint bootstrap yielded an overall $\rho = 0.98 \pm 0.07$ for these traits, extremely close to 1. Results were consistent using varying $P$-value thresholds ($P$ value of 0.05 and 0.0001) from the initial GWAS (Supplementary Fig. 15), and correction methods (ACs versus PCs) in the initial GWAS (Supplementary Fig. 16). Results for standing height show little change using a previously published, alternative, PGS[16] (Fig. 5). Two lung-function traits: FEV1 and forced vital capacity, and the correlated trait[49,50] of sitting height, showed $\rho < 1$ at nominal significance ($P < 0.05$) (Fig. 5, Supplementary Fig. 15), with white blood cell count, red blood cell count and albumin showing $\rho > 1$ ($P < 0.05$). Only the white blood cell count remained significant ($P < 0.05$) with a GWAS $P$-value threshold of 0.0001, and no traits were significant after Bonferroni correction. In future, larger sample sizes may better detect heterogeneity of effect sizes for individual traits.

## Discussion

We introduce new approaches to understand human ancestry and its connections to GWAS and PGS prediction. Decomposing UKB individuals' ancestry into ACs at a subnational level, improves confounding correction by capturing information missing from PCs, reducing likely false positives and likely false negatives. ACs offer better control for geographic effects than current methods, within the UKB at least. Many traits show similar $P$ values using ACs versus other popular approaches, but ACs uniquely avoid likely false positives in 'regional traits' defined on groups of individuals[6]. Together with observations of likely false negatives in GWAS using PCs, and potential larger sample sizes of future studies, incorporating ACs into GWAS may improve stratification correction by reducing overcorrecting linked to local genomic regions, and avoids the issue of deciding the number of PCs in GWAS. The complex patterns of ancestral mixing within UK regions or between, for example, cities and rural areas, suggest particular care is needed for traits showing similar regional variation such as educational attainment[24]. We note that the achievable granularity and interpretation of ACs depends on the size and diversity of the ancestry reference panel used, as well as the related number of identifiable predefined 'ancestry groups'. A

limitation of our approach is, therefore, that current ACs are somewhat United-Kingdom-specific, but region-specific ACs for non-UK GWAS could offer similar benefit. Until then, ACs can complement PCs, though even the best ACs may not completely correct for subtle population stratification.

To compare causal effect sizes across human populations, we consider the mean effect of a causal mutation on a trait[12,51] and compare this across groups. We note that the common and often convenient scaling of genotypes by their population standard deviations also scales effect sizes differently across populations, so to avoid downward biases in estimating causal effect correlations, we avoid such scaling[14], and instead define identical impacts as a mutation causing the same average increase in phenotype in each group. This biologically natural approach provides an interpretable scale for practical applications.

ANCHOR leverages local ancestry inference to estimate the ratio of underlying effect sizes between an admixed group and a reference group similar to that used for the initial GWAS and PGS construction, by assessing the predictive power of PGS among groups in terms of their relative 'mean trait increase per unit PGS increase'. This ratio is 1 if effect sizes are identical, and estimates the correlation between true effect sizes in the two groups (Supplementary Note 2). Overall our results indicate clearly that—within the UK at least—using effect sizes from 100% European individuals yields near-identical performance in individuals of African ancestry for various quantitative phenotypes. This suggests PGS utility across populations, at least for causal mutations not private to one group. Although an underlying correlation between effect sizes $\rho < 1$ is possible (Supplementary Note 2), if coincidentally counterbalanced by larger average effect sizes in African-ancestry individuals, this seems unlikely across diverse quantitative traits a priori. If instead $\rho \approx 1$, most causal effect sizes are very similar between individuals of African ancestry and European ancestry, across the range of quantitative phenotypes we studied. Of interest for further study, a few traits such as FEV1 do suggest evidence of differences.

Gene–gene and gene–environment interactions likely, therefore, do not drive the lack of PGS portability in the UKB. Population structure is also an unlikely cause, given consistent effect size estimation between PGSs constructed by AC-corrected and PC-corrected GWAS. Instead, local LD and SNP frequency differences appear to be the main factors. African-ancestry segments show significantly reduced predictive power (about a threefold reduction) compared to European ancestry segments (Fig. 5). Our ($\rho = 1$) simulations confirm that PGS coefficients match real data for European segments, but show only a modest reduction (less than twofold for traits with >100 causal markers) for African-ancestry segments. This suggests reduced tagging of causal variants by the PGS in African-ancestry regions potentially because of selection against trait-impacting variants[52,53] causing larger frequency differences for true causal variants between populations than the randomly selected variants in our simulations, and the observed strong drop-off between groups. Stronger stratification correction by ACs could help investigate such selection in future.

Our results differ from previous findings[51,54,55] of often considerably lower correlations between European-ancestry and African-ancestry samples using methods that model genetic variance components and local LD differences. For example, one study[51] finds different effects for body mass index in UKB, another[55] finds different effects for height, and an estimated correlation of only ~50% across 26 traits in UKB. Because ANCHOR focuses on PGS prediction with minimal assumptions, it is challenging to attribute our near-parity estimates to confounding. These differences might result from how genotypes are scaled and centered, as shown by the importance of appropriate centering in ANCHOR (Extended Data Fig. 8 and Supplementary Figs. 8 and 9). Alternatively, if causal variants have unique evolutionary histories because of natural selection, this might impact some methods more than others. Further study is needed to understand these discrepancies.

Our results do not imply that gene–environment, or even gene–gene, interactions are absent across the traits we studied. Such interactions likely cause variation in effect size across individuals with African ancestry but must largely be shared with other ancestries to avoid differences in overall (mean) effect sizes across populations. Previous work[9] has shown effect sizes variation within UKB individuals of British ancestry, based on age, gender and socioeconomic status. We also observe differences between males and females in mean effect sizes (Extended Data Fig. 10), and subtle differences among UK groups stratified by ACs (Supplementary Fig. 6), likely reflecting gene–environment interactions. However, the strong lack of portability in African-ancestry individuals seems not driven by these interactions, apart from specific traits like FEV1. This result encourages joint fine-mapping[56] efforts and suggests that improving causal variant tagging is key to applying genetic findings across groups, simplifying the process by avoiding the need of re-estimating effect sizes.

The UKB resource contains diverse ancestries, but collects homogenous data for individuals within a single country, minimizing trait definition differences and environmental effects to better isolate underlying biological impacts. Our results likely rule out differential gene–gene interactions as a major driver of nonportability, even in other settings, because these would still operate strongly within the United Kingdom. However, effect size differences might be stronger between countries with greater environmental differences or differing trait definitions. In future, ANCHOR might be applied to groups outside the United Kingdom, for example, African Americans[14], to analyze various traits, or extended to analyze binary disease traits. As GWAS sample sizes in admixed and other populations grow, methods including ANCHOR will likely uncover variable effect sizes across countries or cohorts, whereas other approaches[10,52,54] enable comparisons of groups of similar ancestry.

## Online content

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

## Methods

Our research complies with all relevant ethical regulations. Collection of the UKB data was approved by the Research Ethics Committee of the UKB and this research has been conducted using the UK Biobank Resource under application number 27960.

### Statistics and reproducibility

Sample size is clearly disclosed in the paper for each different dataset or sub-dataset. The sample size of ancestry-specific analysis in the section 'Polygenic score calculations' was determined by the corresponding genetic ancestry coefficients inferred using our method. Trio individuals with African-background ancestry were detected using the open source software king (v.2.2). No data were excluded unless the participants withdrew their participation from the UKB. No randomization was used; because there are no experimental groups it is not relevant to our study. No blinding is used because it is not relevant to this study; however, group allocation is objective using inferred genetic ancestry.

### Ancestry pipeline

To construct an ancestry 'painting' reference panel, we merged data from Supplementary Table 7, resulting in 9,129 samples and 2,011,414 distinct mutations. After relatedness pruning in plink1.9 (ref. [57]) using the IBS/IBD computation with the '--genome' option and setting a PI_HAT threshold of 0.25 to exclude related samples exceeding this value, we retained 7,775 individuals. Phasing (using SHAPEIT2)[32] and imputation (using IMPUTE4)[25] were performed using the UK10K[58] + 1000 Genomes data as a reference panel, retaining 851,948 sites with a mean IMPUTE4 information score above 0.9, across all the different genotyping platforms simultaneously, as well imputed sites, in the 'draft painting panel'.

The remainder of the panel construction process, detailed in Supplementary Note 1, involved a nested sample hierarchy, chromosome painting using ChromoPainterv2 (ref. [33]), quality control for relatedness, 'surrogate' and 'donor' group formation based on sample labels and admixture estimation by NNLS[22], and additional SNP quality control, resulting in 677,173 SNPs in the 'final painting panel'. We also painted UKB individuals who were born in world regions that we believed our reference individuals sampled from poorly, after excluding samples with, for example, mainly BI ancestry, and used their profile as a surrogate group reference vector.

Annotating population labels into genetic groupings is an art, not a science. Our choices led to 206 surrogate groups that were genetically distinct, which we sum into the 127 interpretable ancestry labels reported in the 'Results' section. For replicability we include the SNP lists, and the final donor and surrogate group annotations (Supplementary Note 1).

### Painting panel processing

To obtain calibrated local ancestry estimates, individuals in the panel must be exchangeable with those we wish to compare. We first construct a SNP and sample list using the process above, excluding UK10K or 1000 Genomes samples; second, phase each sample independently against the UK10K + 1000 Genomes dataset using SHAPEIT2; third, impute each sample independently against the UK10K + 1000 Genomes dataset using IMPUTE4; and fourth, infer best-fit parameters $N_e$ (which controls recombination rate) and $m$ (which controls mutation rate) using ChromoPainterv2 in expectation-maximization mode for 10% of samples randomly chosen for each chromosome, then average these for final parameters. First, to perform a 'leave-one-out' procedure to create reference groups with one fewer sample, we paint with ChromoPainterv2 using inferred parameters and repeat the 'leave-one-out' procedure above. Second, given each sample a 'donor-vector' indicating genome shared (in centi-Morgans) with each donor group, we create a surrogate panel by averaging donor-vector within each surrogate group. Third, we estimate admixture by treating

each sample's donor-vector as a mixture of the surrogate vectors in the surrogate panel using NNLS as described in ref. [22], merging or removing any groups not >50% recovered.

### UK Biobank data included in the analysis

The UKB study includes more than 500,000 UK residents. We analyzed genotypic and phenotypic data under application 27960. In total, 487,409 UKB participants with available autosomal haplotype and genotype imputation data (field IDs 22438 and 22828) were used in our ancestry inference. Quality control, phasing and imputation for the UKB genetic data have been described previously (http://biobank.ctsu.ox.ac.uk/). Demographic and phenotypic data are listed in Supplementary Table 8. For PGS analysis, we selected 53 quantitative traits on which to run GWAS using the following criteria: trait measured on at least 400,000 individuals; LDSC-estimated trait heritability at least 5%; and the trait must be noncategorical.

### Running the ancestry pipeline on UK Biobank

The ancestry pipeline accepts genotype data in various formats as input (Supplementary Note 1). For UKB data, we used available phased haplotypes. We performed initial imputation with IMPUTE4 in batches of 1,000 UKB individuals, then ran the remaining jobs individually: first, remove close relatives and one random individual per donor group from the donor panel; second, paint the target using parameters $Ne$ and $m$ estimated from the panel to obtain a donor-vector; and third, infer global ancestry using NNLS.

### Assign UK counties with predefined UK regions

Within the United Kingdom and Ireland, 23 distinct groupings were identified. Each UK county was assigned to one group, to refine geographic boundaries. We downloaded county-level UK map data (https://gadm.org), mapped the birthplaces of 426,879 UK-born UKB individuals to a county using the R package 'sp', and filtered individuals with >50% ancestry from one group. We then assigned a county to the group with the most remaining individuals born in the county. 'Irish' ancestry was assigned to the Republic of Ireland. Within Cornwall, because both 'Cornwall' and 'Cornwall Tip' localized to this county, we used the R package 'raster' to define ancestry on finer-scale pixels, and assigned locations whose mean 'Cornwall Tip' ancestry was at least 0.2 greater than their mean 'Cornwall' ancestry to the 'Cornwall Tip' group.

### Visualization of ancestry based on UKB birthplace

We used Gaussian kernel smoothing to generate spatial smoothed plots showing average ancestry. For each pixel ($p$) in the rasterized UK map with birthplace coordinates $p = (x_p, y_p)$, we calculated the mean of a quantity of interest $O$ (for example, AC) for each of the $N = 426,879$ UK-born or Irish-born UKB samples, smoothed by the Gaussian kernel:

$$\bar{O}_p = \sum_{i=1}^{N} \frac{e^{-q((x_p - x_i)^2 + (y_p - y_i)^2)}}{\sum_{i=1}^{N} e^{-q((x_p - x_i)^2 + (y_p - y_i)^2)}} O_i \qquad (1)$$

where $O_i$ is the ancestral object and $(x_i, y_i)$ is the birthplace coordinate for individual $i$. $\bar{O}_p$ is the mean of $O$ at pixel $p = (x_p, y_p)$. We used adaptive bandwidth smoothing for $q$ (Supplementary Note 1).

To quantify ancestry mixing across the United Kingdom, we calculated ancestry entropy for each individual:

$$E = -\sum_{j=1}^{n_r} \frac{a_j}{\sum_{k=1}^{n_r} a_k} \log \frac{a_j}{\sum_{k=1}^{n_r} a_k} \qquad (2)$$

where $n_r = 23$ is the number of predefined BI regions, and $a_j (1 \leq j \leq n_r)$ are the ACs for that individual. Adaptive bandwidth Gaussian kernel smoothing was used to plot the UK-wide entropy profile.

## Genome-wide association study

Forty PCs (field ID 22009) were downloaded from the UK Biobank. The first 20 PCs were used in GWAS analysis. We additionally calculated the first 200 UKB PCs using the R packages 'pcapred.largedata' and 'pcapred' (https://github.com/danjlawson/pcapred.largedata, https://github.com/danjlawson/pcapred), which closely matches UKB PCs (>99% correlation for the first 40 PCs) (Supplementary Fig. 17).

Some 127 ACs from the ancestry pipeline were used to regress each of 140 UKB PCs (chosen to exceed the number of ACs) on all the 487,409 UKB samples and predict each PC. Similarly, 140 UKB PCs were used to predict the 127 ACs on the same UKB samples. $R^2$ between the true and predicted PC or AC was then used to evaluate the prediction. In our GWAS study, we analyzed 104 continuous UKB phenotypes with a sample size >10,000 (Supplementary Table 8). In total, 343,047 unrelated white British individuals[25] were included in our GWAS. Association testing was performed for UKB imputed SNPs $G$ (minor allele frequency (MAF) >0.001 and information scores >0.3). Covariates in each association test were 'genotype measurement batch', 'age at recruitment' and 'sex' (field IDs 22000, 21022, and 31), as well as nonlinear terms 'age$^2$', 'age × sex' and 'age$^2$ × sex'. The full model using ACs to correct stratification is:

$$\text{Phenotype} \sim G + \text{age} + \text{sex} + \text{batch} + \text{age}^2$$
$$+ \text{age} \times \text{sex} + \text{age}^2 \times \text{sex} + 127\text{ACs} + \text{error} \tag{3}$$

The model instead using 20 PCs (and similarly when using 100 PCs) is:

$$\text{Phenotype} \sim G + \text{age} + \text{sex} + \text{batch} + \text{age}^2$$
$$+ \text{age} \times \text{sex} + \text{age}^2 \times \text{sex} + 20\text{PCs} + \text{error} \tag{4}$$

Association testing was performed by 'BGENIE' v.1.2 (ref. [25]) using unnormalized phenotypes, to keep estimated effect sizes remain in units of the original phenotypes.

We also performed GWAS using BOLT-LMM[34] with 20 PCs, to test performance of a linear mixed model for the place of birth, north coordinates (field ID 129) phenotype. We performed quality control and LD pruning on the UKB chip genotype SNPs using plink1.9 (ref. [57]) (www.cog-genomics.org/plink/1.9/; Supplementary Note 1) to generate 142,182 independent SNPs ($r^2 < 0.1$) for null model building in BOLT-LMM. The participants and imputed SNPs for association testing remained as for the other GWAS. The 20 PCs included in BOLT-LMM as covariates were calculated by FlashPCA2 (ref. [59]) from the same genetic relationship matrix constructed by the 142,182 independent SNPs and 343,047 UKB white British individuals. Other covariates used in BOLT-LMM were the same as those used in the linear regression based GWAS.

To identify independent genome-wide significant signals, we used a threshold $P < 5 \times 10^{-8}$ and for each SNP more significant than this threshold for either AC or PC, we used LD pruning with a threshold of $r^2 < 0.01$ and window size of 100 kb to keep only the most significant SNP among groups of variants in LD.

We queried AC-significant or PC-significant SNPs in a GWAS database using the R package 'mrcieu/ieugwasr' (https://mrcieu.github.io/ieugwasr/). For most traits we applied a genome-wide threshold of $5 \times 10^{-8}$ for query output $P$ value, relaxed to $1 \times 10^{-6}$ if no SNP met the initial threshold, and returning 'No GWAS hits' if the relaxed threshold was also not met. For traits shown in Fig. 3c–e, queried results containing the same or directly related traits were excluded from categorization. Query results for input SNPs were ranked by ascending GWAS $P$ value across different traits such that the most significant trait was top-ranked (Supplementary Table 3a–e). SNPs without a standard RS catalogue identifier were categorized as 'Not queried'. Queried traits were assigned into the following categories: 'Anthropometric', 'Cognitive function', 'Lipids', 'Education attainment', 'Autoimmune', 'Population density', 'Allergy', 'Socioeconomics status', 'Blood pressure', 'Blood counts' and 'Other medical condition' (Supplementary Table 9).

For 'Waist circumference', we identified 627 independent genome-wide significant hits for either AC- or PC-corrected GWAS. Only five hits were AC-specific with a $-\log_{10} \frac{P_{AC}}{P_{PC}}$ of >1, where $P_{AC}$ and $P_{PC}$ are $P$ values for AC- and PC-based GWAS for 'Waist circumference'. We queried four of those five variants (Fig. 3e), using 'mrcieu/ieugwasr', while the remaining variant (15:84311431:TA:T) lacked an RS catalogue identifier, so was manually queried using opentargets[60] (http://genetics.opentargets.org/). It was strongly associated with many different 'Anthropometric' related traits such as height, trunk fat mass and body fat percentage, and so was labeled as 'Anthropometric'. The other 622 variants possessed shared signals and were therefore not queried.

In assigning categories to SNPs overlapping multiple association types, we ranked as follows to prioritize relevance to the traits studied in a particular GWAS in Fig. 3c–e. For 'Birthplace (North/South)', we assigned the category 'Allergy' to the queried SNP if it is associated with allergic phenotypes such as 'Hay fever'; for 'Employment Score England', we assigned category 'Educational attainment' to the queried SNP if it is associated with 'Educational attainment' or related phenotypes, including 'Age completed full-time education', and then category 'Cognitive function' if associated with such traits as, for example, 'Mood swing'; for 'Waist circumference', we assigned category 'Anthropometric' to queried SNPs associated with, for example, 'Leg fat percentage'. Remaining uncategorized SNPs were categorized by their top-ranked associated trait (Supplementary Table 3a–e).

## Heritability estimation and GWAS inflation assessment

We ran LD-score regression[37] on both AC-corrected and PC-corrected GWAS summary statistics for 99 UKB traits, using precalculated LD scores from the 1000 Genomes phase 3 Utah residents with Northern and Western European ancestry (CEU) panel provided alongside the LDSC software. From LD-score regression output of the intercept and $\chi^2$ values, we bounded these at below 1 to ensure non-negativity, and calculated a trait-wise LDSC attenuation ratio, defined as follows[38]:

$$\text{Attenuation ratio} = \frac{\text{Intercept} - 1}{\chi^2 - 1} \tag{5}$$

## Estimation of allele frequencies for UKB SNPs

Highly differentiated alleles across different regions reflect strong local genetic drift and can reveal natural selection. We derived an expectation-maximization algorithm to estimate maximum-likelihood regional allele frequencies based on our individual ACs (Supplementary Note 1). We applied this to 12,977,776 imputed UKB SNPs with minimum MAF at least 0.01% and imputation information score above 0.9 in 339,304 white British samples with more than 95% UK + Ireland ancestry. Using 'qctool2' (https://www.well.ox.ac.uk/~gav/qctool_v2/index.html), 'bgenix'[61] and plink2.0 (ref. [57]), we conducted genotype data preprocessing, quality control and estimated their allele frequencies across the 23 British–Irish regions.

## Polygenic score calculations

For each of 53 traits, we constructed two PGSs using either ACs or PCs for population stratification correction in a GWAS of 343,047 unrelated white British individuals. PGS construction used the Hapmap3 SNPs[62] with a minimum 1% MAF. For each phenotype, we performed LD-clumping by 'plink1.9'[57] using 1000 Genomes phase 3 CEU[63] as the reference panel with following parameters: significant threshold for index SNPs is 0.05, secondary significance threshold for clumped SNPs is 1, LD threshold for clumping is 0.1 and physical distance threshold for clumping is 500 kb, with respect to the plink command line as follows: '--clump-p1 0.05 --clump-p2 1 --clump-r1 0.1 --clump-kb 500'. We also tested a stricter $P$ value threshold of 0.0001 by setting '--clump-p1 0.0001' in the above plink command. The PGS was evaluated on the remaining 144,362 UKB individuals.

PGS was calculated as a linear sum over SNPs and genotypes:

$$PGS_{AC} = \sum_{j=1}^{n} \widehat{\gamma_{ji}^{AC}} G_j \tag{6}$$

$$PGS_{PC} = \sum_{j=1}^{n} \widehat{\gamma_{ji}^{PC}} G_j \tag{7}$$

where for SNP $j$, $G_j$ is the genotype for the target individual, $\widehat{\gamma_{ji}^{AC}}$ is the estimated effect size from the AC-corrected GWAS and $\widehat{\gamma_{ji}^{PC}}$ is the effect size from the PC-corrected GWAS. Mean-centering genotypes within each population would yield equivalent results, so it was not done here (Supplementary Note 2).

To evaluate $PGS_{AC}$ performance across different ancestries, we identified those individuals among the 144,362 UKB 'test' individuals with at least 50% inferred ancestry from each of seven separate AC-labeled groups: 'South Central England' (1,491 individuals), 'Northumberland' (5,355), 'Republic of Ireland' (13,265), 'Poland' (1,503), 'India' (4,438), 'China' (1,266) and 'West Africa' (7,108).

For each group and phenotype, we evaluated $PGS_{AC}$ for individual $i$, label as $PGS_i$, and fit the model:

$$Y_i = I + \beta PGS_i + \text{covariates} + \varepsilon_i \tag{8}$$

where $Y_i$ is the phenotype for individual $i$, $\varepsilon_i$ is the error, and the model was fit via standard linear regression, with 1,000 bootstrapped sample replicates for evaluating uncertainty. As previously, the covariates included are batch, age, sex, age$^2$, age × sex and age$^2$ × sex, as well as the 127 estimated ACs. The parameter $\beta$ represents an estimator of the increase in the mean phenotype per unit increase in the PGS. We compared $\beta$ values among groups to quantify PGS applicability in distinct groups, and used generalized values in our ANCHOR analysis.

For each group, we defined the 'residual $r^2$', $\Delta R^2$, a scale-independent measure of PGS performance, as 1 minus the ratio of the phenotypic variance remaining after fitting the above model to the (larger) variance when $\beta = 0$. This represents the fraction of variance explained by the PGS after accounting for confounding and covariates.

We also estimated an overall value $\beta_{Obs.Eu}$ for 19,596 individuals of BI ancestry by fitting the same model. The BI individuals were selected from three BI groups—South Central England, Northumberland and Republic of Ireland—filtering out individuals with the sum of these three ancestries less than 0.9.

## Ancestry-aware PGS and correlation in effect sizes

To understand the behavior of the PGS in African-ancestry individuals, we identified 8,003 UKB European–African admixed individuals whose sum of 27 UK or European ancestries and 4 sub-Saharan African ancestries is larger than 90% and sum of 4 sub-Saharan ancestries larger than 10%. We further binned these individuals into five ancestry bins based on their inferred African ancestry: [0.1, 0.35], 373 individuals, mean ancestry 0.207; [0.35, 0.55], 560 individuals, mean ancestry 0.46; [0.55, 0.78], 475 individuals, mean ancestry 0.707; [0.78, 0.95], 2,556 individuals, mean ancestry 0.89; and [0.95, 1], 4,039 individuals, mean ancestry 0.986. Individuals in the bin [0.95, 1] with near 100% African ancestry were only given an 'APGS' PGS in our analysis (see below), the results of which are shown in Fig. 4b,c and Extended Data Fig. 9 and Supplementary Figs. 9 and 12.

In the ANCHOR approach, we generate and analyze ancestry-specific PGS for a trait. We applied HAPMIX[48] to estimate local ancestry genome-wide for each individual, using 1000 Genomes phase 3 (ref. 63) CEU and Yoruba in Ibadan, Nigeria groups, respectively, as European and African ancestry reference panels, with default parameters. For individuals labeled $i = 1, 2, \ldots n$, the output of HAPMIX at each locus $j = 1, 2, \ldots J$ provides probabilities $P_{ij}^{EE}, P_{ij}^{EA}, P_{ij}^{AE}, P_{ij}^{AA}$ that the local ancestry

is each of the four possibilities, ordering the two chromosomes arbitrarily (for example, 'EE' refers to both chromosomes possessing European ancestry). Given genotype $G_{ij}$ at site $j$ for individual $i$, we estimate allele frequencies $f_j^E, f_j^A$ for European and African background, respectively, by fitting the model

$$G_{ij} = f_j^E \left(2P_{ij}^{EE} + P_{ij}^{EA} + P_{ij}^{AE}\right) + f_j^A \left(2P_{ij}^{AA} + P_{ij}^{EA} + P_{ij}^{AE}\right) + \varepsilon_i \tag{9}$$

where $\varepsilon_i$ is the mean-zero noise, by least squares. In practice we fit the equivalent model, noting $P_{ij}^{EA} = P_{ij}^{AE}$ and that the ancestry probabilities sum to 1:

$$G_{ij} = I_j + S_j \times \left(2P_{ij}^{EE} + 2P_{ij}^{EA}\right) + \varepsilon_i \tag{10}$$

where after model fitting we obtain estimates $f_j^E = S_j + I_j/2$ and $f_j^A = I_j/2$. Given the large sample size, these estimates closely match true SNP frequencies for the specific admixing groups (they also correlate strongly with the frequencies of the same variants in relevant 1000 Genomes cohorts) (Supplementary Fig. 18).

HAPMIX also estimates expected allele counts $G_{ij}^E, G_{ij}^A$ at this site for European and African ancestry backgrounds (obtained via summation; Supplementary Note 1), which are transformed to their mean-centered version (Supplementary Note 2), conditional on local ancestry probabilities:

$$\bar{G}_{ij}^E = G_{ij}^E - f_j^E \left(2P_{ij}^{EE} + 2P_{ij}^{EA}\right) \tag{11}$$

$$\bar{G}_{ij}^A = G_{ij}^A - f_j^A \left(2P_{ij}^{AA} + 2P_{ij}^{EA}\right) \tag{12}$$

We apply genomic masking to remove uncertain or short segments (<5 megabases) inferred by HAPMIX. However, simulations show consistent results regardless of masking, provided mean-centering is correctly conducted.

The overall PGS can be decomposed into the European PGS (EPGS) and African PGS (APGS) for African-ancestry individuals $i = 1, 2, \ldots n$ as follows:

$$EPGS_i = \sum_{j=1}^{J} \hat{\gamma}_j \bar{G}_{ij}^E \tag{13}$$

$$APGS_i = \sum_{j=1}^{J} \hat{\gamma}_j \bar{G}_{ij}^A \tag{14}$$

where $\hat{\gamma}_j$ is the estimated per-copy effect size of SNP $j$ on the phenotype. To investigate local (for example, LD) and nonlocal factors (that is, interactions) attenuating the PGS performance in African-ancestry individuals, we fitted the following model to real and simulated data (Supplementary Note 2):

$$Y_i = I + (\beta_{Eu}EPGS_i + \beta_{Af}APGS_i) \times (1 + \omega\theta_i) + \text{covariates} + \varepsilon_i \tag{15}$$

where for individuals $i = 1, 2, \ldots n$, $Y_i$ is the phenotype, $\varepsilon_i$ is the zero-mean noise and the parameters to be estimated are the intercept $I$, and $\beta_{Eu}, \beta_{Af}, \omega$, $EPGS_i$ and $PPGS_i$ are the centralized EPGS and APGS after regressing out the covariates, and $\theta_i$ is the mean genome-wide European ancestry proportion for individual $i$. Covariates are as described in the section 'Polygenic score calculations'.

We fit this model across various individual subsets, combinations of phenotypes and parameter constraints (described below), obtaining parameter estimates via least squares (Supplementary Note 2), and uncertainty estimates by 1,000 bootstrap resamples of individuals and model refitting. The model is linear so trivial to fit unless allowing $\omega \neq 0$;

in this case, conditional on the ratio $\beta_{Eu}/(\beta_{Eu} + \beta_{Af})$ the model is again linear, so we use a grid search (1,000 values) over this ratio from 0 to 1, fitting the linear model for each possible value and then minimizing the achieved sum of squares.

Parameters $\beta_{Eu}$ and $\beta_{Af}$ measure the increase in the phenotype per unit increase in the respective PGS, indicating the scores' predictive power. Local factors mean that we expect $\beta_{Eu} > \beta_{Af}$ (Supplementary Note 2; Extended Data Fig. 9 shows this via simulation). If no additional nonlocal factors contribute to nonportability, then $\beta_{Eu}$ and $\beta_{Af}$ should remain constant across ancestry bins, that is varying $\theta_i$, with $\beta_{Eu}$ also shared between African-ancestry individuals and 100% European ancestry individuals (Supplementary Note 2). With nonlocal factors acting, this no longer holds. To investigate this setting, we allow the predictive power of the two scores varies with genome-wide European or African ancestry that is $\theta_i$, captured by $\omega$. Because local factors still operate, the model covaries the predictive power of the $\beta_{Eu}$, $\beta_{Af}$ parameters together with $\omega$; we lack power with current sample sizes to fit separate effects, and so we simply use $\omega$ to capture linear effects[14] (Supplementary Note 2).

We fit this model to analyze both real and simulated datasets in an identical manner. When jointly analyzing phenotypes and averaging results, we first binned the African-ancestry individuals by their genome-wide ancestry $\theta_i$ (bin boundaries shown on Fig. 4c). Within each bin ancestry varies little, so we set $\omega = 0$ and fit the model independently for each bin for each phenotype. This provides estimates of $\beta_{Eu}$, $\beta_{Af}$ for each phenotype-bin combination, averaged within bins for Fig. 4c, allowing comparisons across bins, and with the effect size $\widehat{\beta_{Obs.Eu}}$ obtained for individuals of mainly European ancestry. By holding $\omega = 0$ fixed but analyzing all individuals jointly for a phenotype produces estimates $\widehat{\beta_{Eu}^{All}}$ and $\widehat{\beta_{Af}^{All}}$ summarizing the predictive power of the PGS across the full African-ancestry sample set, shown in, for example, Figs. 4d and 5. The ratio of the means of the estimates over phenotypes, $\widehat{\beta_{Eu}^{All}}/\widehat{\beta_{Obs.Eu}}$, estimates the correlation $\rho$ in effect sizes between Africa-ancestry and European-ancestry groups, averaged over phenotypes (Supplementary Note 2); estimates of this are shown in, for example, Figs. 4b and 5 for simulated and real data. For an individual phenotype, if the 95% bootstrapped confidence interval of the ratio $\widehat{\beta_{Eu}^{All}}/\widehat{\beta_{Obs.Eu}}$ contains 1, we accept the null hypothesis of shared effect sizes. The ratio $\widehat{\beta_{Af}^{All}}/\widehat{\beta_{Obs.Eu}}$ captures local (for example, LD) effects on prediction, and so as expected is <1 for all simulated and real traits. Finally, we fit the full model allowing $\omega$ to vary. In individuals with 100% African [respectively European] ancestry, this fits the unit increase in the phenotype per unit increase in EPGS as $\widehat{\beta_{Eu}^{Af}} = \widehat{\beta_{Eu}}(1 + \hat{\omega})$; $[\widehat{\beta_{Eu}^{Eu}} = \widehat{\beta_{Eu}}]$, and analogous coefficients $\widehat{\beta_{Af}^{Af}}$, $\widehat{\beta_{Af}^{Af}}$ correspond to the APGS. These coefficients are shown in, for example, Figs. 4d and 5. Ratios of these quantities to $\widehat{\beta_{Obs.Eu}}$, and their bootstrapped CIs, are interpreted exactly as those for 'All' individuals but now projected to estimate properties of individuals—for example, ratios of true effect sizes—whose ancestry is 100% African or even 100% European. These quantities are defined consistently across analyses and subsets of individuals.

## Ancestry-aware simulation

To explore GWAS portability and evaluate performance of ANCHOR, we simulated phenotypes across the entire UKB cohort, and analyzed as for the real data. The UKB imputed genotype data includes 12,690,793 variants after applying the following filters: MAF ≥ 0.001, minor allele count ≥25, genotype missingness ≤5%, Hardy–Weinberg equilibrium $P \geq 1 \times 10^{-10}$ and imputation INFO score ≥0.8. From these, we chose a set of $J$ causal variants ($J = 100$, 1,000 or 10,000) either at randomly selected genomic positions, or with clustering. For the clustered setting we selected $J/10$ nonoverlapping 10 KB regions, each containing an average of five causal variants. The number of variants placed in each region is drawn from a multinomial distribution with parameters $J/2$ (number of clustered variants) and $p_k$, $k = 1, 2, \ldots C/10$ where $p_1 = p_2 = \ldots = p_k$. The remaining 50% of causal variants were uniformly

distributed along the genome. Effects sizes of the causal variants are drawn independently either from a $N(0,1)$ distribution, or following a LDAK model[64] where the effect size of variant $j$ is a draw from a standard normal multiplied by a factor $[2p_j(1 - p_j)]^{-0.25}$, where $p_j$ is that variant's frequency, resulting in larger effects for rarer variants. This generates $3 \times 2 \times 2 = 12$ scenarios; we simulated two different heritabilities 0.3, 0.6 resulting in 24 simulations in total.

Finally, to generate simulated phenotypes $Y_i$ for individual $i = 1$, 2, … N, we apply the following additive model which leverages the actual UKB genotypes and so matches properties of these data, including in particular population stratification:

$$Y_i = \sum_{j=1}^{J} \gamma_j g_{ij} + \varepsilon_i \tag{16}$$

Here $\gamma_j$ and $g_{ij}$ are the effect size and genotype of individual $i$ at site $j = 1$, 2, … J. Noise terms $\varepsilon_i$ are drawn from a normal distribution with mean 0 and variance $\sigma_e^2$. To obtain heritability $h^2$ of 0.3 or 0.6, we set the variance of $\varepsilon_i$ in equation (16), $\sigma_e^2$ is equal to $\frac{1-h^2}{h^2}\sigma_g^2$, where $\sigma_g^2$ is the observed variance of the first term.

We also repeated these simulations, now allowing effect sizes to differ in the 8,003 African-ancestry individuals (Supplementary Note 2). For a correlation in effect sizes of $\rho \in \{0.1, 0.3, 0.5, 0.7, 0.9\}$, we simulated from the following model:

$$\begin{pmatrix} \gamma_j^E \\ \gamma_j^A \end{pmatrix} \sim N\left( \begin{pmatrix} 0 \\ 0 \end{pmatrix}, \begin{pmatrix} \sigma_j^2 & \rho\sigma_j^2 \\ \rho\sigma_j^2 & \sigma_j^2 \end{pmatrix} \right) \tag{17}$$

This can be done efficiently without modifying effect sizes for non African-ancestry individuals, by noting that conditional on the European effect size $\gamma_j^E$ at site $j = 1, 2, \ldots J$, $\gamma_j^A$ has the following distribution:

$$\gamma_j^A | \gamma_j^E \sim N\left(\rho\gamma_j^E, (1 - \rho^2)\sigma_j^2\right) \tag{18}$$

or equivalently,

$$\gamma_j^A = \rho\gamma_j^E + \sigma_j\sqrt{1 - \rho^2}Z \tag{19}$$

where $Z$ is a standard normal random variable. We use this to generate $\gamma_j^A$ for each setting and calculate PGS for African-ancestry individuals. We adjust the phenotypic variance for African-ancestry individuals so as to maintain their heritability at the same value as the UKB samples as a whole. Based on the above scenarios, a simulated phenotype named as, for example, '#causal:1K(uniform)S:0 h2:0.6' means the underlying simulation scenario for this phenotype is: a phenotype with heritability 0.6, in total 1,000 causal variants, uniformly distributed along the genome, and S:0 (versus S:0.5) means there is no effect size scaling used.

## Reporting summary

Further information on research design is available in the Nature Portfolio Reporting Summary linked to this article.

## Data availability

UK and world map data can be accessed through https://gadm.org. UK Biobank data can be downloaded by approved researchers through https://www.ukbiobank.ac.uk. Phased haplotype data from 1000 Genomes used as reference panel for HAPMIX can be accessed through https://www.internationalgenome.org/category/data-access/. POBI was accessed using accession no. EGAS00001000672. GWAS summary level data used in this paper can be queried using the interface implemented by 'mrcieu/ieugwasr': https://gwas.mrcieu.ac.uk and through Open Target at https://www.opentargets.org. HapMap3 variants list can be accessed at https://ftp.ncbi.nlm.nih.gov/hapmap/. All

genetic data used in constructing the ancestry pipeline is provided by third parties and is available for use by others. Variant Frequency information for every SNP in each genetic grouping is available at the University of Bristol data repository, data.bris, at https://doi.org/10.5523/bris.3g5oatl682kz82as80jakjrq91. All other resources were downloaded from their respective websites without registration requirements. The following files have been returned to UK Biobank so that they might be made available to other researchers: (1) ACs on all UK Biobank participants; (2) group-specific allele frequency estimates for 25 M variants; (3) (Mean-centered) European/African genotypes and local ancestry of 8,003 UK Biobank African ancestry individuals (including the variants annotation information); (4) European/African PGS for 8,003 African ancestry individuals across 53 UK Biobank phenotypes. GWAS summary statistics files with population structure corrected by ACs and PCs are available in GWAS catalog (https://www.ebi.ac.uk/gwas/, GCST90310137–GCST90310200 and GCST90429571–GCST90429610).

## Code availability

The ANCHOR software package is available via GitHub at https://github.com/MyersGroup/ANCHOR and Zenodo at https://doi.org/10.5281/zenodo.13847648 (ref. 65). Analysis scripts can be downloaded via GitHub at http://github.com/fuopen/UKB_anc and Zenodo at https://doi.org/10.5281/zenodo.14026928 (ref. 66). External software/packages used in this study are available via GitHub at https://github.com/danjlawson/pcapred.largedata ('pcapred.largedata') and https://github.com/danjlawson/pcapred ('pcapred').

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

## Acknowledgements

This research has been conducted using the UK Biobank Resource under application number 27960. We thank all the participants of the UK Biobank project. POPRES was accessed under application number 16508: 'Admixture and Selection for Fine-Scale Population Genetics of Europe'. This work was supported by the Wellcome Trust grant no. 200186/Z/15/Z awarded to J.M., S.R.M. and G.H., grant no. 224575/Z/21/Z awarded to G.H. and grant no. 212284/Z/18/Z awarded to S.R.M. L.A.F.F. is supported by the Wellcome Trust grant no. 222334/Z/21/Z. Computations used the high-performance computing facilities at the Department of Statistics, University of Oxford and Oxford Biomedical Research Computing facility, a joint development between the Wellcome Centre for Human Genetics and the Big Data Institute supported by Health Data Research UK and the NIHR Oxford Biomedical Research Centre. This work was carried out using the computational facilities of the Advanced Computing Research Centre, University of Bristol—http://www.bris.ac.uk/acrc.

## Author contributions

S.R.M., D.J.L., J.M. and G.H. designed the study. S.H., G.H., J.M., D.J.L. and S.R.M. developed the methods. S.H., G.H., J.M., D.J.L. and S.R.M. performed the main analyses with contributions for specific analyses from L.A.F.F. and S.S. S.H., G.H., J.M., D.J.L. and S.R.M. wrote the manuscript.

## Competing interests

S.H. became a full-time employee of Novo Nordisk Ltd during the drafting of this manuscript. J.M. is a current employee and stockholder of Regeneron Pharmaceuticals. The other authors declare no competing interests.

## Additional information

**Extended data** is available for this paper at https://doi.org/10.1038/s41588-024-02035-8.

**Correspondence and requests for materials** should be addressed to Sile Hu, Daniel J. Lawson or Simon R. Myers.

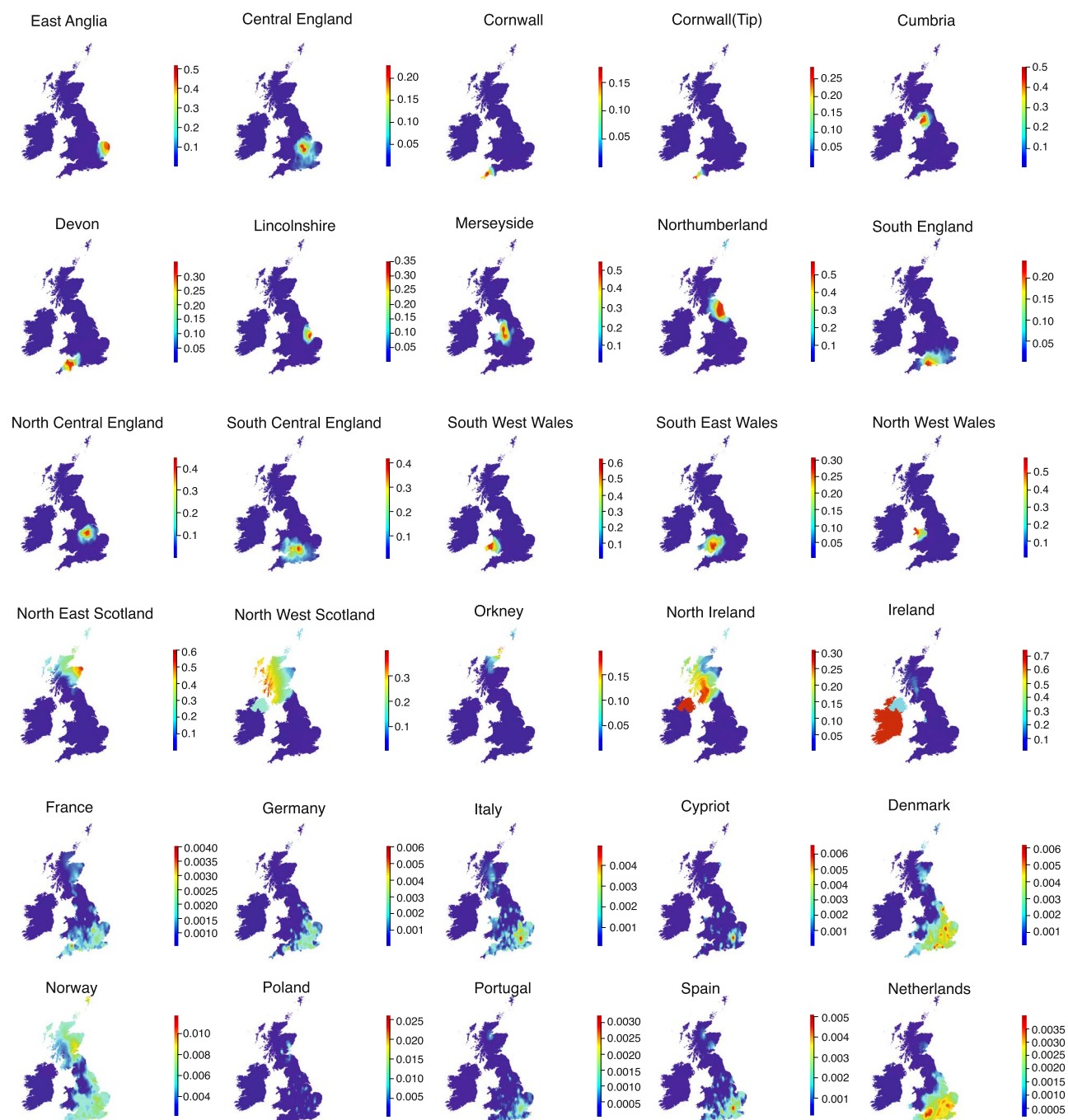

**Extended Data Fig. 1 | Heatmap of regional mean ancestry proportions of 30 UK+EU ancestries based on birthplace among UK-born and Ireland-born UKB participants.** One panel per ancestry, brightest red colour on maps indicates highest ancestry, darkest blue lowest ancestry.

**Extended Data Fig. 2 | World ancestries in the UKB inferred by the ancestry pipeline.** UKB participants are mapped to their self-reported country of birth. World countries are coloured by different colours if they are present in the pipeline. Ethnic groups are abbreviated as follows in the bar plots: Bri = British,

Whi = White, Ire = Irish, Chi = Chinese, Asi = Asian, Pa = Pakistani, Ind = Indian, Mix = Mixed, Car = Caribbean, and Oth = Other Ethnic Group. **a**). Ancestries in Central Asia; **b**). Ancestries in Europe; **c**). Ancestries in Africa; **d**). Ancestries in the Americas. White lines on the map delineate the borders of different countries.

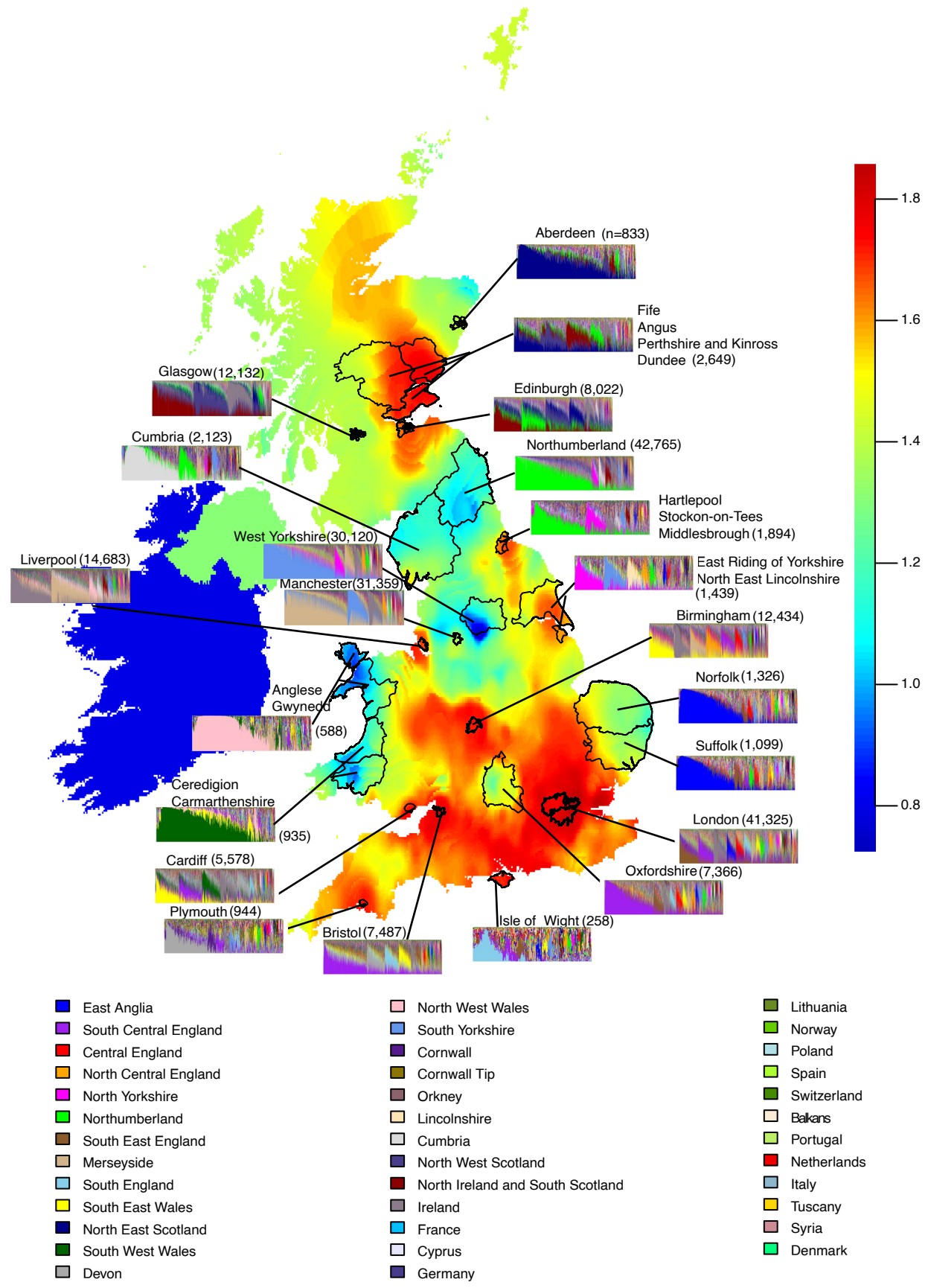

**Extended Data Fig. 3 | Heatmap of regional mean entropy statistic across the UK and Republic of Ireland.** Regional mean entropy indicates varying degrees of admixture in the UK. Areas with high entropy are coloured by red and areas with low entropy are coloured by blue. Selected regions of high or low entropy are highlighted by the boundaries, with the barplots detailing ancestry decomposition for each such region.

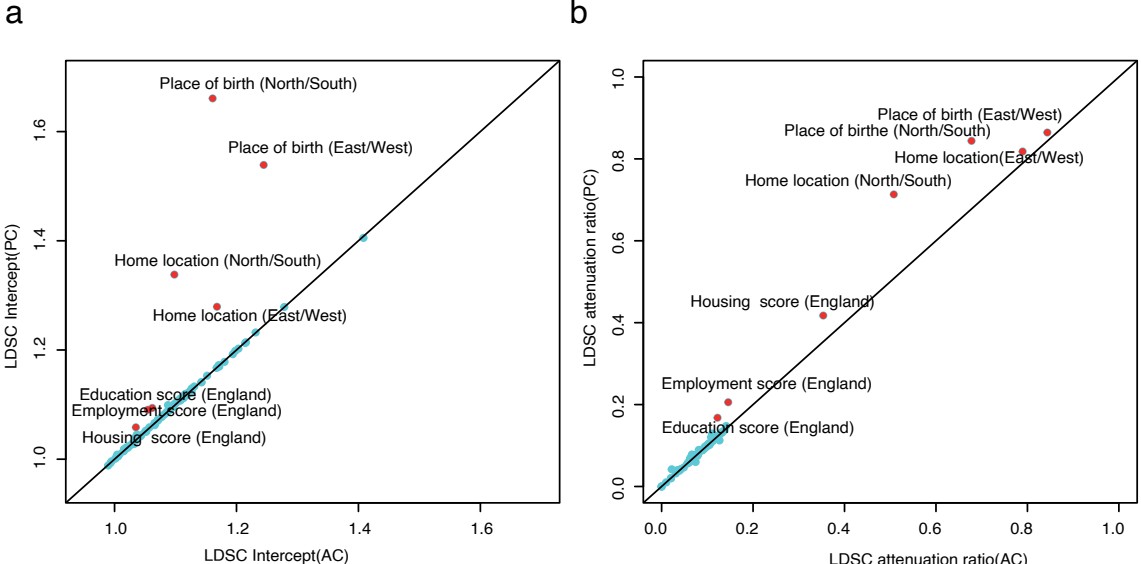

a

b

**Extended Data Fig. 4 | Comparison of inflation between ACs-corrected GWAS and PC-corrected GWAS across 104 quantitative traits by running LD score regression (LDSC). a**). Comparison of the LDSC intercept between AC-corrected GWAS (x-axis) and PC-corrected GWAS (y-axis) where red dots indicate phenotypes with intercept difference between PC-corrected GWAS and AC-corrected GWAS >0.015; **b**). Comparison of the LDSC attenuation ratio between AC-corrected GWAS (x-axis) and PC-correct GWAS (y-axis), where red dots indicate phenotypes with difference of attenuation ratio between PC-corrected GWAS and AC-corrected GWAS >0.02.

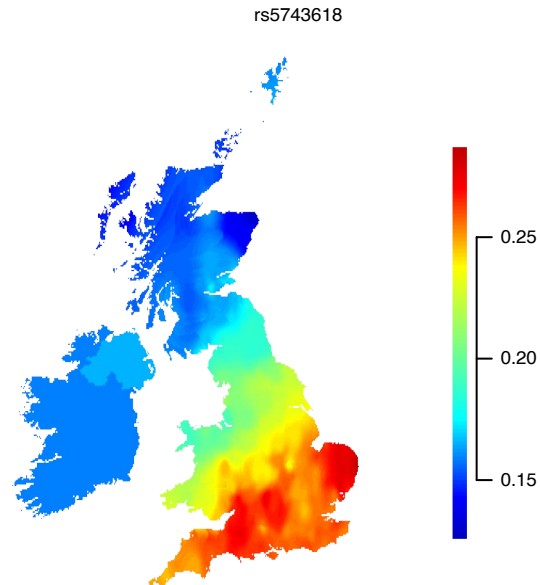

**Extended Data Fig. 5 | Heatmap of regional mean allele frequency of the SNP rs5743618.** Allele frequency is calculated based on genotype and birthplace among UK-born and Ireland-born UKB participants.

# PC loadings on chromosome

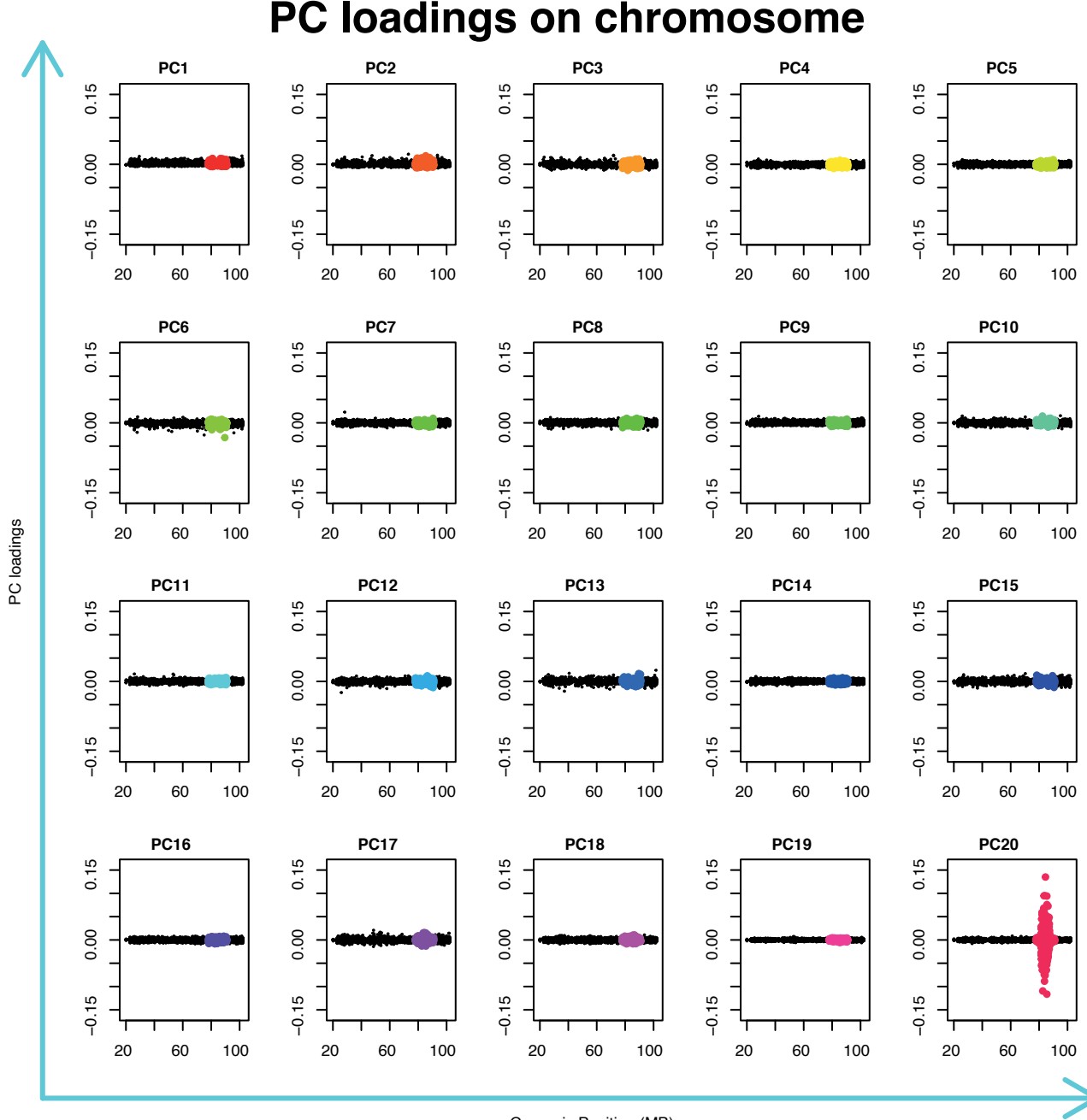

**Extended Data Fig. 6 | SNP loadings of the first 20 PCs along Chromosome 15.** For PC20, high loadings occur in the region 80–90 megabases.

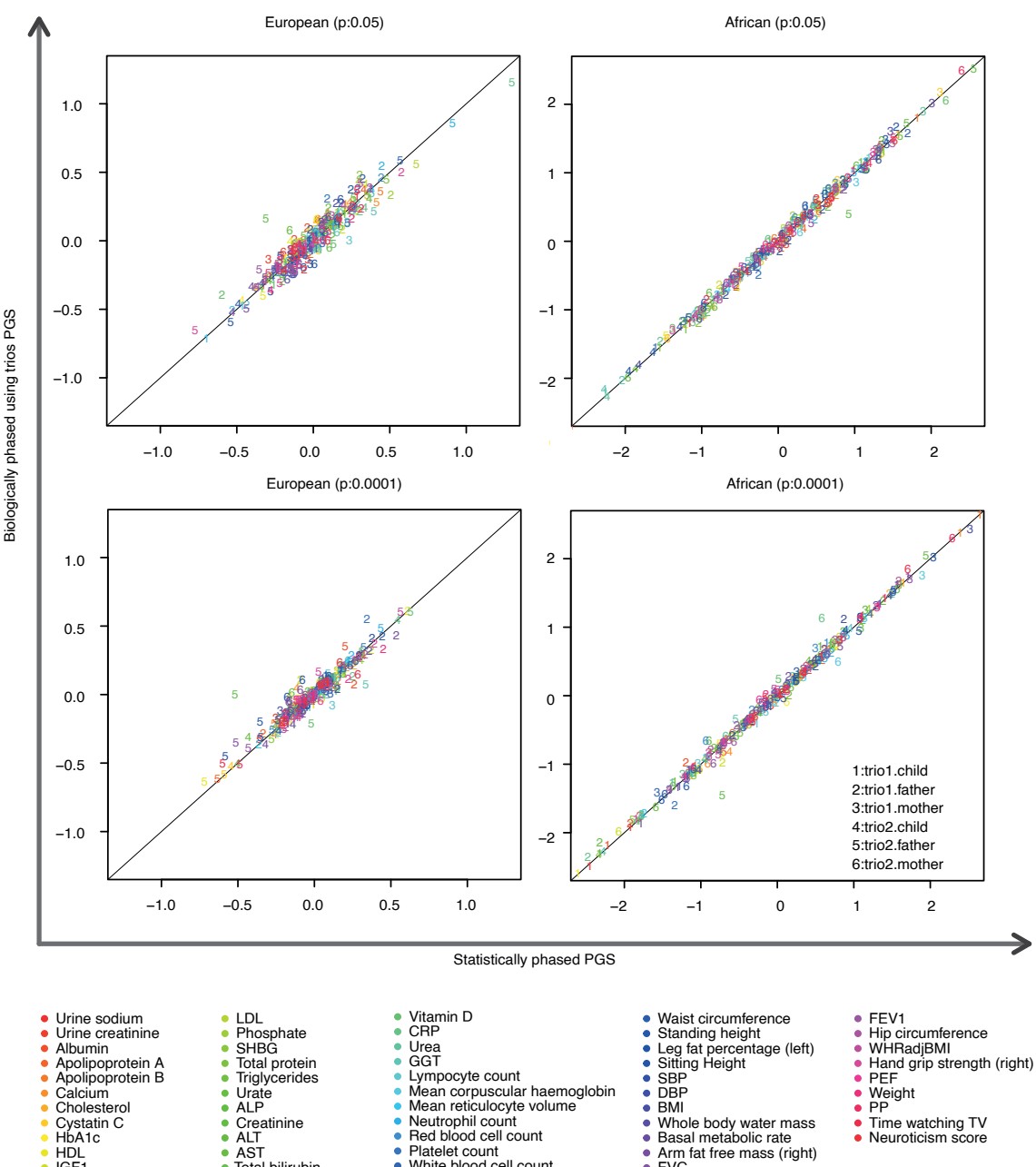

**Extended Data Fig. 7 | Comparison between PGS in trios calculated using either unphased genotypes (x-axis) or trio-phased haplotypes (y-axis) for 6 individuals from 2 trios across 53 traits.** For details of the normalization performed to enable joint comparison of these PGS, see 'Using trios to conduct ancestry aware phasing for PGSs'. In this plot, colours represent one of the 53 UKB phenotypes and numbers represent trio ids as shown in the bottom right panel. 1: child in the first trio; 2: father in the first trio; 3: mother in the first trio; 4: child in the second trio; 5: father in the second trio; 6: mother in the second trio. Top left panel: comparison of unphased and phased EPGS, calculated using only inferred European segments by using the default p-value 0.05 in EPGS construction; Top right panel: comparison of unphased and phased APGSs, calculated using only inferred African segments by using the default p-value 0.05 in EPGS construction; Bottom left panel: comparison of unphased and phased EPGSs, calculated using only inferred European segments by using the alternative p-value 0.0001 in EPGS construction; Bottom right panel: comparison of unphased and phased APGSs, calculated using only inferred African segments by using the alternative p-value 0.0001 in EPGS construction. Note that across traits, individuals and different p-value thresholds, the PGS are very similar.

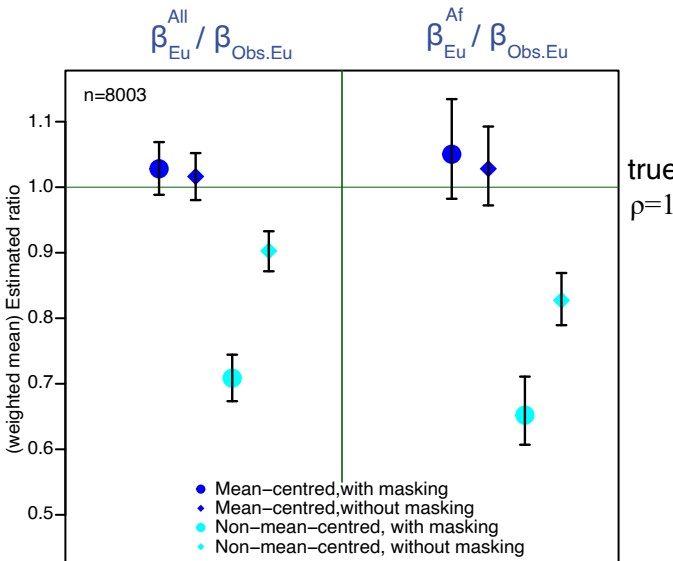

**Extended Data Fig. 8 | Comparison of combined estimate of $\rho$ for 24 simulated traits from PGSs constructed in combination of with/without mean-centring and with/without masking the uncertain and short segments under the null model.** True $\rho=1$ under the null model. Data are presented as (weighted) means with 95% central bootstrapped CIs. Dots separated by vertical line indicate different ratios of estimate: left panel: $\beta^{All}_{Eu}/\beta_{Obs.Eu}$ and right panel: $\beta^{Af}_{Eu}/\beta_{Obs.Eu}$. Horizontal line represents true setting of $\rho$. If confidence interval for each ratio of estimate intersects with $\rho=1$ line, the ratio estimate is accurate. Colour of the dots indicates if mean-centring is applied or not, and shape of the dots indicates if masking uncertain and short segments is applied or not.

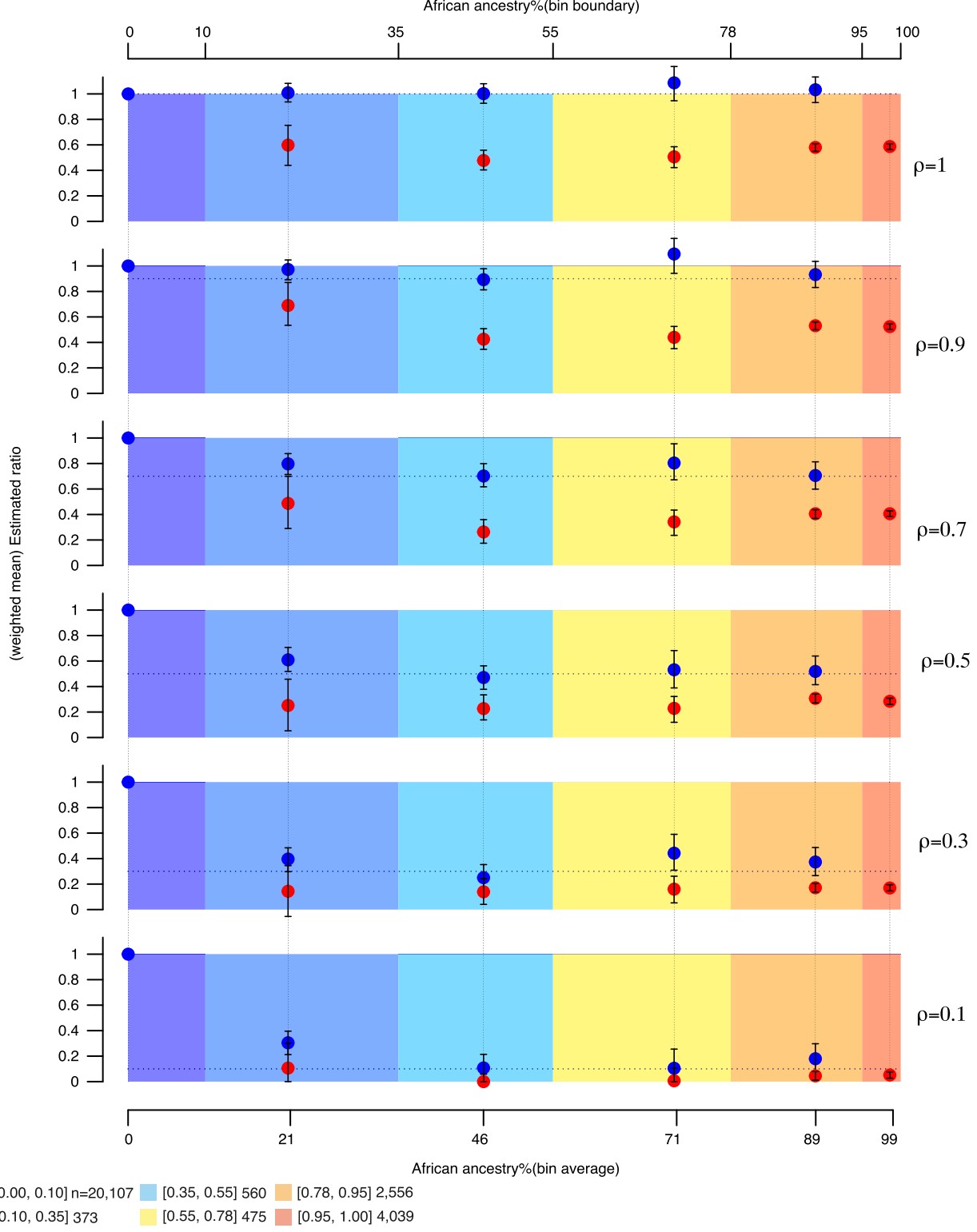

**Extended Data Fig. 9 | Results of application of ANCHOR for 24 UKB simulated traits in bins including individuals with varying African ancestry.** African ancestry is binned as in main Fig. 4c (x-axis; coloured region). Data are presented as (weighted) means with 95% central bootstrapped CIs. For each bin in each panel, estimates of coefficients $\beta_{Eu}/\beta_{Obs,Eu}$ (blue) and $\beta_{Af}/\beta_{Obs,Eu}$ (red) are shown with 95% bootstrapped confidence intervals, representing the ratio of PGS within European and African genomic regions to the PGS obtained from external European samples respectively (Methods). Also shown are these estimates from individuals of ~100% European ($\beta_{Obs,Eu}$; blue horizontal bar) or ~100% African (red horizontal bar) ancestry. From top to bottom, each panel represents descending correlation ρ. The case ρ=1, where only local effects impact portability, corresponds to blue points lying along the blue line, and similarly for red points, as observed. In cases when ρ<1, the dotted horizontal line is plotted at ρ, and the blue points are predicted to lie along this line if $\beta_{Eu}/\beta_{Obs,Eu}$ provides an accurate estimate of ρ, as predicted by theory (Methods; Supplementary Note 2).

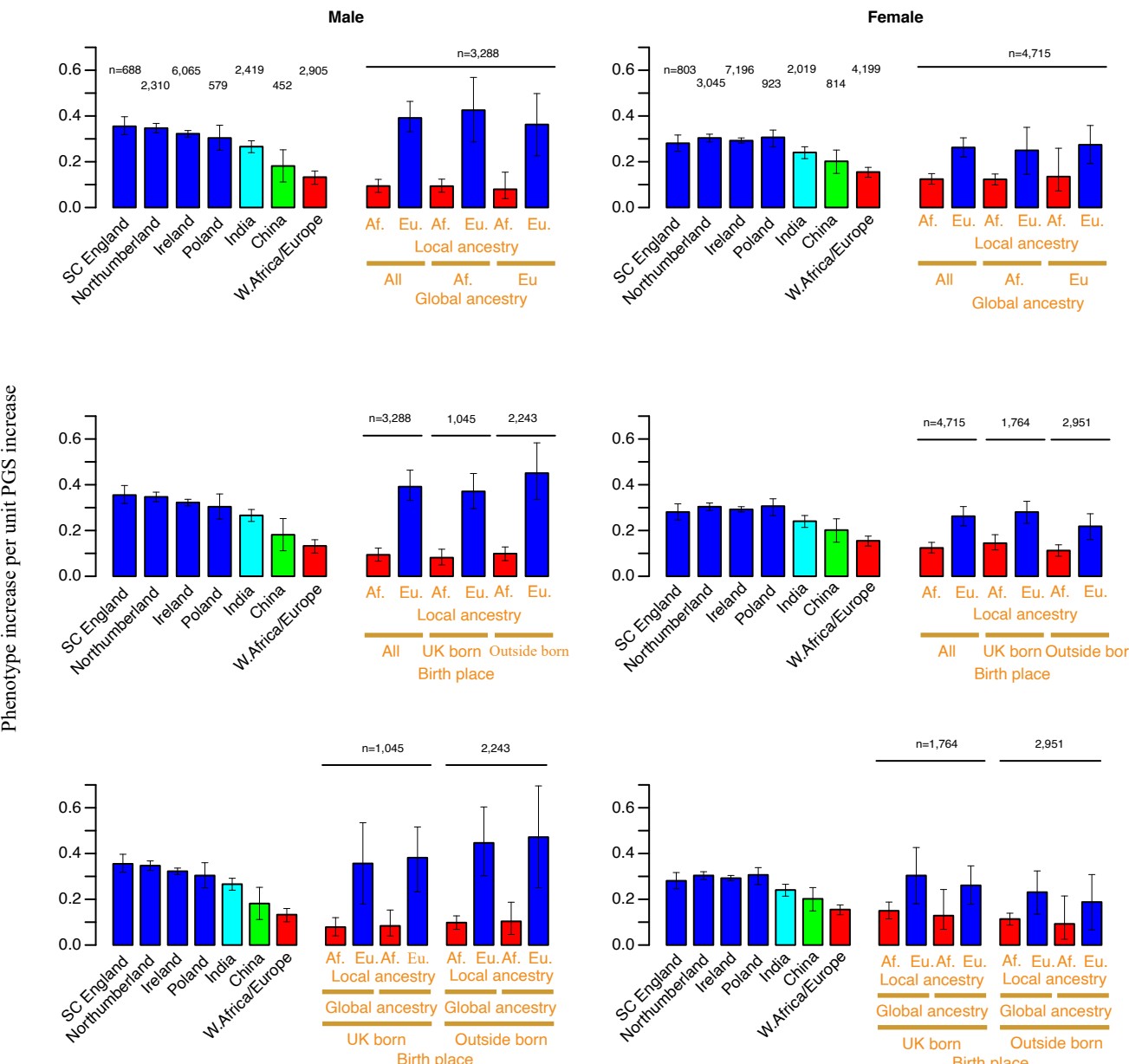

**Extended Data Fig. 10 | Investigation of impact of gender and UK vs non-UK birthplace on the relationship between PGS performance and ancestry, for standing height.** Data are presented as estimated values with 95% central bootstrapped CIs. For each plot, details are as described in the legend for Fig. 4d unless stated otherwise. Left column: analysis results for only male African-ancestry individuals; right column, female individuals. Top row: as Fig. 4d. Middle row: the last four columns in each plot estimate the estimated increase in phenotype per unit increase in the EPGS (blue) or APGS (red) polygenic score, separately for UK-born and non-UK-born individuals (note there is no significant difference in either case). Bottom row: as for the first row, but now applying the ANCHOR model to UK-born or non-UK-born individuals of the specified gender and varying ancestry (see section 'Joint model fitting by accounting for other environment factors'). Once again none of the bootstrapped CIs indicate significant impacts of birthplace on results; and neither gender shows significant evidence of different $\beta_{Eu}/\beta_{Obs.Eu}$ ratios (distinct effect sizes) for African-ancestry (right-hand blue bars) vs UK-ancestry (left hand two blue bars) individuals, indicating results are consistent with $\rho=1$, that is equal effect sizes in either gender and regardless of birthplace and ancestry. (There are obvious differences comparing males and females, corresponding to their different mean heights.).

# Reporting Summary

## Statistics

For all statistical analyses, confirm that the following items are present in the figure legend, table legend, main text, or Methods section.

| n/a | Confirmed | |
|---|---|---|
| ☐ | ☒ | The exact sample size (*n*) for each experimental group/condition, given as a discrete number and unit of measurement |
| ☐ | ☒ | A statement on whether measurements were taken from distinct samples or whether the same sample was measured repeatedly |
| ☐ | ☒ | The statistical test(s) used AND whether they are one- or two-sided<br>*Only common tests should be described solely by name; describe more complex techniques in the Methods section.* |
| ☐ | ☒ | A description of all covariates tested |
| ☐ | ☒ | A description of any assumptions or corrections, such as tests of normality and adjustment for multiple comparisons |
| ☐ | ☒ | A full description of the statistical parameters including central tendency (e.g. means) or other basic estimates (e.g. regression coefficient) AND variation (e.g. standard deviation) or associated estimates of uncertainty (e.g. confidence intervals) |
| ☐ | ☒ | For null hypothesis testing, the test statistic (e.g. *F*, *t*, *r*) with confidence intervals, effect sizes, degrees of freedom and *P* value noted<br>*Give P values as exact values whenever suitable.* |
| ☐ | ☒ | For Bayesian analysis, information on the choice of priors and Markov chain Monte Carlo settings |
| ☐ | ☒ | For hierarchical and complex designs, identification of the appropriate level for tests and full reporting of outcomes |
| ☐ | ☒ | Estimates of effect sizes (e.g. Cohen's *d*, Pearson's *r*), indicating how they were calculated |

*Our web collection on statistics for biologists contains articles on many of the points above.*

## Software and code

Policy information about availability of computer code

| Data collection | No data collection software was used in this study |
|---|---|
| Data analysis | Software developed in this study is being made publicly available via GitHub<br>SHAPEIT2, IMPUTE4, ChromoPainter(V2), plink1.9, plink2.0, R(v4.1.0), R packages: "sp" (v1.6.0), "raster" (v3.6.14), "mrcieu/ieugwasr", "pcapred.largedata" and "pcapred", BGENIE (v1.2), Bolt-LMM (v2.4.1), flashpca2, LDSC(vl.0.1), king (v2.2),qctool 2, bgenix (v1.1.3), hapmix (v2)<br><br>Custom code and programs developed as part of the study: ANCHOR (https://github.com/MyersGroup/ANCHOR), R/shell scripts (https://github.com/fuopen/UKB_anc) |

For manuscripts utilizing custom algorithms or software that are central to the research but not yet described in published literature, software must be made available to editors and reviewers. We strongly encourage code deposition in a community repository (e.g. GitHub). See the Nature Portfolio guidelines for submitting code & software for further information.

# Data

Policy information about <u>availability of data</u>

All manuscripts must include a <u>data availability statement</u>. This statement should provide the following information, where applicable:
- Accession codes, unique identifiers, or web links for publicly available datasets
- A description of any restrictions on data availability
- For clinical datasets or third party data, please ensure that the statement adheres to our <u>policy</u>

UK and world map data can be accessed through: https://gadm.org.
UK Biobank data can be downloaded by approved researchers through: https://www.ukbiobank.ac.uk.
Phased haplotype data from 1000 Genome used as reference panel for HAPMIX can be accessed through: https://www.internationalgenome.org/category/data-access/.
POBI was accessed using #EGAS00001000672.
GWAS summary level data used in this paper can be queried using the interface implemented by "mrcieu/ieugwasr": https://gwas.mrcieu.ac.uk and through Open Target: https://www.opentargets.org.
HapMap3 variants list file can be accessed at: https://ftp.ncbi.nlm.nih.gov/hapmap/.
All genetic data used in constructing the ancestry pipeline is provided by third parties and is available for use by others.
Variant Frequency information for every SNP in each population is available at the University of Bristol data repository, data.bris, at https://doi.org/10.5523/bris.3g5oatl682kz82as80jakjrq91.
All other resources were downloaded from their respective websites without registration requirements. The following files were generated by this study and have been returned to UK Biobank in order that they might be made available to other researchers, in accordance with UK Biobank policy: (a) ACs on all UK Biobank participants (b) population specific allele frequency estimates for 25M variants (c) (Mean-centred) European/African genotypes and local ancestry of 8003 UKB African ancestry individuals (including the variants annotation information) (d) European/African PGS for 8003 African ancestry individuals across 53 UKB phenotypes. GWAS summary statistics files with population structure corrected by ACs and PCs are available in GWAS catalog (https://www.ebi.ac.uk/gwas/, GCST90310137- GCST90310200 and GCST90429571- GCST90429610).

# Research involving human participants, their data, or biological material

Policy information about studies with <u>human participants or human data</u>. See also policy information about <u>sex, gender (identity/presentation), and sexual orientation</u> and <u>race, ethnicity and racism</u>.

| | |
|---|---|
| Reporting on sex and gender | We conducted necessary comparisons between genders for phenotypes differing between sexes e.g. height, to ensure some of our results apply regardless of biological sex. We have read and believe we have complied with Nature Portfolio journals' editorial policies for such analyses. |
| Reporting on race, ethnicity, or other socially relevant groupings | A key part of our study is in comparing and improving applicability of GWAS findings between human populations defined genetically. We also sometimes utilise or stratify by self-declared ethnicity as gathered by the UKB. We have taken care in the approaches used, definition of groups studied, and use of language throughout this work. We have read and believe we have fully complied with Nature Portfolio journals' editorial policies for such analyses. |
| Population characteristics | We do define populations or genetic grouping in this study where necessary for the purposes of study and comparisons on "portability" of GWAS into these groups. We describe how these groups are identified, and investigate characteristics of these groups (only) as they relate to geography, and efficacy of polygenic scores in each group. We otherwise have no discussion of population characteristics; again we believe we comply fully with editorial policies for such analyses. <br><br> Covariate-relevant population characteristics of the human research participants used in this study includes: age, sex, birth place, birth country, ancestry components (ACs) and principal components (PCs). |
| Recruitment | N/A (all subjects were previously recruited by UKB with appropriate ethical approval) |
| Ethics oversight | N/A(our work does not require explicit additional ethical approval beyond that possessed by the UKB. It is conducted under a valid UK Biobank data access agreement acknowledged in the paper. We have complied with UKB data use policies regarding the analysis of fully anonymised subjects. |

Note that full information on the approval of the study protocol must also be provided in the manuscript.

# Field-specific reporting

Please select the one below that is the best fit for your research. If you are not sure, read the appropriate sections before making your selection.

☒ Life sciences  ☐ Behavioural & social sciences  ☐ Ecological, evolutionary & environmental sciences

For a reference copy of the document with all sections, see <u>nature.com/documents/nr-reporting-summary-flat.pdf</u>

# Life sciences study design

All studies must disclose on these points even when the disclosure is negative.

| | |
|---|---|
| Sample size | Sample size has been clearly disclosed in the manuscript for each different data/sub-data sets. Sample size of Ancestry specific analysis in the polygenic score section was determined by the corresponding genetic ancestry coefficients inferred by our method. Trio individuals with African-background ancestry were detected by open source software "king (v2.2)"<br><br>In analysis of fine-scale population structure in the UK Biobank, we use all the available samples unless individuals were listed in the withdrawn list. In GWAS analysis, White British samples for GWAS analysis was selected completely the same as the UKB data flagship paper (Bycroft C. et.al. Nature 2018).<br>In the polygenic score analysis, we choose 7 ancestry groups which sufficiently represent the main population groups in the UK, and individuals in each groups were chosen purely based on their ancestry inference. No individuals were deliberately dropped in the analysis unless they were in the UKB withdrawn list. 8,003 individuals were chosen in the ANCHOR analysis, and sample selection is purely based on the their European and West African ancestry information. Such sample selection criteria ensures the maximum available samples have been used and results based on those samples were accurate and less-biased in the UKB. No samples were deliberately dropped in the analysis unless they were in the UKB withdrawn list. |
| Data exclusions | No data was excluded unless the participants withdrew their participation from the UKB |
| Replication | All our analyses relate to UK Biobank samples; however, we have used independent test data in many places to validate and replicate our findings. We also made steps to ensure our results could be replicated by other researchers and our methods applied in other groups. All of the codes and scripts used in our experiments are available in open-source format and can be accessed freely for academic research. Versions of the software/tools used in our analysis have been documented. |
| Randomization | No randomization was used, since there are no experimental groups so it is not relevant to our study. |
| Blinding | No blinding is used as it is not relevant to this study: however group allocation is objective using inferred genetic ancestry. |

# Behavioural & social sciences study design

All studies must disclose on these points even when the disclosure is negative.

| | |
|---|---|
| Study description | *Briefly describe the study type including whether data are quantitative, qualitative, or mixed-methods (e.g. qualitative cross-sectional, quantitative experimental, mixed-methods case study).* |
| Research sample | *State the research sample (e.g. Harvard university undergraduates, villagers in rural India) and provide relevant demographic information (e.g. age, sex) and indicate whether the sample is representative. Provide a rationale for the study sample chosen. For studies involving existing datasets, please describe the dataset and source.* |
| Sampling strategy | *Describe the sampling procedure (e.g. random, snowball, stratified, convenience). Describe the statistical methods that were used to predetermine sample size OR if no sample-size calculation was performed, describe how sample sizes were chosen and provide a rationale for why these sample sizes are sufficient. For qualitative data, please indicate whether data saturation was considered, and what criteria were used to decide that no further sampling was needed.* |
| Data collection | *Provide details about the data collection procedure, including the instruments or devices used to record the data (e.g. pen and paper, computer, eye tracker, video or audio equipment) whether anyone was present besides the participant(s) and the researcher, and whether the researcher was blind to experimental condition and/or the study hypothesis during data collection.* |
| Timing | *Indicate the start and stop dates of data collection. If there is a gap between collection periods, state the dates for each sample cohort.* |
| Data exclusions | *If no data were excluded from the analyses, state so OR if data were excluded, provide the exact number of exclusions and the rationale behind them, indicating whether exclusion criteria were pre-established.* |
| Non-participation | *State how many participants dropped out/declined participation and the reason(s) given OR provide response rate OR state that no participants dropped out/declined participation.* |
| Randomization | *If participants were not allocated into experimental groups, state so OR describe how participants were allocated to groups, and if allocation was not random, describe how covariates were controlled.* |

# Ecological, evolutionary & environmental sciences study design

All studies must disclose on these points even when the disclosure is negative.

| | |
|---|---|
| Study description | *Briefly describe the study. For quantitative data include treatment factors and interactions, design structure (e.g. factorial, nested, hierarchical), nature and number of experimental units and replicates.* |

| Research sample | *Describe the research sample (e.g. a group of tagged Passer domesticus, all Stenocereus thurberi within Organ Pipe Cactus National Monument), and provide a rationale for the sample choice. When relevant, describe the organism taxa, source, sex, age range and any manipulations. State what population the sample is meant to represent when applicable. For studies involving existing datasets, describe the data and its source.* |
|---|---|
| Sampling strategy | *Note the sampling procedure. Describe the statistical methods that were used to predetermine sample size OR if no sample-size calculation was performed, describe how sample sizes were chosen and provide a rationale for why these sample sizes are sufficient.* |
| Data collection | *Describe the data collection procedure, including who recorded the data and how.* |
| Timing and spatial scale | *Indicate the start and stop dates of data collection, noting the frequency and periodicity of sampling and providing a rationale for these choices. If there is a gap between collection periods, state the dates for each sample cohort. Specify the spatial scale from which the data are taken* |
| Data exclusions | *If no data were excluded from the analyses, state so OR if data were excluded, describe the exclusions and the rationale behind them, indicating whether exclusion criteria were pre-established.* |
| Reproducibility | *Describe the measures taken to verify the reproducibility of experimental findings. For each experiment, note whether any attempts to repeat the experiment failed OR state that all attempts to repeat the experiment were successful.* |
| Randomization | *Describe how samples/organisms/participants were allocated into groups. If allocation was not random, describe how covariates were controlled. If this is not relevant to your study, explain why.* |
| Blinding | *Describe the extent of blinding used during data acquisition and analysis. If blinding was not possible, describe why OR explain why blinding was not relevant to your study.* |

Did the study involve field work?  ☐ Yes  ☐ No

## Field work, collection and transport

| Field conditions | *Describe the study conditions for field work, providing relevant parameters (e.g. temperature, rainfall).* |
|---|---|
| Location | *State the location of the sampling or experiment, providing relevant parameters (e.g. latitude and longitude, elevation, water depth).* |
| Access & import/export | *Describe the efforts you have made to access habitats and to collect and import/export your samples in a responsible manner and in compliance with local, national and international laws, noting any permits that were obtained (give the name of the issuing authority, the date of issue, and any identifying information).* |
| Disturbance | *Describe any disturbance caused by the study and how it was minimized.* |

# Reporting for specific materials, systems and methods

We require information from authors about some types of materials, experimental systems and methods used in many studies. Here, indicate whether each material, system or method listed is relevant to your study. If you are not sure if a list item applies to your research, read the appropriate section before selecting a response.

## Materials & experimental systems

| n/a | Involved in the study |
|---|---|
| ☒ | ☐ Antibodies |
| ☒ | ☐ Eukaryotic cell lines |
| ☒ | ☐ Palaeontology and archaeology |
| ☒ | ☐ Animals and other organisms |
| ☒ | ☐ Clinical data |
| ☒ | ☐ Dual use research of concern |
| ☒ | ☐ Plants |

## Methods

| n/a | Involved in the study |
|---|---|
| ☒ | ☐ ChIP-seq |
| ☒ | ☐ Flow cytometry |
| ☒ | ☐ MRI-based neuroimaging |

## Antibodies

| Antibodies used | N/A |
|---|---|
| Validation | N/A |

# Eukaryotic cell lines

Policy information about cell lines and Sex and Gender in Research

| | |
|---|---|
| Cell line source(s) | N/A |
| Authentication | N/A |
| Mycoplasma contamination | N/A |
| Commonly misidentified lines (See ICLAC register) | N/A |

# Palaeontology and Archaeology

| | |
|---|---|
| Specimen provenance | N/A |
| Specimen deposition | N/A |
| Dating methods | N/A |

☐ Tick this box to confirm that the raw and calibrated dates are available in the paper or in Supplementary Information.

| | |
|---|---|
| Ethics oversight | N/A |

Note that full information on the approval of the study protocol must also be provided in the manuscript.

# Animals and other research organisms

Policy information about studies involving animals; ARRIVE guidelines recommended for reporting animal research, and Sex and Gender in Research

| | |
|---|---|
| Laboratory animals | N/A |
| Wild animals | N/A |
| Reporting on sex | N/A |
| Field-collected samples | N/A |
| Ethics oversight | N/A |

Note that full information on the approval of the study protocol must also be provided in the manuscript.

# Clinical data

Policy information about clinical studies
All manuscripts should comply with the ICMJE guidelines for publication of clinical research and a completed CONSORT checklist must be included with all submissions.

| | |
|---|---|
| Clinical trial registration | N/A |
| Study protocol | N/A |
| Data collection | N/A |
| Outcomes | N/A |

# Dual use research of concern

Policy information about dual use research of concern

## Hazards

Could the accidental, deliberate or reckless misuse of agents or technologies generated in the work, or the application of information presented in the manuscript, pose a threat to:

| No | Yes | |
|---|---|---|
| ☒ | ☐ | Public health |
| ☒ | ☐ | National security |
| ☒ | ☐ | Crops and/or livestock |
| ☒ | ☐ | Ecosystems |
| ☒ | ☐ | Any other significant area |

## Experiments of concern

Does the work involve any of these experiments of concern:

| No | Yes | |
|---|---|---|
| ☒ | ☐ | Demonstrate how to render a vaccine ineffective |
| ☒ | ☐ | Confer resistance to therapeutically useful antibiotics or antiviral agents |
| ☒ | ☐ | Enhance the virulence of a pathogen or render a nonpathogen virulent |
| ☒ | ☐ | Increase transmissibility of a pathogen |
| ☒ | ☐ | Alter the host range of a pathogen |
| ☒ | ☐ | Enable evasion of diagnostic/detection modalities |
| ☒ | ☐ | Enable the weaponization of a biological agent or toxin |
| ☒ | ☐ | Any other potentially harmful combination of experiments and agents |

# Plants

| | |
|---|---|
| Seed stocks | *Report on the source of all seed stocks or other plant material used. If applicable, state the seed stock centre and catalogue number. If plant specimens were collected from the field, describe the collection location, date and sampling procedures.* |
| Novel plant genotypes | *Describe the methods by which all novel plant genotypes were produced. This includes those generated by transgenic approaches, gene editing, chemical/radiation-based mutagenesis and hybridization. For transgenic lines, describe the transformation method, the number of independent lines analyzed and the generation upon which experiments were performed. For gene-edited lines, describe the editor used, the endogenous sequence targeted for editing, the targeting guide RNA sequence (if applicable) and how the editor was applied.* |
| Authentication | *Describe any authentication procedures for each seed stock used or novel genotype generated. Describe any experiments used to assess the effect of a mutation and, where applicable, how potential secondary effects (e.g. second site T-DNA insertions, mosiacism, off-target gene editing) were examined.* |

# ChIP-seq

## Data deposition

☐ Confirm that both raw and final processed data have been deposited in a public database such as GEO.

☐ Confirm that you have deposited or provided access to graph files (e.g. BED files) for the called peaks.

| | |
|---|---|
| Data access links<br>*May remain private before publication.* | *For "Initial submission" or "Revised version" documents, provide reviewer access links. For your "Final submission" document, provide a link to the deposited data.* |
| Files in database submission | *Provide a list of all files available in the database submission.* |
| Genome browser session<br>(e.g. UCSC) | *Provide a link to an anonymized genome browser session for "Initial submission" and "Revised version" documents only, to enable peer review. Write "no longer applicable" for "Final submission" documents.* |

## Methodology

| | |
|---|---|
| Replicates | *Describe the experimental replicates, specifying number, type and replicate agreement.* |
| Sequencing depth | *Describe the sequencing depth for each experiment, providing the total number of reads, uniquely mapped reads, length of reads and whether they were paired- or single-end.* |
| Antibodies | *Describe the antibodies used for the ChIP-seq experiments; as applicable, provide supplier name, catalog number, clone name, and lot number.* |
| Peak calling parameters | *Specify the command line program and parameters used for read mapping and peak calling, including the ChIP, control and index files used.* |

| Data quality | *Describe the methods used to ensure data quality in full detail, including how many peaks are at FDR 5% and above 5-fold enrichment.* |
| Software | *Describe the software used to collect and analyze the ChIP-seq data. For custom code that has been deposited into a community repository, provide accession details.* |

# Flow Cytometry

## Plots

Confirm that:

☐ The axis labels state the marker and fluorochrome used (e.g. CD4-FITC).

☐ The axis scales are clearly visible. Include numbers along axes only for bottom left plot of group (a 'group' is an analysis of identical markers).

☐ All plots are contour plots with outliers or pseudocolor plots.

☐ A numerical value for number of cells or percentage (with statistics) is provided.

## Methodology

| Sample preparation | *Describe the sample preparation, detailing the biological source of the cells and any tissue processing steps used.* |
| Instrument | *Identify the instrument used for data collection, specifying make and model number.* |
| Software | *Describe the software used to collect and analyze the flow cytometry data. For custom code that has been deposited into a community repository, provide accession details.* |
| Cell population abundance | *Describe the abundance of the relevant cell populations within post-sort fractions, providing details on the purity of the samples and how it was determined.* |
| Gating strategy | *Describe the gating strategy used for all relevant experiments, specifying the preliminary FSC/SSC gates of the starting cell population, indicating where boundaries between "positive" and "negative" staining cell populations are defined.* |

☐ Tick this box to confirm that a figure exemplifying the gating strategy is provided in the Supplementary Information.

# Magnetic resonance imaging

## Experimental design

| Design type | *Indicate task or resting state; event-related or block design.* |
| Design specifications | *Specify the number of blocks, trials or experimental units per session and/or subject, and specify the length of each trial or block (if trials are blocked) and interval between trials.* |
| Behavioral performance measures | *State number and/or type of variables recorded (e.g. correct button press, response time) and what statistics were used to establish that the subjects were performing the task as expected (e.g. mean, range, and/or standard deviation across subjects).* |

## Acquisition

| Imaging type(s) | *Specify: functional, structural, diffusion, perfusion.* |
| Field strength | *Specify in Tesla* |
| Sequence & imaging parameters | *Specify the pulse sequence type (gradient echo, spin echo, etc.), imaging type (EPI, spiral, etc.), field of view, matrix size, slice thickness, orientation and TE/TR/flip angle.* |
| Area of acquisition | *State whether a whole brain scan was used OR define the area of acquisition, describing how the region was determined.* |

Diffusion MRI  ☐ Used  ☐ Not used

## Preprocessing

| Preprocessing software | *Provide detail on software version and revision number and on specific parameters (model/functions, brain extraction, segmentation, smoothing kernel size, etc.).* |
| Normalization | *If data were normalized/standardized, describe the approach(es): specify linear or non-linear and define image types used for transformation OR indicate that data were not normalized and explain rationale for lack of normalization.* |

| | |
|---|---|
| Normalization template | *Describe the template used for normalization/transformation, specifying subject space or group standardized space (e.g. original Talairach, MNI305, ICBM152) OR indicate that the data were not normalized.* |
| Noise and artifact removal | *Describe your procedure(s) for artifact and structured noise removal, specifying motion parameters, tissue signals and physiological signals (heart rate, respiration).* |
| Volume censoring | *Define your software and/or method and criteria for volume censoring, and state the extent of such censoring.* |

## Statistical modeling & inference

| | |
|---|---|
| Model type and settings | *Specify type (mass univariate, multivariate, RSA, predictive, etc.) and describe essential details of the model at the first and second levels (e.g. fixed, random or mixed effects; drift or auto-correlation).* |
| Effect(s) tested | *Define precise effect in terms of the task or stimulus conditions instead of psychological concepts and indicate whether ANOVA or factorial designs were used.* |

Specify type of analysis: ☐ Whole brain  ☐ ROI-based  ☐ Both

| | |
|---|---|
| Statistic type for inference<br><br>(See Eklund et al. 2016) | *Specify voxel-wise or cluster-wise and report all relevant parameters for cluster-wise methods.* |
| Correction | *Describe the type of correction and how it is obtained for multiple comparisons (e.g. FWE, FDR, permutation or Monte Carlo).* |

## Models & analysis

n/a | Involved in the study
☐ | ☐ Functional and/or effective connectivity
☐ | ☐ Graph analysis
☐ | ☐ Multivariate modeling or predictive analysis

| | |
|---|---|
| Functional and/or effective connectivity | *Report the measures of dependence used and the model details (e.g. Pearson correlation, partial correlation, mutual information).* |
| Graph analysis | *Report the dependent variable and connectivity measure, specifying weighted graph or binarized graph, subject- or group-level, and the global and/or node summaries used (e.g. clustering coefficient, efficiency, etc.).* |
| Multivariate modeling and predictive analysis | *Specify independent variables, features extraction and dimension reduction, model, training and evaluation metrics.* |

