## [Peer Review File · Nature Genetics]

Fine-scale population structure and widespread conservation of genetic effect sizes between human groups across traits

Corresponding Author: Professor Simon Myers

Version 0:

Decision Letter:

27th Oct 2023

Dear Simon,

Firstly, our sincere apologies to you and your co-authors for the extreme length of time in review - unfortunately we had to wait quite a while for a second review to come in such that we had enough feedback for a decision!

Your Article, "Leveraging fine-scale population structure reveals conservation in genetic effect sizes between human populations across a range of human phenotypes" has now been seen by 2 referees. You will see from their comments below that while they find your work of interest, some important points are raised. We are interested in the possibility of publishing your study in Nature Genetics, but would like to consider your response to these concerns in the form of a revised manuscript before we make a final decision on publication.

In brief, the two reviews both acknowledge the potential interest in your work, although we note that they disagree on which is the more impactful and important results (ACs/population structure vs. the PGS/effect sizes).

Amongst other specific points, Referee #1 highlights some issues with the ancestry analysis, suggesting that the methodological basis needs to be clarified (we note Reviewer #2 agrees with this). Reviewer #2 thinks that some of the stronger claims made are not supported by the results presented.

In our reading of these reviews, we think there is a clear path to publication; none of the requested additions seem unduly onerous, so we hope you will be able to completely fulfill all of these specific requests. We'd particularly highlight the fact that both reviewers find the presentation difficult to follow, and we'd recommend that you ask a non-expert colleague to read a revision to ensure it's accessible!

To guide the scope of the revisions, the editors discuss the referee reports in detail within the team, including with the chief editor, with a view to identifying key priorities that should be addressed in revision and sometimes overruling referee requests that are deemed beyond the scope of the current study. We hope that you will find the prioritized set of referee points to be useful when revising your study. Please do not hesitate to get in touch if you would like to discuss these issues further.

We therefore invite you to revise your manuscript taking into account all reviewer and editor comments. Please highlight all changes in the manuscript text file. At this stage we will need you to upload a copy of the manuscript in MS Word .docx or similar editable format.

*2) If you have not done so already please begin to revise your manuscript so that it conforms to our Article format instructions, available

[here](http://www.nature.com/ng/authors/article_types/index.html).
Refer also to any guidelines provided in this letter.

*3) Include a revised version of any required Reporting Summary: <https://www.nature.com/documents/nr-reporting-summary.pdf>

Link Redacted

Sincerely,

Michael Fletcher, PhD
Senior Editor, Nature Genetics

ORCID: 0000-0003-1589-7087

Referee expertise: statistical genetics.

Reviewers' Comments:

Reviewer #1:

Remarks to the Author:

Hu et al. developed an ancestry inference pipeline to break down an individual's genome into ancestry components (ACs) using phasing and chromosome painting with carefully constructed reference panels. These ACs serve as powerful tools for investigating the interplay between genetic ancestry, geography, and self-identified ancestry. They also outperform principal components (PCs) in capturing fine-scale population structure, thereby aiding in mitigating confounding in genome-wide association studies (GWAS). The study also examined the portability of polygenic scores (PGS) in 8,000 admixed individuals, decomposing PGS into African-like (APGS) and European-like (EPGS) segments. Their findings suggest that the performance of APGS and EPGS is independent of global ancestry. Based on these findings, they argue that causal effect sizes are likely similar across ancestries.

The paper appears to consist of two somewhat disjointed projects. The first part involves inferring fine-scale population structure and ancestry components (ACs) (Figures 1-3), which I believe constitutes a significant contribution to the field. The second part focuses on polygenic score (PGS) performance in admixed samples (Figure 4). These two parts may not seamlessly integrate into a single manuscript. I appreciate that ACs do play a role in the PGS analysis, both in the GWAS step and in identifying admixed samples. However, as the authors themselves stated, ACs and principal components (PCs) yield similar results in this context. Also, defining broadly African- and European-like segments may not heavily rely on inferring fine-scale population structure. Given these considerations, I find the manuscript title inaccurate, as I do not believe the authors "leveraged" fine-scale population structure in their PGS analyses. Additionally, as I outline below, I think the second part is less clear and convincing than the first part; therefore, the predominant emphasis of the paper on the PGS results, as presented, overshadows ACs, which I consider a significant—and in my subjective opinion, the main—contribution of the paper.

I will split my major comments into two sections to address Figures 1-3 and Figure 4 separately.

Major comments:

Related to Figures 1-3:

- The labels for the 127 reference populations do not seem to be listed anywhere (unless I have overlooked them). It would be ideal to display these labels on the geographic maps, as shown in Figure 2. This would facilitate the interpretation of the results.
- In Figure 2b, the colored regions are not explicitly delineated; it's evident that they do not align with country borders but are associated (via white lines) with bar plots that are separated by countries.
- In Figures 3c-e, the logical use of previously detected associations (coloring of points) appears inconsistent: in 3d, previous associations are used as evidence for true positives, while in 3c, numerous variants claimed to be false positives are labeled with previous associations.
- The arguments supporting selection on the hay fever variants in Figure 3c appear speculative. In general, if the geographical distribution of a particular variant does not align with ancestry for any reason, ACs may not effectively account for confounding in a birthplace GWAS. For instance, the distribution of the hay fever variant might reflect proximity to urban versus rural areas (e.g., if individuals experiencing more severe allergic reactions choose to reside in regions with lower allergen exposure), rather than being attributed to selection.
- It would be beneficial for the scientific community if the authors could provide summary statistics for GWAS with AC corrections for the traits they have analyzed, if not already done.

Related to Figure 4:

- The authors' rationale for mean-centering genotypes by local ancestry is inadequately explained or justified, even in the supplementary note. This is a crucial aspect, especially given that the authors assert that their results are sensitive to this procedure, and indeed, they do observe differences in effects with non-centered genotypes. Specifically, there is no apparent reason to believe that equation (3) in the supplementary note 2 is the correct generative model. As formulated, the interpretation of regression coefficients is challenging to comprehend. For example, under the assumption of no interactions and no LD differences across populations, equation (3) models the effect of the genotype as dependent on local ancestry, which is evidently incorrect. Seemingly, the only reason mean-centering was performed was that not doing so yields different results. The authors should provide clear statistical and biological reasoning as to why they made this choice. If they have empirically determined in simulations that mean-centering is indeed necessary, further investigation is warranted to gain a deeper understanding of this phenomenon.
- The ratio $B_{eu}/B_{obs.eu}$ serves as an estimator of ρ when the causal effect sizes exhibit comparable variance across populations. However, effect size variance may vary across populations due to factors such as assortative mating, the strength of stabilizing selection, or environmental influences modulating variance. Consequently, $B_{eu}/B_{obs.eu}=1$ does not necessarily imply $\rho=1$.
- The confidence intervals (CIs) are notably large, especially for mixed ancestry groups. This could be attributed to the fact that the majority of the admixed individuals (approximately 6600 out of 8000) have more than 90% African ancestry. Consequently, I have concerns about whether the analysis is sufficiently powered to detect deviations from the null of no interactions.
- The choice of traits is somewhat limited, and the selected complex traits appear to be somewhat correlated. It might be worthwhile to expand the analysis to include more traits, especially those that are potentially more influenced by interactions, such as behavioral traits like neuroticism scores.
- The PGS model constructed based on pruning and thresholding uses a relatively high p-value threshold of 0.05, which may not be optimal. The noisy effect estimates of weak associations likely hamper the performance of PGS, although I note that they will not bias the analyses.
- The section title "biological effects are highly similar..." is too strong. It is not clear what is meant by "biological effects." The authors likely refer to the "effect of causal variants."
- Regarding GxE interactions, it is plausible that environmental effects do not track ancestry in the UK, but this may not apply universally to other regions, such as the US. Thus, statements like "providing evidence that gene-environment and gene-gene interactions do not play major roles in the poor prediction of European-ancestry PRS scores in African populations" are overly general. The paper contains many such blanket statements that are not substantiated by the presented analyses.
- The figure is generally too challenging to understand.

Minor comments:

- There are numerous typos in the paper and supplementary notes, such as "focussing" in the abstract.

- Figure captions are generally uninformative.
- The use of the Beta symbol for both GWAS effects and the regression coefficient for PGS is confusing.
- I presume by "Bangladesh" in line 211 the authors mean "Gujarat".
- Regarding ACs predicting PCs and vice versa, the text could clarify that predictions are based on "linear" combinations of ACs or PCs.
- Given that the true positives and negatives are unknown in Figures 3c-e, the language should always be conservative, using terms like "likely" false positives or "likely" false negatives.
- Clarify the unit of the y-axis in Figure 4c. I assume that phenotypic values are normalized to enable the averaging of PGS across traits.
- In the caption of Figure 2, specify that the color legends are different for panels a and b. Additionally, in panel b, the legend for the "ethnicity key" should clarify that these identifications are self-reported.
- Consider presenting the legend outside the plot regions in Figure 3d for clarity. Additionally, it is not apparent whether the shape of the points adds value to the message of the figure.
- I suggest avoiding the statement "data not shown", and instead, providing plots or tables to support claims.

Reviewer #2:

Remarks to the Author:

Hu et al review:

Hu et al present a series of analyses to estimate the similarity in genetic effect size estimates across diverse genetic ancestries. Overall, I find the work to be of interest with a few nice contributions - particularly the ancestry components and the careful analysis of the predictive performance of PRS condition on a local ancestry model.

The paper can, in a sense, be broken into three pieces:

- 1) The development and application of 'ancestry components'
- 2) The incorporation of those components into genetic association models
- 3) Analysis of PRS performance across individuals with different fractions of genetic ancestries

Overall, I think the paper spends a bit more time than necessary in the main section about the population genetic inference pieces (1 and 2 from above), at least, in terms of the main message, which I think is about the nature of the similarity in genetic effects.

I would also call attention to how often the term 'population' is used and how that means potentially different things at different places in the paper specifically:

-Populations as reflecting genetic ancestries and groups e.g., (at the end of the Abstract - African populations are invoked, where I think African genetic ancestries are meant;)

-Populations as the groups output from the ancestry inference pipeline - indeed the author acknowledge that there is no clear answer to these approaches, describing this work as "an art rather than a science"

-Population as a subsample of the UKB cohort as in "As expected, we observe a strong decline in both ΔR^2 and β with increasing genetic distance from the discovery population of British-ancestry individuals"

It might aid clarity and the somewhat subjective nature of the genetic ancestry solution chosen for analyses to describe it as something other than 'population.'

For section 3 - the analyses presented are to assess the similarity in the effects across genetic ancestries.

First and foremost the notation of the analyses are difficult to follow and I would strongly encourage the inclusion of the regression models for the analyses in the main text that generate: β_{Eu} , $\beta_{Obs.E}$, β_{Af} , as well as the Beta (global + local) models. Right now this only described in words and it's quite difficult to get a clear picture of what individuals are included in each analysis and what regression coefficients are being compared.

I think I have pieced what is being performed and in essence the primary result is that the PGS performance of European ancestry chunks perform consistently across individuals regardless of the degree of recent genetic admixture. Further, most traits seem to show a similar pattern, suggesting that this may well be a more general property. This results is quite a strong observation in support of the relatively minimal impact of GxG, given the differences in frequencies of common variants

observed across many genetic ancestries.

They also appear to argue that the correlation in effect estimates is consistent across genetic ancestries - which, at least to my read, is not clear that this hypothesis is actually being tested - the consistency of the attenuation ratio of PRS trained in European ancestries and applied to African ancestry tracts across different mixing proportions of African ancestry does not actually guarantee that there isn't some scalar difference on the betas - nor does it necessarily rule out that there are some specific genetic effects to either genetic ancestry which would drop out of the estimator [having been missed all together].

Relatedly, the claims about GxE are also predicated on a comparatively similar environment - at least contrasted with all environments everywhere - the authors acknowledge this, but do not soften claims about the strong expectation that the genetic effects will be the same across genetic ancestries. Again - I think this is supported, but I do not think it is proven that this isn't operating especially as any notion of power to detect such differences are largely absent from the work performed.

Minor points

The statement: "Thus at this level, our pipeline is able to capture geographic and ethnicity information" - seems somewhat tautological - at least insofar as I understand the paper - the genetic ancestry space is calculated and then geographic labels are used to refine and update genetic clusters - meaning that the geographic information is used to facilitate the solution [and so it should be correlated].

In the description of the hay fever example - migration is an alternate explanation to natural selection - i.e., individuals for whom hay fever was more bothersome in south England might have been more motivated to migrate.

The phrase 'true genetic ancestry' is unknown - it is unclear on what is meant by true ancestry here.

Similarly statements such as "minority of countries (e.g. Philippines, Japan) are inferred to be relatively genetically homogeneous, with most inferred as complex mixtures." are almost certainly a reflection of sample size in the UKB rather than the absence of fine-scale structure in a group of people living in e.g., Japan or the Philippines.

Reviewer #3:

None

Version 1:

Decision Letter:

30th Apr 2024

Dear Simon,

Firstly, my sincere apologies on behalf of the journal for the very prolonged period in review - I am glad to say that the wait was a constructive one, in terms of the reviews returned.

Your Article, "Fine-scale population structure and conservation in genetic effect sizes between human populations across a range of human phenotypes" has now been seen by the original 2 referees. You will see from their comments below that while they find your work has improved and they seem almost satisfied and thus supportive of publication, there are still a few important points remaining. We continue to be interested in the possibility of publishing your study in Nature Genetics, but would like to consider your response to these concerns in the form of a revised manuscript before we make a final decision on publication.

In brief, the reviewers ask for further minor revisions, potentially including further analysis. We think these requests are reasonable and achievable without undue further effort, and we hope you will be able to fulfil them.

To guide the scope of the revisions, the editors discuss the referee reports in detail within the team, including with the chief editor, with a view to identifying key priorities that should be addressed in revision and sometimes overruling referee requests that are deemed beyond the scope of the current study. We hope that you will find the prioritized set of referee points to be useful when revising your study. Please do not hesitate to get in touch if you would like to discuss these issues further.

We therefore invite you to revise your manuscript taking into account all reviewer and editor comments. Please highlight all changes in the manuscript text file. At this stage we will need you to upload a copy of the manuscript in MS Word .docx or similar editable format.

*2) If you have not done so already please begin to revise your manuscript so that it conforms to our Article format instructions, available

[here](http://www.nature.com/ng/authors/article_types/index.html).

*3) Include a revised version of any required Reporting Summary: <https://www.nature.com/documents/nr-reporting-summary.pdf>

Link Redacted

We hope to receive your revised manuscript within four to eight weeks. If you cannot send it within this time, please let us know.

Nature Genetics is committed to improving transparency in authorship. As part of our efforts in this direction, we are now requesting that all authors identified as 'corresponding author' on published papers create and link their Open Researcher and Contributor Identifier (ORCID) with their account on the Manuscript Tracking System (MTS), prior to acceptance. ORCID helps the scientific community achieve unambiguous attribution of all scholarly contributions. You can create and link your ORCID from the home page of the MTS by clicking on 'Modify my Springer Nature account'. For more information please visit please visit www.springernature.com/orcid.

Sincerely,

Michael Fletcher, PhD
Senior Editor, Nature Genetics
ORCID: 0000-0003-1589-7087

Reviewers' Comments:

Reviewer #1:

Remarks to the Author:

I appreciate the authors' thorough response to my initial comments. I have no major remaining concerns; nevertheless, I list a few minor comments and suggestions below:

- I understand that the authors changed the title of the paper in response to my comment, but I'm not sure it reads well now.

- It would be desirable if the authors could make the individual-level ACs available through UKB, allowing other UKB researchers using the data to apply them in their analyses.

- I remain unsure about how the portability of PGS benefits from the use of ACs instead of PCs. Isn't the PGS model itself based on GWAS with PC correction? Does it make a difference if ACs were used for correction instead? If not, does this suggest that confounding is not a factor contributing to the decrease in portability, which on surface seems contradictory to the findings in Figure 3 indicating confounding as an issue for some traits? I suspect confounding will be an issue for traits like educational attainment? I guess "regional traits" have low heritability and are thus excluded from the PGS analyses?

- I appreciate the new version of Supplementary Note 2. However, it could benefit from providing some "verbal" intuition as to "why" the parameters B_A and B_E become dependent on ancestry proportions without mean-centering. Is the difference in

MAF between European and African groups the primary reason? Does it matter that ascertainment is done in Europeans, resulting in selected variants having high MAF in Europeans? Would this be an issue if the true causal effects were hypothetically known and used in the PGS, i.e., in the absence of LD issues?

- Both "focusing" and "focussing" appear in the text now. Additionally, there are still some typos, such as "fitted groups. and contrast this" in line 226.

Reviewer #2:

Remarks to the Author:

Overall, the authors have substantially improved the manuscript and broadly speaking have addressed the critiques.

I only have a few minor points which I would encourage the authors to consider:

1) Concerning the apparent heterogeneity in FEV - it might be worth investigating whether smoking associated genetic loci are driving some of the apparent heterogeneity in effect sizes. If there are differences in the rates of smoking across genetic ancestries then this would translate to different effect sizes of genetic influences on smoking which might be observable by inspecting the top loci for smoking.

2) The granularity and interpretation of ancestry components surely has some degree of sample size dependence [e.g., the granularity apparent in the UK ancestry components in contrast to say the Afghani ones is a consequence of the relative sample sizes]. Might the authors consider including some statement about how such granularity might be achieved with larger cohorts globally and that the number of 'ancestry groups,' reflects the sample and larger samples would yield a different set of solutions.

3) One other test for claim of better control of stratification is to investigate whether the within sib predictions from the AC-corrected GWAS outperforms the PC-corrected GWAS. While I don't think this is essential, if it is straightforward to test it might strengthen the claims that the control of stratification is superior.

Version 2:

Decision Letter:

Our ref: NG-A63253R1

28th Jun 2024

Dear Simon,

Thank you for submitting your revised manuscript "Fine-scale population structure and widespread conservation in genetic effect sizes between human groups across traits" (NG-A63253R1).

We have made an editorial check of your revisions in response to the last round of review and we are satisfied such that these changes do not require further peer review, and therefore we'll be happy in principle to publish it in Nature Genetics, pending minor revisions to satisfy any final requests and to comply with our editorial and formatting guidelines.

Sincerely,

Michael Fletcher, PhD
Senior Editor, Nature Genetics
ORCID: 0000-0003-1589-7087

We would like to thank both reviewers and the editor for their very careful and constructive criticism of the manuscript. Through considerable additional analysis and rewriting, we have been able to fully address the concerns raised and improve the manuscript as a result. A point-by-point response is below.

Reviewers' Comments:

Reviewer #1:

Remarks to the Author:

Hu et al. developed an ancestry inference pipeline to break down an individual's genome into ancestry components (ACs) using phasing and chromosome painting with carefully constructed reference panels. These ACs serve as powerful tools for investigating the interplay between genetic ancestry, geography, and self-identified ancestry. They also outperform principal components (PCs) in capturing fine-scale population structure, thereby aiding in mitigating confounding in genome-wide association studies (GWAS). The study also examined the portability of polygenic scores (PGS) in 8,000 admixed individuals, decomposing PGS into African-like (APGS) and European-like (EPGS) segments. Their findings suggest that the performance of APGS and EPGS is independent of global ancestry. Based on these findings, they argue that causal effect sizes are likely similar across ancestries.

The paper appears to consist of two somewhat disjointed projects. The first part involves inferring fine-scale population structure and ancestry components (ACs) (Figures 1-3), which I believe constitutes a significant contribution to the field. The second part focuses on polygenic score (PGS) performance in admixed samples (Figure 4). These two parts may not seamlessly integrate into a single manuscript. I appreciate that ACs do play a role in the PGS analysis, both in the GWAS step and in identifying admixed samples. However, as the authors themselves stated, ACs and principal components (PCs) yield similar results in this context. Also, defining broadly African- and European-like segments may not heavily rely on inferring fine-scale population structure. Given these considerations, I find the manuscript title inaccurate, as I do not believe the authors "leveraged" fine-scale population structure in their PGS analyses. Additionally, as I outline below, I think the second part is less clear and convincing than the first part; therefore, the predominant emphasis of the paper on the PGS results, as presented, overshadows ACs, which I consider a significant—and in my subjective opinion, the main—contribution of the paper.

I will split my major comments into two sections to address Figures 1-3 and Figure 4 separately.

We thank the reviewer for highlighting how ACs are a significant contribution to the field from our manuscript. To better reflect what the paper contains, we've changed the title of the manuscript to "**Fine-scale population structure and conservation in genetic effect sizes between human populations across a range of human phenotypes**". We believe the AC section is an essential precursor to the PGS analyses, given population stratification is hypothesised as a driver of the lack of PGS portability across populations. In addition, we have significantly improved the PGS part of the manuscript by adding substantial new analysis and thoroughly refining the text to address the concerns of the reviewers.

Please find our point by point responses below.

Major comments:

Related to Figures 1-3:

- The labels for the 127 reference populations do not seem to be listed anywhere (unless I have overlooked them). It would be ideal to display these labels on the geographic maps, as shown in Figure 2. This would facilitate the interpretation of the results.

Our updated Supplementary table 1 included names for all of the 127 reference populations. We also thank the reviewer for the suggestion of labelling maps, and we have done this, for example in the new Supplementary Fig 2. We have updated legends accordingly.

Supplementary Fig.2. World ancestries in the UKB inferred by the ancestry pipeline. UKB participants are mapped to their self-reported country of birth. World countries are coloured by different colours if they are present in the pipeline a). Ancestries in Central Asia; b). Ancestries in Europe; c). Ancestries in Africa; d). Ancestries in the Americas. White lines on the map delineate the borders of different countries.

- In Figure 2b, the colored regions are not explicitly delineated; it's evident that they do not align with country borders but are associated (via white lines) with bar plots that are separated by countries.

This is right – necessarily not all populations (using external reference data) align precisely with country boundaries, but information on Biobank individuals is country-of-birth. To address this point, we updated Figure 2b and Supplementary Figure 2 by adding country borders (white lines), while colours remain informative for reference groups. We updated the figure legends to now say “White lines on the map delineate the borders of different countries.”

Figure 2. a) ... b) As a, but showing decomposition for Asian, Oceanian and selected east African countries. Colours are as shown on the map, with colours for ancestry from additional regions given in the legend. White lines on the map delineate the borders of different countries. Colour legends are different for panels a) and b).

- In Figures 3c-e, the logical use of previously detected associations (coloring of points) appears inconsistent: in 3d, previous associations are used as evidence for true positives, while in 3c, numerous variants claimed to be false positives are labeled with previous associations.

We agree that this may have been confusing to readers, and have made modifications, added explanations and performed additional analyses to avoid this. The essential point is that 3c was chosen as a “null trait” not expected to show associations. Firstly, to clarify that Figure panels 3c-e share the same colouring of points throughout, we have now moved the legend underneath all three panels of Fig3c-e. We edited the main text and figure caption to add additional explanation. In particular, a new paragraph discussing the prior GWAS data – for the figure 3c trait of employment score – reads:

Figure 3. Comparison between AC-corrected and PC-corrected GWAS. ... c-e) Comparison of AC-corrected (x-axis) and PC-corrected (y-axis) $-\log_{10}(\text{p-values})$ in three exemplar GWAS for labelled traits; all plots are coloured according to the legend shown at bottom, indicating prior evidence from GWAS for each SNP in particular phenotypic categories (grey SNPs show no prior GWAS evidence, perhaps consistent with likely false positive associations). In each plot, the points show only independent SNPs showing $p < 5 \times 10^{-8}$ for one or both approaches.

“We saw strong enrichment within the AC-based associations versus the PC-only associations of prior GWAS signals (Supplementary Table 4: $\text{OR}=11$, $p=0.0013$). Indeed, the PC-only associations showed no significantly greater overlap (18%) with previous GWAS signals compared to the null trait of latitude ($\text{OR}=1.6$; $P=0.44$), but 71% of AC-based associations overlap GWAS signals ($\text{OR}=18$; $p=1.8 \times 10^{-10}$ vs latitude). Moreover, when we examined types of trait showing association, many AC-based signals were significant in previous individual-based GWAS for educational attainment or socioeconomic status (Fig.3d, Supplementary Table 3c). These trait classes were further enriched relative to PC-only hits ($\text{OR}=16$, $p=0.005$) which rarely show such evidence, suggesting PC-only hits are likely false positives (Fig.3d).”

This provides more neutral discussion, and formalised testing - showing there is statistical evidence that prior GWAS evidence is enriched in AC hits vs PC-only hits, and for a “real” trait vs. a null trait, and explaining which prior GWAS studies show overlap. We believe this addresses the concern that there are still some GWAS hits overlapping the PC-only hits for this trait, while remaining neutral about the interpretation of such evidence. We believe the strong odds ratio of 18 now better demonstrates the fact that there is a very large difference in properties of hits supported by ACs in Figure 3d, vs. hits for latitude (i.e. Figure 3c). We also believe it is now more clear that we consistently view prior GWAS evidence as being – broadly - supportive of associations, perhaps, being valid.

- The arguments supporting selection on the hay fever variants in Figure 3c appear speculative. In general, if the geographical distribution of a particular variant does not align with ancestry for any reason, ACs may not effectively account for confounding in a birthplace GWAS. For instance, the distribution of the hay fever variant might reflect proximity to urban versus rural areas (e.g., if individuals experiencing more severe allergic reactions choose to reside in regions with lower allergen exposure), rather than being attributed to selection.

We agree that apart from selection, there are other explanations that hay fever variants in figure 3c shows regional difference, and unusual migration patterns are one of the plausible factors that driving the differentiation. We therefore integrated the suggestions from both reviewers and revised the text to give a more balanced explanation on the hay fever variants. We deleted the sentence “The attendant signal suggests that this selection (of migration or varying reproductive success depending on allele carried) has continued until recent times” as too speculative, and rewrote the text as follows:

“While any residual GWAS hits associated with birthplace may reflect inadequate adjustment for stratification, the few association signals identified using AC’s show a modest enrichment in SNPs showing prior GWAS evidence (Supplementary Table 4: OR=5.9, p=0.039), compared to the PC-only associations, only 12% of which have prior GWAS evidence. Specifically, the strongest remaining signal ($p < 10^{-15}$) after using ACs is for a region of chromosome 4 around rs5743618 that contains several toll-like-receptor genes, including TLR1 and TLR10 which play a role in innate immunity^{40,41}. This is one of the strongest GWAS hits in the genome for hay-fever in European cohorts^{42,43}, with weaker effects on asthma risk. Because the protective allele against hayfever is more common in south England where hay-fever is most prevalent (Supplementary Fig.8), the geographic differentiation⁴⁴ of this variant might reflect selective migration of people carrying this variant and/or past natural selection.”

- It would be beneficial for the scientific community if the authors could provide summary statistics for GWAS with AC corrections for the traits they have analyzed, if not already done.

We agree – we’ve therefore harmonised the AC-corrected and PC-corrected GWAS summary data, and uploaded the data to the GWAS catalog. We also update the “Data availability” section accordingly:

“...GWAS summary statistics files with population structure corrected by ACs and PCs are available in GWAS catalog (<https://www.ebi.ac.uk/gwas/>, GCP ID GCP000799).”

Related to Figure 4:

- The authors' rationale for mean-centering genotypes by local ancestry is inadequately explained or justified, even in the supplementary note. This is a crucial aspect, especially given that the authors assert that their results are sensitive to this procedure, and indeed, they do observe differences in effects with non-centered genotypes. Specifically, there is no apparent reason to believe that equation (3) in the supplementary note 2 is the correct generative model. As formulated, the interpretation of regression coefficients is challenging to comprehend. For example, under the assumption of no interactions and no LD differences across populations, equation (3) models the effect of the genotype as dependent on local ancestry, which is evidently incorrect. Seemingly, the only reason mean-centering was performed was that not doing so yields different results. The authors should provide clear statistical and biological reasoning as to why they made this choice. If they have empirically determined in simulations that mean-centering is indeed necessary, further investigation is warranted to gain a deeper understanding of this phenomenon.

We agree with the reviewer that the rationale of mean-centring genotypes by local ancestry was in need of better clarification, as it is a crucial part of the approach. We have made a series of changes, in particular to Supplementary Note 2, to address these comments.

- 1) On model definitions, first, we apologise that the original text was not sufficiently clear that the actual generating function in supplementary Note 2 was equation (2) but not (3). In the new version (many equations are renumbered due to additional sections) we write, in the first main section:

“To begin to derive the ANCHOR procedure, for our “true” underlying model, we consider the setting where our phenotype...[equation 2]...we assume this form for the true behaviour throughout this Note. Our null model of shared effects is then simply that the same terms γ_j apply identically across individuals, whatever their admixture proportions and whatever their population of origin, though the other terms may vary. The alternative is that γ_j values might differ across populations, or groups of different ancestry proportions.”

We also added an “Overview of approach” section to the Note, to make the Note easier to navigate for readers. In the overview section, we now make the issue of centring immediately clear: “We will also show (and demonstrate using simulations) that there is a “correct” way to construct *EPGS* and *APGS*, and mean-centring genotypes conditional on local ancestry, while other constructions cause biases.”

Following the models we actually fit with ANCHOR, to make clear the difference between the true (but unknown) underlying model and the one we can fit using the estimated PGS from a GWAS, we write:

“Thus, this model can be fit from real data, provided *EPGS* and *APGS* can be obtained. We wish to use this model to approximate the true underlying model (1)/(4). Properties of the resulting coefficients β_A and β_E and their estimators are the key to the ANCHOR approach.”

In several other places – in particular the start of the section titled “**Populations incorporating admixed individuals**”, we further highlight the distinctions between different models.

- 2) On the mean-centring aspect of ANCHOR, we have introduced two new/rewritten subsections in the Supplementary Note 2, entitled “**The impact of centring conditional on local ancestry**” and “**Biases using non-centred polygenic scores**”. These avoid the difficulty in the prior version of understanding exactly why mean-centring matters. It is not the case mean-centring was performed because not doing so yields different results (!) In theory (and in practice – see point 3) it is necessary to perform such centring, or the results of ANCHOR become biased. This is indeed an intuitively surprising fact, and in our initial implementation of ANCHOR we ourselves did not initially understand why the answers were sensitive to centring – it is the theoretical derivation that identifies which approach is “correct”. Full details are in the new Supplement, but briefly in the first subsection we note “there are several natural alternative approaches, and it is not obvious yet which is better.” We therefore define different natural approaches to constructing PGS, without local-ancestry-dependent mean-centring, and show that they are each different from the ANCHOR approach, but equivalent to each other. We show

that we can compare approaches using a common form they all share, but with different coefficients π_j and λ_j , which are zero (only) if mean-centring is used. In **“Estimates of β_A and β_E in the ANCHOR approach”**, we show that when π_j and λ_j are both zero, ANCHOR can estimate common values for these parameters if $\rho=1$, regardless of ancestry composition of the sample. However, in the new section and **“Biases using non-centred polygenic scores”** we show that if mean-centring is *not* used, we can obtain estimates of β_A and β_E that depend on local ancestry, a bias suggesting that effect sizes vary between groups even in the case where $\rho=1$ truly. This occurs because **without mean centring the fitted EPGS and APGS contain terms that depend on the allele frequency differences between populations, multiplied by local ancestry**. Because local ancestry varies between samples, in fitting say β_E we end up partially fitting these terms – biasing things, especially if most of the genome is from population **A**. The problem with fitting these terms is that their optimal slope depends on frequencies of causal variants in population **A**, which are largely unknown due to tagging – if a GWAS is performed in population **E**, it may not tag population **A** variants well.

- 3) To support the theoretical findings and show centring is important in practice, we reanalysed our simulated data under the null case of complete correlation ($\rho=1$), exactly as previously except this time *without* performing mean-centring. This shows directly that biased results occur in this case across 24 simulated traits. The $\beta_E/\beta_{Obs.Eu}$ estimated from the non-mean-centred model fitting is reduced relative to mean-centring across each of the 24 simulated traits, sometimes significantly so. The combined estimate of ρ becomes strongly significantly less than 1 ($p<0.001$), implying a significant downward bias. The bias is eliminated by centring genotypes, whether or not we filter regions of uncertain ancestry, and is actually worse without centring when such filtering is performed. We present these results in the revised manuscript (Supplementary Fig.15-17; Supplementary Note 2).

Supplementary Fig.15. Comparison of combined estimate of ρ for 24 simulated traits from PGSs constructed in combination of with/without mean-centring and with/without masking the uncertain and short segments, given true $\rho=1$ under the null model. Dots separated by vertical line indicate different ratios of estimate: left panel: $\beta_{Eu}^{All} / \beta_{Obs.Eu}$ and right panel: $\beta_{Eu}^{Af} / \beta_{Obs.Eu}$. Horizontal line represents true setting of ρ . If confidence interval for each ratio of estimate intersects with $\rho=1$ line, the ratio estimate is accurate. Colour of the dots indicates if mean-centring is applied or not, and shape of the dots indicates if masking uncertain and short segments is applied or not.

Supplementary Fig.16. ANCHOR results for each simulated trait under the null model simulation. a) PGSs are constructed with masking uncertain and short segments and b). PGSs are constructed without masking

uncertain and short segments. Every two columns separated by the vertical green dash line show ratios of estimates defined as in Main Figure 4e. Odd columns show result with PGSs constructed by “mean-centring” and even columns show result with PGSs constructed without “mean-centring”. Rows show simulated phenotypes. The final row shows combined estimates. The first and second two columns estimate the underlying correlation in causal effect sizes between European-ancestry and either all 8003 African-ancestry individuals, or individuals of 100% African-ancestry, respectively, while the third two columns additionally estimate the impact of local effects on predictive power for African-ancestry segments.

Supplementary Fig.17. Results of application of ANCHOR for 24 UKB simulated traits, for individuals of varying African ancestry binned as in main Figure 4c (x-axis; coloured region). For each bin in each panel, estimates of coefficients $\beta_{Eu}/\beta_{Obs.Eu}$ (blue) and $\beta_{Af}/\beta_{Obs.Eu}$ (red) are shown with 95% bootstrapped confidence intervals, representing the ratio of PGS within European and African genomic regions to the PGS obtained from external European samples respectively (Methods). Also shown are these estimates from individuals of ~100% European ($\beta_{Obs.Eu}$; blue horizontal bar) or ~100% African (red horizontal bar) ancestry. Each panel represent one of the combinations of with/without mean-centring and with/without masking the uncertain and short segments for 24 simulated traits under the null model simulation: panels in the top represent PGS constructed with mean centring, and panels in the bottom represent PGS constructed without mean-centring; panels in the left represent PGS constructed with masking the uncertain and short segments and panels in the right represent PGS constructed without masking the uncertain and short segments.

4) More generally, in the supplement we edited throughout for clarity, and added additional subsection headings so it is easier to navigate and identify answers regarding particular aspects of the method.

- The ratio $B_{eu}/B_{obs.eu}$ serves as an estimator of ρ when the causal effect sizes exhibit comparable variance across populations. However, effect size variance may vary across populations due to factors such as assortative mating, the strength of stabilizing selection, or

environmental influences modulating variance. Consequently, $B_{eu}/B_{obs.eu}=1$ does not necessarily imply $\rho=1$.

We agree with the reviewer fully in terms of the technical part of this comment. We extended the discussion in the Supplementary Note covering this point in the section “**Parameter estimation in real samples: variable effect sizes between populations**”. We also edited the main text to now discuss this point more precisely (though for reasons of space we have still tried to be relatively brief):

In the results: “In turn (Supplementary Note 2) this implies either conserved effect sizes between European-ancestry and African-ancestry individuals in UKB, or that – if effect sizes are *not* conserved, they must be systematically larger on average across all these phenotypes in African-ancestry individuals, in such a manner as to coincidentally balance the impact of incomplete correlation.”

In the discussion: “Although it is formally possible (Supplementary Note 2) this could occur if the underlying correlation between effect sizes $\rho < 1$, but this is coincidentally counterbalanced by larger average effect sizes in African-ancestry individuals, this seems unlikely across diverse quantitative traits *a priori*.”

In the somewhat longer technical details in the Supplementary Note 2, we make the following main points. First, for any given trait it is completely correct that the ρ estimate is not always the correlation, if effect size variance differs between populations. However, ρ does always have the interpretation of the relative increase in the trait in a second population, per unit increase in the average genetically predicted value of the trait in a first population. This is a natural measure – for example, of how well a PGS scoring a trait in Europeans could indicate changes in that trait in African-ancestry people. It seems natural to us (perhaps even more so than correlation) as a measure of “usefulness” of genetic findings from one group, in another. Secondly, to see $\rho=1$ in the presence of different effect sizes we need a “cancelling effect” which requires, in essence, luck. In particular, cancelling can occur if the effect size variance is greater in Africans than Europeans because the correlation itself is not above 1. This might well occur for one trait, of course. However, once we average over (now) 53 traits, our confidence intervals become quite tight (0.91–1.05) and our point estimates are extremely close to 1. For this to occur if the true correlation is, say, 0.8 would mean genetic effects of the same variants are on average systematically stronger across all these traits in African-ancestry people, which we think is unlikely.

We think this issue will be partially solved – in future – by analysing larger datasets (giving tighter CIs for single traits) and more traits. (Complementary approaches can also help by directly examining individual effect sizes at individual loci to separate possibilities. They also cannot yet easily be applied widely, as at present they are underpowered due to the small number of completely known causal variants and/or sample sizes.)

- The confidence intervals (CIs) are notably large, especially for mixed ancestry groups. This could be attributed to the fact that the majority of the admixed individuals (approximately 6600 out of 8000) have more than 90% African ancestry. Consequently, I have concerns about

whether the analysis is sufficiently powered to detect deviations from the null of no interactions.

We also wish – of course – some CIs could be made smaller. However, we believe other approaches in the literature obtain, similarly, noticeable levels of uncertainty for individual traits – it is mainly a consequence of the low predictive power of currently constructable PGS. Our estimates are relatively tight for some traits, e.g. height and platelet count, whether these PGS have stronger predictive power, so we believe that their precision will continue to improve in future applications. One advantage of ANCHOR is that we can combine across traits, and (see above comment) this appears to indicate shared effect sizes among traits operate quite widely. We see evidence of deviation from identical effect sizes for some traits – e.g. White blood cell counts in the new analysis, albeit not yet overwhelmingly significant. Actually, the number of individuals with African ancestry less than 90% is 2,681 and the number of individuals with African ancestry less than 95% is almost 4,000. At least if we combine traits, we observe that the ANCHOR estimator $\beta_E/\beta_{Obs.Eu}$ is relatively informative even in samples with high African ancestry (see new Figure 4b-c and Figure 4e). Our simulations use the same number of African-European individuals as the real data (see new Supplementary Fig. 22).

a**b****c****d****e**
Figure 4. Separation of local and non-local factors influencing polygenic score portability. a) Test principles: in UKB samples containing segments of European (blue) and African (red) ancestries, a single causal variant contributing to a trait is captured by a tag SNP within the PGS. The predictive power of this tag SNP (pink arrow thickness) can vary according to local factors varying with “local” ancestry (top vs bottom chromosomes), or non-local factors impacting the effect of the causal SNP itself, quantified by Genome-wide “global” ancestry (left vs. right individuals); ANCHOR separates the contribution of these to PGS portability. b-d. All CIs shown are 95% bootstrapped intervals. β values represent estimated increase in phenotype per unit increase in the PGS for individuals of European ancestry ($\beta_{Obs.Eu}$) and for European (β_{Eu}^{All}) or African (β_{Af}^{All}) ancestry segments in African-ancestry individuals: ratios of these quantities are used in ANCHOR (Methods). Other β_j^i estimates are similar, but refer to local ancestry j and global ancestry i . b) Performance of ANCHOR in simulations of 24 traits (Methods) and 53 UKB quantitative traits with PGSs constructed using different p -value thresholds (p -value=0.05 and p -value=0.0001), and varying true correlation ρ between underlying effect sizes in African-ancestry individuals (x-axis) and European-ancestry individuals shows that ANCHOR-produced values of $\beta_{Eu}^{All}/\beta_{Obs.Eu}$ (y-axis) accurately estimate ρ across five distinct African ancestry bins (colours denote bins, defined as in c). Estimates of $\beta_{Eu}^{All}/\beta_{Obs.Eu}$ by fitting PGSs constructed using p -value=0.05 and p -value=0.0001 for 53 UKB traits are shown in the box on the right c) Application of ANCHOR for 53 UKB traits, for individuals of varying African ancestry binned as shown (x-axis; coloured regions). For each bin, mean estimates across traits of ratios $\beta_{Eu}/\beta_{Obs.Eu}$ (blue) and $\beta_{Af}/\beta_{Obs.Eu}$ (red) are shown (Methods). Also shown are these ratios for individuals of ~100% European (leftmost point at $y=1$) or ~100% African (red horizontal bar) ancestry. CIs crossing the line $y=1$ are consistent with the setting $\rho=1$ where true effect sizes are identical to ~100% European-ancestry individuals, and similarly for red points/bar. d) Mean increase in standing height per unit increase in its PGS (columns) for 7 ancestral groups, alongside local-ancestry-specific effect size estimates from ANCHOR model fitting (final 6 columns; Methods), for global and local ancestry combinations as labelled on the x-axis and above the bars, $\beta_j^{Af}/\beta_j^{Eu}$ refers to local ancestry j and global ancestry of 100% African/100% European, respectively (Methods). e) ANCHOR results by trait. The first and second columns estimate ρ either across “all” 8003 African-ancestry individuals, or for hypothetical projected “Af” individuals of 100% African-ancestry, respectively, while the third column shows greatly reduced predictive power for African-ancestry segments, as expected (Supplementary Note 2). Rows show phenotypes: the first row for standing height uses a previously published PGS¹⁶, the rows highlighted in dark green show combined estimates.

Supplementary Fig.22. Results of application of ANCHOR for 24 UKB simulated traits, for individuals of varying African ancestry binned as in main Figure 4c (x-axis; coloured region). For each bin in each panel, estimates of coefficients $\beta_{Eu}/\beta_{Obs.Eu}$ (blue) and $\beta_{Af}/\beta_{Obs.Eu}$ (red) are shown with 95% bootstrapped confidence intervals, representing the ratio of PGS within European and African genomic regions to the PGS obtained from external European samples respectively (Methods). Also shown are these estimates from individuals of ~100% European ($\beta_{Obs.Eu}$; blue horizontal bar) or ~100% African (red horizontal bar) ancestry. From top to bottom, each panel represents descending correlation ρ . The case $\rho=1$, where only local effects impact portability, corresponds to blue points lying along the blue line, and similarly for red points, as observed. In cases when $\rho < 1$, the dotted horizontal line is plotted at ρ , and the blue points are predicted to lie along this line if $\beta_{Eu}/\beta_{Obs.Eu}$ provides an accurate estimate of ρ , as predicted by theory (Methods; Supplementary Note 2).

- The choice of traits is somewhat limited, and the selected complex traits appear to be somewhat correlated. It might be worthwhile to expand the analysis to include more traits,

especially those that are potentially more influenced by interactions, such as behavioral traits like neuroticism scores.

We thank the reviewer for this suggestion, and we have followed it by lowered the threshold of heritability from 10% to 5%. This allowed us to expand the number of traits we analyse from 29 to 53, including neuroticism score and time spent watching TV. Although more uncertain in general, a combined analysis of these additional traits gave results very similar to our initial results (new Figure 4b-c and Figure 4e). We added a summary of these results into the manuscript (Figure 4e) and complete results in Supplementary Figure S24.

Supplementary Fig.24. ANCHOR results for all the 53 UKB traits. Every two columns separated by the vertical green dash line show ratios of estimates defined as in Main Figure 4e. Odd columns show result with PGSs constructed by using default p-value 0.05 in thresholding and even columns show result with PGSs constructed by using alternative p-value 0.0001 in thresholding. Rows show each of 53 UKB phenotypes. The final two row show combined estimates. The first and second two columns estimate the underlying correlation in causal effect sizes between European-ancestry and either all 8003 African-ancestry individuals, or individuals of 100% African-ancestry, respectively, while the third

two columns additionally estimate the impact of local effects on predictive power for African-ancestry segments. Overall the ratio estimation from two different p-value thresholds in PGS construction are very consistent across most of the traits.

- The PGS model constructed based on pruning and thresholding uses a relatively high p-value threshold of 0.05, which may not be optimal. The noisy effect estimates of weak associations likely hamper the performance of PGS, although I note that they will not bias the analyses.

This is an important thing to test and we agree that high p-value threshold of 0.05 might not be optimal, so we now also performed a new PGS analysis using a p-value threshold of 10^{-4} both in simulation and in UKB traits, to address this concern. We compared performance of PGS using both thresholds and we found our results remain essentially unchanged. We updated Figure 4b,e and created new supplementary figures 21 and 24 to elucidate this.

Supplementary Fig.21: . ANCHOR results for all the 24 simulated traits under null model. Every two columns separated by the vertical green dash line show ratios of estimates defined as in Main Figure 4e. Odd columns show result with PGSs constructed by using default p-value 0.05 in thresholding and even columns show result with PGSs constructed by using alternative p-value 0.0001 in thresholding. Rows show each of 24 simulated phenotypes. The final two row show combined estimates. The first and second two columns estimate the underlying correlation in causal effect sizes between European-ancestry and either all 8003 African-ancestry individuals, or individuals of 100% African-ancestry, respectively, while the third two columns additionally estimate the impact of local effects on predictive power for African-ancestry segments. Overall the ratio estimation from two different p-value thresholds in PGS construction are very consistent across most of the traits.

- The section title "biological effects are highly similar..." is too strong. It is not clear what is meant by "biological effects." The authors likely refer to the "effect of causal variants."

Good point - we agree that the statement is too strong and revised this title as suggested to "Effect sizes of causal variants are highly similar across ancestries for a range of human traits".

- Regarding GxE interactions, it is plausible that environmental effects do not track ancestry in the UK, but this may not apply universally to other regions, such as the US. Thus, statements like "providing evidence that gene-environment and gene-gene interactions do not play major roles in the poor prediction of European-ancestry PRS scores in African populations" are overly general. The paper contains many such blanket statements that are not substantiated by the presented analyses.

We thank the reviewer for pointing out the overly general or blanket statement and we've revised overly general statements throughout the revised manuscript. For this specific statement we added the specifier "for these traits in the UK".

- The figure is generally too challenging to understand.

We've adjusted, and simplified main text figures Fig2-4, and many supplementary figures, and rewritten the figure legends.

Minor comments:

- There are numerous typos in the paper and supplementary notes, such as "focussing" in the abstract.

We've carefully corrected typos in the paper and supplementary notes (this particular word appears to allow both "focusing" and "focussing" as correct spellings)

- Figure captions are generally uninformative.

We've rewritten the figure captions to make them clearer.

- The use of the Beta symbol for both GWAS effects and the regression coefficient for PGS is confusing.

That's true - we've changed mathematical notation to avoid this clash. In the revised manuscript and supplementary note, the GWAS effect sizes are now represented by γ , while we reserve the Beta symbol to mean regression coefficients for PGS only.

- I presume by "Bangladesh" in line 211 the authors mean "Gujarat".

Thank you for catching this – we now use the word "Gujarat" in line 211.

- Regarding ACs predicting PCs and vice versa, the text could clarify that predictions are based on "linear" combinations of ACs or PCs.

We have now clarified that the predictions are based on linear combination of ACs or PCs. In caption for main Fig. 3, we added this clarification:

Figure 3. Comparison between AC-corrected and PC-corrected GWAS. a-b). Predictions are based on "linear" combinations of ACs or PCs. a)

- Given that the true positives and negatives are unknown in Figures 3c-e, the language should always be conservative, using terms like "likely" false positives or "likely" false negatives.

We agree and have changed the language by using terms "likely false positives" and "likely false negatives" throughout the revised manuscript; see above response for other changes we made relating to this figure.

- Clarify the unit of the y-axis in Figure 4c. I assume that phenotypic values are normalized to enable the averaging of PGS across traits.

In the revised manuscript, we've changed the entire y-axis from the phenotypic values to ratio estimate $\beta_E/\beta_{Obs.Eu}$ so the value can be normalised and averaged across traits in a more meaningful way. The units are therefore naturally the (dimensionless) ratio of $\beta_E/\beta_{Obs.Eu}$

- In the caption of Figure 2, specify that the color legends are different for panels a and b. Additionally, in panel b, the legend for the "ethnicity key" should clarify that these identifications are self-reported.

To address this, in the caption we've specified that "the colour legends are different between panels a and b". In addition, we revised the legend for the "ethnicity key" as suggested.

- Consider presenting the legend outside the plot regions in Figure 3d for clarity. Additionally, it is not apparent whether the shape of the points adds value to the message of the figure.

We thank the reviewer for this suggestion. We've updated Figure 3 by moving the legend out of figure 3d. In addition, we removed the legend denoting the shape of the points, as we agree it does not add much to the key message.

- I suggest avoiding the statement "data not shown", and instead, providing plots or tables to support claims.

We removed two "data not shown" statements in the main manuscript.

For the first one, we provided Supplementary Figure 15-17 to support our claims for the reason of mean-centring.

For the second, we added a citation to Hou et al's recent NG paper (in their study they ruled out the of different relative effect sizes on African and European chromosomes within an individual, which was the issue referred to).

In Supplementary Note2, the statement "data not shown" regarded the consistency between our UKB-based estimates of European/African effect allele frequency, and 1000G direct allele frequencies. We instead include the new Supplementary Fig.27 (in the main text and Supplementary Note 2) to illustrate this directly:

"Given the large sample size, these estimates are expected to closely match the true SNP frequencies for the specific admixing groups (they also correlate strongly with the frequencies of the same variants in 1000G cohort, Supplementary Fig.27)."

Supplementary Fig.27. Comparison of effect allele frequency in 1000G cohort and effect allele frequency estimated by ANCHOR. Left: Effect allele frequency in European (blue dots);right: Effect allele frequency in African (red dots).

Reviewer #2:

Remarks to the Author:

Hu et al review:

Hu et al present a series of analyses to estimate the similarity in genetic effect size estimates across diverse genetic ancestries. Overall, I find the work to be of interest with a few nice contributions - particularly the ancestry components and the careful analysis of the predictive performance of PRS condition on a local ancestry model.

The paper can, in a sense, be broken into three pieces:

- 1) The development and application of ‘ancestry components’
- 2) The incorporation of those components into genetic association models
- 3) Analysis of PRS performance across individuals with different fractions of genetic ancestries

Overall, I think the paper spends a bit more time than necessary in the main section about the population genetic inference pieces (1 and 2 from above), at least, in terms of the main message, which I think is about the nature of the similarity in genetic effects.

We thank the reviewer for their valuable comments of highlighting the nature of the similarity in genetic effects. Given the first reviewer found the ancestry part (1-2) most valuable, we hope that our thorough revision addressing all reviewers’ comments will better illustrate the value of both the genetic ancestry (1-2) and PGS sections (3).

I would also call attention to how often the term ‘population’ is used and how that means potentially different things at different places in the paper specifically:

-Populations as reflecting genetic ancestries and groups e.g., (at the end of the Abstract — African populations are invoked, where I think African genetic ancestries are meant;)

-Populations as the groups output from the ancestry inference pipeline— - indeed the author acknowledge that there is no clear answer to these approaches, describing this work as “an art rather than a science”

-Population as a subsample of the UKB cohort as in “As expected, we observe a strong decline in both ΔR^2 and β with increasing genetic distance from the discovery population of British-ancestry individuals”

It might aid clarity and the somewhat subjective nature of the genetic ancestry solution chosen for analyses to describe it as something other than ‘population.’

This is an important suggestion. We agree using the “population” in different contexts may confuse the audience, so we have revised the text so that we refer to the genetic ancestry solution as “genetic clusters” or “groups”. Further, we now describe GWAS samples as “cohorts” rather than “populations”.

For section 3— - the analyses presented are to assess the similarity in the effects across genetic ancestries.

First and foremost the notation of the analyses are difficult to follow and I would strongly encourage the inclusion of the regression models for the analyses in the main text that generate:

β_{Eu} , $\beta_{Obs.E}$, β_{Af} , as well as the Beta (global + local) models. Right now this only described in words and it’s quite difficult to get a clear picture of what individuals are included in each analysis and what regression coefficients are being compared.

To try to make the paper easier to follow, we've changed our mathematical notations to avoid these confusing symbols. In the revised main manuscript and supplementary note, the β is used to represent the coefficient of polygenic score, whereas γ is used to represent the GWAS effect size. We have rewritten Supplementary Note 2 to better explain various details of the approach, and in the main text section "**Effect sizes of causal variants are highly similar across ancestries for a range of human traits**", we now introduce these parameters in (we hope) a more intuitive manner, especially addressing which coefficients are being compared: "Consider a true (unknown) underlying PGS operating in European-ancestry UKB individuals, such that each unit increase in the PGS yields a unit increase in the trait in European ancestry segments. We define ρ as the average trait increase observed instead in European segments of admixed individuals. Under reasonable assumptions (Supplementary Note 2), ρ is equal to the ratio $\beta_{Eu}/\beta_{Obs.Eu}$ and ANCHOR provides estimates of ρ by estimating this ratio..."

(this section is more broadly rewritten for clarity; see revised text)

I think I have pieced what is being performed and in essence the primary result is that the PGS performance of European ancestry chunks perform consistently across individuals regardless of the degree of recent genetic admixture. Further, most traits seem to show a similar pattern, suggesting that this may well be a more general property. This results is quite a strong observation in support of the relatively minimal impact of GxG, given the differences in frequencies of common variants observed across many genetic ancestries.

They also appear to argue that the correlation in effect estimates is consistent across genetic ancestries— - which, at least to my read, is not clear that this hypothesis is actually being tested— - the consistency of the attenuation ratio of PRS trained in European ancestries and applied to African ancestry tracts across different mixing proportions of African ancestry does not actually guarantee that there isn't some scalar difference on the betas— - nor does it necessarily rule out that there are some specific genetic effects to either genetic ancestry which would drop out of the estimator [having been missed all together].

This is indeed what we see – PGS performance of European ancestry chunks is similar in European, compared to almost 100% African, genetic backgrounds, across phenotypes. On the interpretation, reviewer 1 made an overlapping point and see the detailed response above. Briefly, in the revised manuscript we broadened the validity of this finding to additional phenotypes and PGS approaches, but we also improved our description of the meaning of this result. It is possible to have a scalar transformation of effect sizes, but only if it is accompanied by incomplete correlation between groups in effect sizes – so it has to be an *increase* in effect size, on average in just the right (in-) convenient way to cancel the incomplete correlation. We think this is probably (very?) unlikely looking across many phenotypes, but it is now more clearly stated as a possibility. Regardless of this interpretation, the ratio we estimate has the concrete interpretation that a unit of increase in a PRS in European corresponds to a proportional increase close to 1 of increase in a PRS in Africans, and this is of direct relevance for theoretical PGS transferability.

Relatedly, the claims about GxE are also predicated on a comparatively similar environment - at least contrasted with all environments everywhere - the authors acknowledge this, but do not soften claims about the strong expectation that the genetic effects will be the same across

genetic ancestries. Again - I think this is supported, but I do not think it is proven that this isn't operating especially as any notion of power to detect such differences are largely absent from the work performed.

We want the message to be proportionate to the evidence and agree with your phrasing. To address this, we have softened the claims to highlight that the only interactions we can rule out are "ancestry-specific". We therefore say "If instead non-local ancestry-specific (gene-gene or gene-environment) interaction effects do occur, then we expect $\rho \neq 1$, and if the variance of underlying genetic effect sizes – which measures the average strength of genetic effects genomewide – is the same in each group (Supplementary Note 2), then $\rho < 1$ (Supplementary Note 2). Therefore, testing for $\rho \neq 1$ provides a test for differing effect sizes across groups." which feeds into the nuanced conclusion "Our results do not imply that gene-environment, or even gene-gene, interactions are not operating across the quantitative traits we investigated here, or other traits. Interactions still likely operate to cause variation in effect size across individuals with African-ancestry. Our results though imply these must largely be shared with other groups of different ancestries, so as to not produce differences in overall (mean) effect sizes across populations."

Minor points

The statement: "Thus at this level, our pipeline is able to capture geographic and ethnicity information" - seems somewhat tautological - at least insofar as I understand the paper - the genetic ancestry space is calculated and then geographic labels are used to refine and update genetic clusters - meaning that the geographic information is used to facilitate the solution [and so it should be correlated].

As the reviewer notes, indeed the geographic and ethnic labels are used to refine the genetic clustering of the individuals contained in the reference set. However, here we are referring to our inferred ancestry in the UKB individuals, whom were not included in the painting panel reference and hence not included in the genetic clustering. Despite this, as we note, their inferred ancestry patterns show a strong correlation with their reported birthplace and ethnicity.

In the description of the hay fever example - migration is an alternate explanation to natural selection - i.e., individuals for whom hay fever was more bothersome in south England might have been more motivated to migrate.

We agree with this, and the other reviewer (reviewer 1) also raised the similar concern. We have revised the manuscript in terms of both reviewers' comments. Please find our detailed response to reviewer 1.

The phrase 'true genetic ancestry' is unknown - it is unclear on what is meant by true ancestry here.

We agree with this and have revised the text to avoid the word "true" with regards to ancestry.

Similarly statements such as "minority of countries (e.g. Philippines, Japan) are inferred to be relatively genetically homogeneous, with most inferred as complex mixtures." are almost certainly a reflection of sample size in the UKB rather than the absence of fine-scale structure in a group of people living in e.g., Japan or the Philippines.

We agree with this, and have revised the manuscript: "While some countries (e.g. Philippines, Japan) are inferred to be relatively genetically homogeneous, perhaps due to small sample sizes, multiple inferred ancestry patterns are observed across people within most countries."

Reviewer #3:

None

We would like to thank both reviewers and the editor for their further comments to help us improve our paper. A point-by-point response is below.

Reviewers' Comments:

Reviewer #1:

Remarks to the Author:

I appreciate the authors' thorough response to my initial comments. I have no major remaining concerns; nevertheless, I list a few minor comments and suggestions below:

- I understand that the authors changed the title of the paper in response to my comment, but I'm not sure it reads well now.

We agree and (re)changed the title to "Fine-scale population structure and widespread conservation in genetic effect sizes between human groups across traits".

- It would be desirable if the authors could make the individual-level ACs available through UKB, allowing other UKB researchers using the data to apply them in their analyses.

We already returned the ACs to the UKB in August, 2023 – it appears there is a backlog in releasing UKB returns and we have pursued this (several times) with the UKB access team to ensure release occurs as quickly as possible, though restrictions prevent our self-releasing these data.

- I remain unsure about how the portability of PGS benefits from the use of ACs instead of PCs. Isn't the PGS model itself based on GWAS with PC correction? Does it make a difference if ACs were used for correction instead? If not, does this suggest that confounding is not a factor contributing to the decrease in portability, which on surface seems contradictory to the findings in Figure 3 indicating confounding as an issue for some traits? I suspect confounding will be an issue for traits like educational attainment? I guess "regional traits" have low heritability and are thus excluded from the PGS analyses?

At the outset we wanted to test whether cryptic population structure might contribute to lack of portability, something we test in the paper (Supplementary Fig. 10 and Supplementary Table 6). For traits highlighted in Figure 3, we saw differences in association study results for a subset of hits; however for many traits, results were similar. For polygenic scores, AC and PC corrected GWAS indeed yield different results for some traits in Figure 3 (e.g. birthplace). However, we agree that our findings support the idea that AC's reduce bias (and they lead to quite different mean values in different populations to PC's), but that cryptic structure does not seem to be a major cause of non-portability between groups. However, for completeness we have now performed a new analysis using PCs (rather than AC's) for ANCHOR. Results are essentially identical to before (Supplementary Figure 25). Indeed, regional traits are excluded for heritability reasons. We agree educational attainment might show unusual confounding and would recommend (perhaps with additional measures) AC's for this case.

We slightly revised the main text accordingly:

“... The joint bootstrap yielded an overall estimate $\rho = 0.98 \pm 0.07$ for these traits, extremely close to 1. Results were also similar using varying p -value thresholds (p -value 0.05 and 0.0001) from the initial GWAS for SNPs to be included in the PGS, when correcting using AC's or PC's in this initial GWAS, and for Standing height showed little change using a previously published alternative PGS¹⁶ (Fig.4e, Supplementary Fig. 24-5).”

and in the discussion we added a sentence “Meanwhile, consistent estimation of effect size between PGSs constructed by AC-corrected GWAS and PC-corrected GWAS also reveals that population structure is unlikely to be the main cause of lack of portability.”

Finally in Methods we clarified slightly our selection criteria for traits (noting educational attainment is excluded as categorical): “In polygenic score analysis, we selected 53 quantitative traits on which to run GWAS using the following criteria: trait measured on at least 400,000 individuals; LDSC-estimated trait heritability at least 5%; trait must be non-categorical”.

- I appreciate the new version of Supplementary Note 2. However, it could benefit from providing some “verbal” intuition as to “why” the parameters B_A and B_E become dependent on ancestry proportions without mean-centering. Is the difference in MAF between European and African groups the primary reason? Does it matter that ascertainment is done in Europeans, resulting in selected variants having high MAF in Europeans? Would this be an issue if the true causal effects were hypothetically known and used in the PGS, i.e., in the absence of LD issues?

As suggested, we now added additional “verbal” intuition in the revised supplementary note2, as follows:

“Although the explanation is somewhat technical, in brief mean-centring is necessary in practice because three phenomena interact: first, underlying “true” effect sizes operating in the admixed population are not known. Secondly, SNP allele frequencies differ between admixing populations. Thirdly, ancestry varies randomly along the genome in admixed individuals. If any of those three were not true, mean-centring would not be needed, but in practice all occur, in almost all real-world applications. When they do occur, the average value of the phenotype estimate from the PGS when applied to admixed individuals depends on their random local ancestry, and mean SNP frequencies in E and A . Because effect sizes are only estimates, this adds additional ancestry-dependent noise to phenotype prediction, and we show below this noise biases the resulting the $EPGS$ and $APGS$ coefficients, in theory and in practice.”

We have struggled somewhat to offer a completely simple intuition, because mean-centring is necessary in the real world only because each of above three phenomena interact. If any of those 3 were not true, it would not be needed! (LD of course enters the picture when our effect sizes identify the wrong SNPs, contributing to misestimated effect sizes).

For example to answer your last question, if hypothetically the true causal effect sizes were available and used in the PGS, and if they are identical in the admixed and non-admixed groups, mean-centring is not needed. This is mentioned in Supplementary Note 2: “There are a few cases where mean-centring has no effect..... if the regression coefficients...are perfectly estimated but we do not expect to be close to this case in practice” (the other case is where ancestry does not actually randomly vary along the genome).

- Both "focusing" and "focussing" appear in the text now. Additionally, there are still some typos, such as "fitted groups. and contrast this" in line 226.

Thanks – now corrected.

Reviewer #2:

Remarks to the Author:

Overall, the authors have substantially improved the manuscript and broadly speaking have addressed the critiques.

I only have a few minor points which I would encourage the authors to consider:

1) Concerning the apparent heterogeneity in FEV - it might be worth investigating whether smoking associated genetic loci are driving some of the apparent heterogeneity in effect sizes. If there are differences in the rates of smoking across genetic ancestries then this would translate to different effect sizes of genetic influences on smoking which might be observable by inspecting the top loci for smoking.

Very interesting comment. In case it is of interest (although we think appropriately not included in the paper, because it relates to only one phenotype), we retrieved smoking GWAS results based on UKB data from Wootton, et al. *Psychol Med* (2020)¹ and constructed PGSs of FEV1/FVC as suggested stratified by smoking association: first, we categorised the smoking loci as weak smoking loci, moderate smoking loci and top loci by apply three different p-value thresholds: 0.05, 0.001 and 5×10^{-8} separately. We next constructed PGSs for FEV1 and FVC by including/excluding the smoking loci (the “weak” smoking loci explain 20-30% of the overall predictive power, the others <5%).

Running ANCHOR after excluding smoking-related loci (below plot, “exclusion”) yields similar – but actually slightly stronger - results to the results by using all the variants, while using only the “smoking” loci (“inclusion”) suggests similar effect sizes in admixed and non-admixed groups (but it is uncertain). So by this exploratory analysis, smoking loci seem unlikely to be the major driving factor for the heterogeneity of effect size for FEV1 and FVC across ancestries.

2) The granularity and interpretation of ancestry components surely has some degree of sample size dependence [e.g., the granularity apparent in the UK ancestry components in contrast to say the Afghani ones is a consequence of the relative sample sizes]. Might the authors consider including some statement about how such granularity might be achieved with larger cohorts globally and that the number of 'ancestry groups,' reflects the sample and larger samples would yield a different set of solutions.

We agree with the reviewer’s comment that the granularity and interpretation of AC has some degree of sample size dependence. We have added a sentence in the discussion:

“We note that the achievable granularity and interpretation of ACs depends on the size and diversity of the ancestry reference panel used, as well as the related number of identifiable pre-defined “ancestry groups”.”

3) One other test for claim of better control of stratification is to investigate whether the within sib predictions from the AC-corrected GWAS outperforms the PC-corrected GWAS. While I don't think this is essential, if it is straightforward to test it might strengthen the claims that the control of stratification is superior.

In Supplementary Figure 6 we tested control of stratification using an alternative approach, based on the LD-score intercept/attenuation, suggesting that stratification is often not a serious problem in UKB, but for traits where issues might occur (with a stronger attenuation), AC’s improve matters, as shown in a couple of cases in Figure 3.

We very much like the idea of using sib-pairs as suggested, to eliminate stratification differences by comparing between sibs, but note that it offers challenges for the “regional” traits we identified as showing the strongest issues. For example birthplace, or regional education/employment score seem quite unlikely to vary between sibs, who are mainly born/educated in the same place. Of the 22,655 full sibling pairs using the method introduced

in Hinke, S & Vitt, N. *Nature communications* (2024)², 935 fall in our “test data”, including those samples not used for original GWAS, which often retained one of each pair. This means power to identify performance differences is likely quite limited unless they are large. Nonetheless, we conducted spot-testing of particular traits, but we did not observe significant differences between AC’s and PC’s. For north/south birthplace, employment score and education score, both AC’s and PC’s achieved predictive power (for predicting inter-sib differences) ~0 as expected.

Thus, we suggest pursuing this direction in further work, after removing sib-pairs at the outset and reconducting GWAS to improve power (it is perhaps most interesting for traits that show potential stratification issues in a particular GWAS).

References

1. Wootton, R.E. *et al.* Evidence for causal effects of lifetime smoking on risk for depression and schizophrenia: a Mendelian randomisation study. *Psychological medicine* **50**, 2435-2443 (2020).
2. von Hinke, S. & Vitt, N. An analysis of the accuracy of retrospective birth location recall using sibling data. *Nature Communications* **15**, 2665 (2024).